# An Accelerated Algorithm for Stochastic Bilevel Optimization under Unbounded Smoothness

**Xiaochuan Gong**     **Jie Hao**     **Mingrui Liu**[*]
Department of Computer Science
George Mason University
{xgong2, jhao6, mingruil}@gmu.edu

## Abstract

This paper investigates a class of stochastic bilevel optimization problems where the upper-level function is nonconvex with potentially unbounded smoothness and the lower-level problem is strongly convex. These problems have significant applications in sequential data learning, such as text classification using recurrent neural networks. The unbounded smoothness is characterized by the smoothness constant of the upper-level function scaling linearly with the gradient norm, lacking a uniform upper bound. Existing state-of-the-art algorithms require $\widetilde{O}(\epsilon^{-4})$ oracle calls of stochastic gradient or Hessian/Jacobian-vector product to find an $\epsilon$-stationary point. However, it remains unclear if we can further improve the convergence rate when the assumptions for the function in the population level also hold for each random realization almost surely (e.g., Lipschitzness of each realization of the stochastic gradient). To address this issue, we propose a new Accelerated Bilevel Optimization algorithm named AccBO. The algorithm updates the upper-level variable by normalized stochastic gradient descent with recursive momentum and the lower-level variable by the stochastic Nesterov accelerated gradient descent algorithm with averaging. We prove that our algorithm achieves an oracle complexity of $\widetilde{O}(\epsilon^{-3})$ to find an $\epsilon$-stationary point, when the lower-level stochastic gradient has a small variance $O(\epsilon)$. Our proof relies on a novel lemma characterizing the dynamics of stochastic Nesterov accelerated gradient descent algorithm under distribution drift with high probability for the lower-level variable, which is of independent interest and also plays a crucial role in analyzing the hypergradient estimation error over time. Experimental results on various tasks confirm that our proposed algorithm achieves the predicted theoretical acceleration and significantly outperforms baselines in bilevel optimization. The code is available here.

## 1 Introduction

Bilevel optimization receives tremendous attention recently in the machine learning community, due to its applications in meta-learning [27, 59], hyperparameter optimization [27, 25], data hyper-cleaning [44], continual learning [7, 37], and reinforcement learning [47]. The bilevel optimization problem has the following formulation:

$$\min_{x \in \mathbb{R}^{d_x}} \Phi(x) := f(x, y^*(x)), \;\; \text{s.t.,} \;\; y^*(x) \in \arg\min_{y \in \mathbb{R}^{d_y}} g(x, y), \tag{1}$$

where $f$ and $g$ are upper-level and lower-level functions respectively. For example, in meta-learning [26, 27], $x$ denotes the layers of neural networks for shared representation learning, $y$ denotes the task-specific head encoded in the last layer, and the formulation (1) aims to learn the a

---

[*]Corresponding Author.

38th Conference on Neural Information Processing Systems (NeurIPS 2024).

common representation learning encoder $x$ such that it can be quickly adapted to downstream tasks by only updating the task-specific head $y$. In machine learning, people typically consider stochastic optimization setting such that $f(x, y) = \mathbb{E}_{\xi \sim \mathcal{D}_f}[F(x, y; \xi)]$ and $g(x, y) = \mathbb{E}_{\zeta \sim \mathcal{D}_g}[G(x, y; \zeta)]$, where $\mathcal{D}_f$ and $\mathcal{D}_g$ are the underlying unknown data distributions for $f$ and $g$ respectively, and one can access noisy observations of $f$ and $g$ based on sampling from $\mathcal{D}_f$ and $\mathcal{D}_g$.

There emerges a wave of studies for algorithmic design and analysis for solving the bilevel optimization problem (1) under different assumptions of $f$ and $g$. Most theoretical work assumes the upper-level function is smooth (i.e., gradient is Lipschitz) and nonconvex, and the lower-level function is strongly convex [30, 44, 41, 33, 48]. However, as pointed out by [75, 15], certain neural networks such as recurrent neural networks [22], long-short term memory networks [40] and transformers [65] have smoothness constants that scale with gradient norm, potentially leading to unbounded smoothness constants (i.e., gradient Lipschitz constant can be infinity). Motivated by this, Hao et al. [38] designed the first bilevel optimization algorithm to handle the cases where $f$ is nonconvex with potentially unbounded smoothness and $g$ is strongly convex. The algorithm in [38] achieves $\widetilde{O}(\epsilon^{-4})$ oracle complexity for finding an $\epsilon$-stationary point (i.e., a point $x$ such that $\|\nabla \Phi(x)\| \leq \epsilon$). Gong et al. [32] proposed an single-loop algorithm under the same setting as in [38] and also achieved $\widetilde{O}(\epsilon^{-4})$ oracle complexity. This complexity result is worse than the $\widetilde{O}(\epsilon^{-3})$ oracle complexity under the relatively easier setting where $f$ has a Lipschitz gradient, and each realization of the stochastic oracle calls is Lipschitz with respect to its argument (e.g., almost-sure Lipschitz oracle) [71, 46, 18, 34, 43]. This naturally motivates us to study the following question:

**Is it possible to improve the $\widetilde{O}(\epsilon^{-4})$ oracle complexity for bilevel optimization problems where the upper-level function is nonconvex with unbounded smoothness and the lower-level function is strongly convex, by assuming that the properties of the function at the population level also hold almost surely for each random realization?**

In this paper, we give a positive answer to this question by designing a new algorithm named AccBO with an improved oracle complexity of $\widetilde{O}(\epsilon^{-3})$, when the lower-level stochastic gradient has a small variance $O(\epsilon)$. Our algorithm is inspired by momentum-based variance reduction techniques used in nonconvex smooth optimization [18] under the almost-sure Lipschitz stochastic gradient oracle framework. The innovation of AccBO lies in its update rules: it employs normalized stochastic gradient descent with recursive momentum for the upper-level variable and stochastic Nesterov accelerated gradient descent with averaging for the lower-level variable. Our approach differs from existing accelerated bilevel optimization algorithms, such as those proposed by [71, 46] in two key ways: (i) while these algorithms use recursive momentum for the upper-level variable update, AccBO utilizes normalized recursive momentum to address the unbounded smoothness of the upper-level function; (ii) for the lower-level variable update, we use stochastic Nesterov accelerated gradient descent with averaging, in contrast to the recursive momentum method used by the other algorithms. The primary challenge in analyzing the convergence rate of AccBO arises from the need to simultaneously control errors from both upper-level and lower-level variables, given the unbounded smoothness, large learning rate, and recursive momentum in the upper-level problem. Our main contributions are summarized as follows.

- We design a new algorithm named AccBO for solving bilevel optimization problems where the upper-level function is nonconvex with unbounded smoothness and the lower-level function is strongly convex. AccBO leverages normalized recursive momentum for the upper-level variable and Nesterov momentum for the lower-level variable under the stochastic setting to achieve acceleration. To the best of our knowledge, the simultaneous usage of these two techniques in stochastic bilevel optimization is novel and has not been previously explored in the bilevel optimization literature.

- We prove that the AccBO algorithm requires $\widetilde{O}(\epsilon^{-3})$ oracle calls for finding an $\epsilon$-stationary point, when the variance of the lower-level stochastic gradient is $O(\epsilon)$. This complexity strictly improves the state-of-the-art oracle complexity for unbounded smooth nonconvex upper-level problem and strongly-convex lower-level problem as described in [38, 32] [2]. To achieve this result, we introduce novel proof techniques for analyzing the dynamics of stochastic Nesterov accelerated gradient descent under distribution drift with high probability

---

[2] Note that even if the lower-level stochastic gradient variance is $O(\epsilon)$, the oracle complexity required in [38, 32] is still $\widetilde{O}(\epsilon^{-4})$ since the oracle complexity required for their upper-level problem is already $\widetilde{O}(\epsilon^{-4})$.

for the lower-level variable, which are crucial for analyzing the hypergradient error and also of independent interest.

- We empirically verify the effectiveness of our proposed algorithm on various tasks, including deep AUC maximization and data hypercleaning. Our algorithm indeed achieves the predicted theoretical acceleration and significantly outperforms baselines in bilevel optimization.

## 2  Related Work

**Relaxed Smoothness.** The concept of relaxed smoothness was initially introduced by [75], inspired by the loss landscapes observed in recurrent neural networks and long-short term memory networks. They show that techniques such as gradient clipping and normalization could improve the performance compared with gradient descent in these scenarios. It inspired further investigations that concentrate on various aspects, including improved analysis on gradient clipping and normalization [74, 45], adaptive algorithms [15, 51, 68, 24], federated algorithms [54, 16, 17], generalized assumptions [15, 14], and recursive momentum based methods with faster rates [60, 56]. The work of [38, 32] considered a relaxed smoothness condition for the upper-level problem in the bilevel optimization setting.

**Bilevel Optimization.** Bilevel optimization refers to a special kind of optimization where one problem is embedded within another. It was first introduced by [9]. Early works developed specific bilevel optimization algorithms with asymptotic convergence analysis [67, 1, 69]. Ghadimi and Wang [30] initiated the study of non-asymptotic convergence for gradient-based methods in bilevel optimization where the upper-level problem is smooth and the lower-level problem is strongly convex. This field saw further advancements with improved complexity results [41, 44, 11, 21, 12] and fully first-order algorithms [48, 52]. There is a line of work which leverages almost-sure Lipschitz oracle (e.g., stochastic gradient) to obtain improved convergence rates of bilevel optimization algorithms [71, 46]. When the lower-level function is not strongly convex, several algorithmic framework and approximation schemes were proposed [61, 63, 49, 62, 55, 10]. The setting considered in this paper is very close to [38, 32], where the upper-level function is nonconvex and unbounded smooth, and the lower-level function is strongly convex. However, the work of [38, 32] do not have an accelerated rate $\widetilde{O}(\epsilon^{-3})$ for finding an $\epsilon$-stationary point as established in this paper.

**Nesterov Accelerated Gradient and Variants.** Nesterov Accelerated Gradient (NAG) method was introduced by [58] for deterministic convex optimization problems. The stochastic version of NAG (SNAG) was extensively studied in the literature [3, 5, 66, 13]. To the best of our knowledge, none of them provide a high probability analysis for SNAG. In the online learning setting, there is a line of work focusing on the perspective of sequential stochastic/online optimization with distributional drift [6, 70, 57, 20]. While these studies provide valuable insights into adaptive techniques and performance bounds under distributional drift, they do not explore the potential integration of such methods with bilevel optimization problems, nor do they consider the application of SNAG within this framework.

## 3  Problem Setup and Preliminaries

Define $\langle \cdot, \cdot \rangle$ and $\| \cdot \|$ as the inner product and the Euclidean norm. Throughout the paper, we use asymptotic notation $\widetilde{O}(\cdot), \widetilde{\Theta}(\cdot), \widetilde{\Omega}(\cdot)$ to hide polylog factors in $\epsilon^{-1}$ and $1/\delta$. Denote $f \colon \mathbb{R}^{d_x} \times \mathbb{R}^{d_y} \to \mathbb{R}$ as the upper-level function, and $g \colon \mathbb{R}^{d_x} \times \mathbb{R}^{d_y} \to \mathbb{R}$ as the lower-level function. The hypergradient $\nabla \Phi(x)$ has the following form [30]:

$$\nabla \Phi(x) = \nabla_x f(x, y^*(x)) - \nabla_{xy}^2 g(x, y^*(x)) \left[ \nabla_{yy}^2 g(x, y^*(x)) \right]^{-1} \nabla_y f(x, y^*(x)). \qquad (2)$$

To avoid the Hessian inverse computation, we typically use the following Neumann series to approximate the hypergradient [30, 46, 41]. In particular, for the stochastic setting, define

$$\bar{\nabla} f(x, y; \bar{\xi}) = \nabla_x F(x, y; \xi) - \frac{Q}{l_{g,1}} \nabla_{xy}^2 G(x, y; \zeta^{(0)}) \prod_{i=1}^{\mathsf{q}(Q)} \left( I - \frac{\nabla_{yy}^2 G(x, y; \zeta^{(i)})}{l_{g,1}} \right) \nabla_y F(x, y; \xi),$$

where $\mathsf{q}(Q) \sim \mathrm{Uniform}\{0, \ldots, Q-1\}$, $\bar{\xi} := \{\xi, \zeta^{(0)}, \ldots, \zeta^{(\mathsf{q}(Q))}\}$ and we use the convention that $\prod_{i=1}^{j} A_i = I$ if $j = 0$. Then $\mathbb{E}_{\bar{\xi}}[\bar{\nabla} f(x, y; \bar{\xi})]$ is a good approximation of $\nabla \Phi(x)$ if $y$ and $y^*(x)$ are close [30].

Throughout the paper, we make the following assumptions.

**Assumption 3.1** $((L_{x,0}, L_{x,1}, L_{y,0}, L_{y,1})$-smoothness [38]). *Let $z = (x, y)$ and $z' = (x', y')$, there exists $L_{x,0}, L_{x,1}, L_{y,0}, L_{y,1} > 0$ such that for all $z, z'$, if $\|z - z'\| \leq 1/\sqrt{2(L_{x,1}^2 + L_{y,1}^2)}$, then $\|\nabla_x f(z) - \nabla_x f(z')\| \leq (L_{x,0} + L_{x,1}\|\nabla_x f(z)\|)\|z - z'\|$ and $\|\nabla_y f(z) - \nabla_y f(z')\| \leq (L_{y,0} + L_{y,1}\|\nabla_y f(z)\|)\|z - z'\|$.*

**Remark**: Assumption 3.1 is introduced by [38] for describing the bilevel optimization problems with recurrent neural networks. This assumption can be regarded as a block-wise relaxed smoothness assumptions for two blocks $x$ and $y$, which is a variant of the relaxed smoothness assumption [75] and the coordinate-wise relaxed smooth assumption [15].

**Assumption 3.2.** *Suppose the followings hold for objective functions $f$ and $g$: (i) $f$ is continuously differentiable and $(L_{x,0}, L_{x,1}, L_{y,0}, L_{y,1})$-smooth in $(x, y)$; (ii) For every $x$, $\|\nabla_y f(x, y)\| \leq l_{f,0}$ for all $y$; (iii) For every $x$, $g(x, y)$ is $\mu$-strongly-convex in $y$ for $\mu > 0$; (iv) $g$ is $l_{g,1}$-smooth jointly in $(x, y)$; (v) $g$ is twice continuously differentiable, and $\nabla_{xy}^2 g, \nabla_{yy}^2 g$ are $l_{g,2}$-Lipschitz jointly in $(x, y)$.*

**Remark**: Assumption 3.2 is standard in the bilevel optimization literature [48, 38, 30]. Assumption 3.2 (i) characterizes the unbounded smoothness of the upper-level function and is empirically observed in recurrent neural networks [38].

**Assumption 3.3.** *The following stochastic estimators are unbiased and have the following properties:*

$$\mathbb{E}_{\xi \sim \mathcal{D}_f}[\|\nabla_x F(x, y; \xi) - \nabla_x f(x, y)\|^2] \leq \sigma_{f,1}^2, \quad \mathbb{E}_{\xi \sim \mathcal{D}_f}[\|\nabla_y F(x, y; \xi) - \nabla_y f(x, y)\|^2] \leq \sigma_{f,1}^2,$$

$$\Pr(\|\nabla_y G(x, y; \xi) - \nabla_y g(x, y)\| \geq \lambda) \leq 2\exp(-2\lambda^2/\sigma_{g,1}^2) \quad \forall \lambda > 0,$$

$$\mathbb{E}_{\zeta \sim \mathcal{D}_g}[\|\nabla_{xy}^2 G(x, y; \zeta) - \nabla_{xy}^2 g(x, y)\|^2] \leq \sigma_{g,2}^2, \quad \mathbb{E}_{\zeta \sim \mathcal{D}_g}[\|\nabla_{yy}^2 G(x, y; \zeta) - \nabla_{yy}^2 g(x, y)\|^2] \leq \sigma_{g,2}^2.$$

**Remark:** Assumption 3.3 assumes the stochastic oracle for the upper-level problem has bounded variance, which is standard in nonconvex stochastic optimization [28–30]. It also assumes the stochastic oracle for the lower-level problem is light-tailed, which is common for the high probability analysis for the lower-level problem [50, 39]. Note that the same assumption is also made in [38, 32] for the bilevel problems with a unbounded smooth upper-level function.

**Assumption 3.4.** $F(x, y; \xi)$ *and* $G(x, y; \zeta)$ *satisfy Assumption 3.2 for every $\xi$ and $\zeta$ almost surely.*

**Remark:** Assumption 3.4 assumes that each random realization of the upper- and lower-level functions satisfies the same property as in the population level. Note that this condition is the key to achieve an improved $\widetilde{O}(\epsilon^{-3})$ oracle complexity under various settings, including both single-level nonconvex smooth problems [23, 18, 64] and bilevel problems with nonconvex smooth upper-level objectives [71, 46]. Furthermore, this assumption is shown to be necessary for achieving improved oracle complexity in single-level problems [2].

## 4 Algorithm and Analysis

### 4.1 Main Challenges and Algorithm Design

**Main Challenges.** We begin by explaining why existing bilevel optimization algorithms and their corresponding analysis techniques are insufficient in our setting. First, most algorithms developed for bilevel optimization require the upper-level function is smooth (i.e., the gradient of the upper-level function is Lipschitz) [30, 44, 41, 71, 46, 21, 48]. They characterize the estimation error of the optimal solution for the lower-level problem, utilize an approximate hypergradient descent approach and the descent lemma for $L$-smooth functions to prove the convergence. In particular, they demonstrate that a potential function, incorporating both the function value and the bilevel error from the lower-level problem, progressively decreases in expectation. However, when the upper-level function is $(L_{x,0}, L_{x,1}, L_{y,0}, L_{y,1})$-smooth as illustrated in Assumption 3.1, the previous algorithms and analyses relying on $L$-smoothness do not work. The reason is that the hypergradient bias depends on the approximation error of the lower-level variable as well as the hypergradient itself: these elements are statistically dependent and the standard potential function argument with an expectation-based analysis would not work. To address this issue, the work of [38, 32] requires a careful high probability analysis in the unbounded smoothness setting and obtains $\widetilde{O}(\epsilon^{-4})$ oracle complexity. Such a requirement of high probability analysis prevents us from leveraging the momentum-based

---

**Algorithm 1** STOCHASTIC NESTEROV ACCELERATED GRADIENT METHOD (SNAG)

---
1: **Input:** $x, \tilde{y}_{-1}, \tilde{y}_0, \tilde{\alpha}, T_0$                          # SNAG$(x, \tilde{y}_0, \tilde{\alpha}, T_0)$
2: **for** $t = 0, 1, \ldots, T_0 - 1$ **do**
3:     Sample $\tilde{\pi}_t$ from distribution $\mathcal{D}_g$
4:     $\tilde{z}_t = \tilde{y}_t + \gamma(\tilde{y}_t - \tilde{y}_{t-1})$
5:     $\tilde{y}_{t+1} = \tilde{z}_t - \tilde{\alpha}\nabla_y G(x, \tilde{z}_t; \tilde{\pi}_t)$
6: **end for**

---

variance reduction technique for updating the lower-level variable. For example, the work [46] which has $\widetilde{O}(\epsilon^{-3})$ oracle complexity in the smooth case leverages the momentum-based variance reduction technique [18] for updating the lower-level variable with an expectation-based analysis, but it seems difficult to establish a high probability analysis for the momentum-based variance reduction algorithm in terms of the lower-level variable. Second, the recent work of Hao et al. [38] and Gong et al. [32] considered that the upper-level function is unbounded smooth and addressed this issue by performing normalized stochastic gradient with momentum for the upper-level variable and periodic updates or stochastic gradient descent for the lower-level variable, but their oracle complexity is not better than $\widetilde{O}(\epsilon^{-4})$. These facts indicate that we need new algorithm design and analysis techniques to get potential acceleration.

**Algorithm Design.** To obtain potential acceleration and enable a high probability analysis for the lower-level variable, our key idea is to update the upper-level variable by normalized stochastic gradient descent with recursive momentum and update the lower-level variable by the stochastic Nesterov accelerated gradient (SNAG) method. Different from [38, 32], the key innovation of our algorithm design is that we achieve acceleration for both upper-level and lower-level problems simultaneously but without affecting each other. The upper-level update rule can be regarded as a generalization of the acceleration technique (e.g., the momentum-based variance reduction technique) [18, 56] from single-level to bilevel problems. The main challenge is that we need to deal with the accumulated error of the recursive momentum over time due to the hypergradient bias, which is caused by the inaccurate estimation of the optimal lower-level variable. Therefore we require a very small tracking error between the iterate of the lower-level variable and the optimal lower-level solution defined by the upper-level variable (i.e., $y^*(x)$) at every iteration. This requirement is satisfied by executing SNAG method under the distribution drift for the lower-level problem, where the drift is caused by the change of the upper-level variable over time. Note that we can provide a high probability analysis of the SNAG method under distributional drift, which strictly improves the analysis of SGD under distributional drift in [19] in the small stochastic gradient noise regime.

The detailed description of our algorithm is illustrated in Algorithm 2. At the very beginning, we run a certain number of iterations of SNAG for the fixed upper-level variable $x_0$ (line 2) as the warm-start stage, and then update the lower-level variable by SNAG (line $8 \sim 20$) with averaging (line 21) and update the upper-level variable by normalized stochastic gradient descent with recursive momentum (line $23 \sim 24$). Note that we have two options for implementing SNAG. In Option I (line $8 \sim 9$), the algorithm simply runs SNAG under distribution drift caused by the sequence $\{x_t\}$. Option I is specifically designed for a particular subset of bilevel optimization problems where the lower-level function is a quadratic function. Option II (line $11 \sim 20$) is designed for a broader range of bilevel optimization problems, accommodating general strongly-convex lower-level functions. In Option II, we run SNAG with periodic updates: the lower-level update is performed for $N$ iterations only when the iteration number $t$ is a multiple of $I$.

## 4.2 Main Results

We first introduce some useful notations. Let $\sigma(\cdot)$ be the $\sigma$-algebra generated by the random variables in the argument. We define the following filtrations for $t \geq 1$: $\mathcal{F}^{\text{init}} = \sigma(\tilde{\pi}_0, \ldots, \tilde{\pi}_{T_0-1})$, $\mathcal{F}_t = \sigma(\bar{\xi}_0, \ldots, \bar{\xi}_{t-1})$, $\widetilde{\mathcal{F}}_t^1 = \sigma(\pi_0, \ldots, \pi_{t-1})$, and we also define $\widetilde{\mathcal{F}}_t^2 = \sigma(\pi_t^0, \ldots, \pi_t^{N-1})$ when $t$ is a multiple of $I$. We use $\mathbb{E}_t$, $\mathbb{E}_{\mathcal{F}_t}$ and $\mathbb{E}$ to denote the conditional expectation $\mathbb{E}[\cdot \mid \mathcal{F}_t]$, the expectation over $\mathcal{F}_t$ and the total expectation over $\mathcal{F}_T$ respectively.

**Theorem 4.1.** *Suppose Assumptions 3.1 to 3.4 hold. Let $\{x_t\}$ be the iterates produced by Algorithm 2. For any given $\delta \in (0, 1)$ and small enough $\epsilon$ (see exact choice in (54)), if $\sigma_{g,1} = O(\sqrt{\epsilon})$ as defined in (55), and we set parameters $\alpha^{\text{init}}, \alpha, \beta, \gamma, \eta, \tau, I, N, S, Q, T_0$ (see exact choices in (56), (57), (58),*

---

**Algorithm 2** ACCELERATED BILEVEL OPTIMIZATION ALGORITHM (ACCBO)

---

1: **Input:** $\alpha^{\text{init}}, \alpha, \beta, \gamma, \eta, \tau, I, S, T_0, T$, set $x_0, y_0^{\text{init}} = 0$
2: $y_0 = \text{SNAG}(x_0, y_0^{\text{init}}, \alpha^{\text{init}}, T_0)$, and set $y_{-1} = \hat{y}_0 = y_0$      # Warm-start
3: **for** $t = 0, 1, \ldots, T - 1$ **do**
4:      Sample $\mathsf{q}(Q) \sim \text{Uniform}\{0, \ldots, Q - 1\}$ and $\{\zeta_{t,s}^{(0)}, \ldots, \zeta_{t,s}^{(\mathsf{q}(Q))}\}_{s=1}^S \sim \mathcal{D}_g$
5:      Sample $\{\xi_{t,s}\}_{s=1}^S \sim \mathcal{D}_f$, denote $\bar{\xi}_t := \cup_{s=1}^S \{\mathsf{q}(Q), \xi_{t,s}, \zeta_{t,s}^{(0)}, \ldots, \zeta_{t,s}^{(\mathsf{q}(Q))}\}$
6:      # Lower-Level: Stochastic Nesterov Accelerated Gradient Descent with Averaging
7:      # Option I: from Line 8 $\sim$ 9 (for quadratic lower-level function)
8:      $z_t = y_t + \gamma(y_t - y_{t-1})$
9:      $y_{t+1} = z_t - \alpha \nabla_y G(x_t, z_t; \pi_t)$, where $\pi_t \sim \mathcal{D}_g$
10:     # Option II: from Line 11 $\sim$ 20 (for general strongly convex lower-level function)
11:     **if** $t > 0$ and $t$ is a multiple of $I$ **then**
12:        Set $y_t^0 = y_t^{-1} = y_t$
13:        **for** $j = 0, 1, \ldots, N - 1$ **do**
14:           $z_t^j = y_t^j + \gamma(y_t^j - y_t^{j-1})$
15:           $y_t^{j+1} = z_t^j - \alpha \nabla_y G(x_t, z_t^j; \pi_t^j)$, where $\pi_t^j \sim \mathcal{D}_g$
16:        **end for**
17:        $y_{t+1} = y_t^{N+1}$
18:     **else**
19:        $y_{t+1} = y_t$
20:     **end if**
21:     $\hat{y}_{t+1} = (1 - \tau)\hat{y}_t + \tau y_{t+1}$                                # Averaging
22:     # Upper-Level: Normalized Stochastic Gradient Descent with Recursive Momentum
23:     $m_t = \beta m_{t-1} + (1 - \beta)\bar{\nabla}f(x_t, \hat{y}_t; \bar{\xi}_t) + \beta(\bar{\nabla}f(x_t, \hat{y}_t; \bar{\xi}_t) - \bar{\nabla}f(x_{t-1}, \hat{y}_{t-1}; \bar{\xi}_t))$ if $t \geq 1$ else $m_0 = \bar{\nabla}f(x_0, \hat{y}_0; \bar{\xi}_0)$
24:     $x_{t+1} = x_t - \eta \frac{m_t}{\|m_t\|}$
25: **end for**

---

(59)*, and* (60)*) as*

$$\alpha^{\text{init}} = \widetilde{\Theta}(\epsilon^4), \quad \alpha = \widetilde{\Theta}(\epsilon^2), \quad 1 - \beta = \widetilde{\Theta}(\epsilon^2), \quad \eta = \widetilde{\Theta}(\epsilon^2), \quad \tau = \widetilde{\Theta}(\epsilon), \quad \gamma = O(1),$$

$$T_0 = \widetilde{O}(\epsilon^{-2}), \quad I = \widetilde{O}(\epsilon^{-1}), \quad N = \widetilde{O}(\epsilon^{-1}), \quad Q = \widetilde{O}(1), \quad S = \widetilde{O}(1).$$

*Then with probability at least* $1 - 2\delta$ *over the randomness in* $\sigma(\mathcal{F}^{\text{init}} \cup \widetilde{\mathcal{F}}_T^1)$ *(for Option I) or* $\sigma(\mathcal{F}^{\text{init}} \cup (\cup_{t \leq T} \widetilde{\mathcal{F}}_t^2))$ *(for Option II), Algorithm 2 guarantees* $\frac{1}{T}\sum_{t=1}^T \mathbb{E}\|\nabla\Phi(x_t)\| \leq 20\epsilon$ *within* $T = \frac{4d_0}{\eta\epsilon} = \widetilde{O}(\epsilon^{-3})$ *iterations, where* $d_0 := \Phi(x_0) - \inf_x \Phi(x)$ *and the expectation is taken over the randomness over* $\mathcal{F}_T$*. For Option I, it requires* $T_0 + SQT = \widetilde{O}(\epsilon^{-3})$ *oracle calls of stochastic gradient or Hessian/Jacobian vector product. For Option II, it requires* $T_0 + \frac{NT}{I} + SQT = \widetilde{O}(\epsilon^{-3})$ *oracle calls of stochastic gradient or Hessian/Jacobian vector product.*

**Remark:** Theorem 4.1 established an improved $\widetilde{O}(\epsilon^{-3})$ oracle complexity for finding an $\epsilon$-stationary point when the lower-level standard deviation $\sigma_{g,1} = O(\sqrt{\epsilon})$. This complexity result strictly improves the $\widetilde{O}(\epsilon^{-4})$ obtained by [38, 32] when the upper-level function is nonconvex and unbounded smooth. This complexity result also matches that in the single-level unbounded smooth setting [56] and is nearly optimal in terms of the dependency on $\epsilon$ [2]. The full statement of Theorem 4.1 is included in Theorem E.2.

### 4.3 Proof Sketch

In this section, we provide a roadmap of proving Theorem 4.1 and the main steps. The detailed proofs can be found in Appendix D and E. The key idea is to prove two things: (1) the lower-level iterate is very close to the optimal lower-level variable at every iteration; (2) two consecutive iterates of the lower-level iterates are close to each other. In particular, define $y_t^* = y^*(x_t)$, and we aim to prove that $\|\hat{y}_t - y_t^*\| \leq O(\epsilon)$ and $\|\hat{y}_{t+1} - \hat{y}_t\| \leq O(\epsilon^2)$ for every $t$. These two requirements are essential to control the hypergradient estimation error (i.e., $\|m_t - \nabla\Phi(x_t)\|$) caused by inaccurate estimate of the lower-level problem. Lemma 4.7 provides the guarantee for the lower-level problem, and Lemma 4.8

characterizes the hypergradient estimation error. Equipped with these two lemmas, we can adapt the momentum-based variance reduction techniques [18, 56] to the upper-level problem and prove the main theorem.

The main technical contribution of this paper is to provide a general framework for proving the convergence of SNAG under distributional drift in Section 4.3.1, which can be leveraged as a tool to control the lower-level error in bilevel optimization and derive the Lemma 4.7, as illustrated in Section 4.3.2. In particular, we can regard the change of the upper-level variable $x$ at each iteration as the distributional drift for the lower-level problem: the drift is small due to the normalization operator of the upper-level update rule and also the Lipschitzness of $y^*(x)$. Once we have the general lemma for tracking the minimizer for any fixed distributional drift over time, this lemma can be applied to our algorithm analysis and establish guarantees for the bilevel problem.

### 4.3.1 Stochastic Nesterov Accelerated Gradient Descent under Distributional Drift

In this section, we study the sequences of stochastic optimization problems $\min_{w \in \mathbb{R}^d} \phi_t(w)$ indexed by time $t \in \mathbb{N}$. We denote the minimizer and the minimal value of $\phi_t$ as $w_t^*$ and $\phi_t^*$, and we define the *minimizer drift* at time $t$ to be $\Delta_t := \|w_t^* - w_{t+1}^*\|$. With a slight abuse of notation [3], we consider the SNAG algorithm applied to the sequence $\{\phi_t\}_{t=1}^T$, where $T$ is the total number of iterations:

$$
\begin{aligned}
z_t &= w_t + \gamma(w_t - w_{t-1}) \\
w_{t+1} &= w_t + \gamma(w_t - w_{t-1}) - \alpha g_t,
\end{aligned}
\tag{3}
$$

where $g_t = \nabla \phi_t(z_t; \xi_t)$ is the stochastic gradient evaluated at $z_t$ with random sample $\xi_t$. Define $\varepsilon_t = g_t - \nabla \phi_t(z_t)$ as the stochastic gradient noise at $t$-th iteration. Define $\mathcal{H}_t = \sigma(\xi_1, \ldots, \xi_{t-1})$ as the filtration, which is the $\sigma$-algebra generated by all random variables until $t$-th iteration. We make the following assumption, which is the same as Assumption 3 in [20] for high probability analysis.

**Assumption 4.2.** *Function $\phi_t : \mathbb{R}^d \to \mathbb{R}$ is $\mu$-strongly convex and $L$-smooth for constants $\mu, L > 0$. Also, there exists constants $\Delta, \sigma > 0$ such that the drift $\Delta_t^2$ is sub-exponential conditioned on $\mathcal{H}_t$ with parameter $\Delta^2$ and the noise $\varepsilon_t$ is norm sub-Gaussian conditioned on $\mathcal{H}_t$ with parameter $\sigma/2$.*

**Lemma 4.3.** *Suppose Assumption 4.2 holds and let $\{w_t\}$ be the iterates produced by the update rule (3) with constant learning rate $\alpha \leq 1/25L$, and set $\gamma = \frac{1 - \sqrt{\mu\alpha}}{1 + \sqrt{\mu\alpha}}$. Define $\theta_t = [(w_t - w_t^*)^\top, (w_{t-1} - w_t^*)^\top]^\top \in \mathbb{R}^{2d}$, and the potential function $V_t$ as*

$$
V_t = \theta_t^\top \mathbf{P} \theta_t + \phi_t(w_t) - \phi_t(w_t^*), \quad \text{where} \quad \mathbf{P} = \frac{1}{2\alpha} \begin{bmatrix} 1 & \sqrt{\mu\alpha} - 1 \\ \sqrt{\mu\alpha} - 1 & (1 - \sqrt{\mu\alpha})^2 \end{bmatrix} \otimes \mathbf{I}_d.
$$

*Then for any given $\delta \in (0, 1)$ and all $t \geq 0$, the following holds with probability at least $1 - \delta$ over the randomness in $\mathcal{H}_t$ (here $e$ denotes the base of natural logarithms):*

*(i) (With drift) Let $\phi_t(w) := \frac{\mu}{2} \|w - w_t^*\|^2$, then $V_t \leq \left(1 - \frac{\sqrt{\mu\alpha}}{4}\right)^t V_0 + \left(\frac{5\sqrt{\alpha}\sigma^2}{\sqrt{\mu}} + \frac{80\Delta^2}{\alpha}\right) \ln \frac{eT}{\delta}$.*

*(ii) (Without drift) Let $\phi_t(w) \equiv \phi(w)$ be any general strongly convex functions with $\Delta = 0$, then*
$$
V_t \leq \left(1 - \frac{\sqrt{\mu\alpha}}{4}\right)^t V_0 + \frac{5\sqrt{\alpha}\sigma^2}{\sqrt{\mu}} \ln \frac{eT}{\delta}.
$$

**Remark**: When $\{\phi_t\}_{t=1}^T$ is a sequence of quadratic functions with moving minimizers, Lemma 4.3 provides a high probability tracking guarantee for SNAG with distributional drift, which is useful to provide guarantees for Option I in Algorithm 2. Note that this guarantee strictly improves the guarantee of stochastic gradient descent with distributional drift (e.g., [20, Theorem 6]) *in the small stochastic gradient noise regime* and therefore is of independent interest. In particular, for small $\alpha$, the decaying factor in the first term is improved from $1 - \frac{\mu\alpha}{2}$ to $1 - \frac{\sqrt{\mu\alpha}}{4}$, the drift term is improved from $\frac{\Delta^2}{\mu\alpha^2}$ to $\frac{\Delta^2}{\alpha}$, and the variance term becomes a bit worse (from $\alpha\sigma^2$ to $\frac{\sqrt{\alpha}\sigma^2}{\sqrt{\mu}}$). When $\sigma$ is small enough, the variance term becomes insignificant compared with the drift term, then Lemma 4.3 provides an improved convergence rate with high probability. To the best of our knowledge, such an improved guarantee for SNAG with distributional drift is first shown in this work. When there is no drift, Lemma 4.3 also provides a high probability guarantee for SNAG. It holds for any smooth and strongly convex function $\phi$, and it is useful to provide guarantees for Option II of Algorithm 2.

---

[3] The notation in Section 4.3.1 is independent of that in other sections, although there may be incidental overlaps in terminology.

### 4.3.2 Application of Stochastic Nesterov Accelerated Gradient to Bilevel Optimization

Inspired by [38, 32], we can regard $\phi_t(\cdot)$ as $\phi_t(\cdot) := g(x_t, \cdot)$ in the bilevel setting, and then we have $\Delta_t = \eta l_{g,1}/\mu$ for every $t$ due to the upper-level update rule and the Lipschitzness of $y^*(x)$. Therefore we can focus on the high probability analysis on the lower-level variable without worrying about the randomness from the upper-level. Throughout, we assume Assumption 3.1, 3.2, 3.3 and 3.4 hold. In addition, the failure probability $\delta \in (0, 1)$ and $\epsilon > 0$ are chosen in the same way as in Theorem 4.1.

**Lemma 4.4** (Warm-start). *Let $\{y_t^{\text{init}}\}$ be the iterates produced by line 2 of Algorithm 2. Set $\alpha^{\text{init}} = \widetilde{\Theta}(\epsilon^4)$, $\sigma_{g,1} = (\mu\alpha)^{1/4}\tilde{\sigma}_{g,1}$, and $\phi_t(y) \equiv g(x_0, y)$. Then $\|y_{T_0}^{\text{init}} - y_0^*\| \leq \sqrt{\frac{\mu\alpha}{32}}\frac{\epsilon}{L_0}$ holds with probability at least $1 - \delta$ over the randomness in $\widetilde{\mathcal{F}}^{\text{init}}$ (we denote this event as $\mathcal{E}_{\text{init}}$) in $T_0 = \widetilde{O}(\epsilon^{-2})$ iterations.*

**Remark:** Lemma 4.4 shows that for fixed initialization $x_0$, running SNAG for at most $T_0 = \widetilde{O}(\epsilon^{-2})$ iterations can guarantee that the Euclidean distance between the lower-level variable $y_{T_0}^{\text{init}}$ and the optimal solution $y^*(x_0)$ is at most $O(\epsilon)$, with high probability.

**Lemma 4.5** (Option I). *Under event $\mathcal{E}_{\text{init}}$, let $\{y_t\}$ be the iterates produced by Option I. Set $\alpha = \widetilde{\Theta}(\epsilon^2)$, $\sigma_{g,1} = (\mu\alpha)^{1/4}\tilde{\sigma}_{g,1}$, and $\phi_t(y) = g(x_t, y) = \frac{\mu}{2}\|y - y_t^*\|^2$. Then for any $t \in [T]$, Algorithm 2 guarantees with probability at least $1 - \delta$ over the randomness in $\widetilde{\mathcal{F}}_T^1$ (we denote this event as $\mathcal{E}_y^1$) that $\|y_t - y_t^*\| \leq \epsilon/2L_0$.*

**Lemma 4.6** (Option II). *Under event $\mathcal{E}_{\text{init}}$, let $\{y_t\}$ be the iterates produced by Option II. Set $\alpha = \widetilde{\Theta}(\epsilon^2)$, $N = \widetilde{O}(\epsilon^{-1})$, $I = \widetilde{O}(\epsilon^{-1})$, $\sigma_{g,1} = (\mu\alpha)^{1/4}\tilde{\sigma}_{g,1}$, and $\phi_t(y) = g(x_t, y)$ when $t$ is a multiple of $I$ (i.e., $x_t$ is fixed for each update round of Option II so $g$ can be general functions). Then for any $t \in [T]$, Algorithm 2 guarantees with probability at least $1 - \delta$ over the randomness in $\sigma(\cup_{t \leq T}\widetilde{\mathcal{F}}_t^2)$ (we denote this event as $\mathcal{E}_y^2$) that $\|y_t - y_t^*\| \leq \epsilon/L_0$.*

**Remark**: Lemma 4.5 and Lemma 4.6 show that, under event $\mathcal{E}_{\text{init}}$ and both option I and option II, the algorithm guarantees that each iterate $y_t$ is $O(\epsilon)$-close to the the optimal lower-level variable $y_t^*$ at every iteration $t$ with high probability.

**Lemma 4.7** (Averaging). *Under event $\mathcal{E}_{\text{init}} \cap \mathcal{E}_y^1$ (Option I) or $\mathcal{E}_{\text{init}} \cap \mathcal{E}_y^2$ (Option II), set $\tau = \sqrt{\mu\alpha}$ in the averaging step (line 21 of Algorithm 2). Then for any $t \geq 0$ we have $\|\hat{y}_t - y_t^*\| \leq \frac{2\epsilon}{L_0}$ and $\|\hat{y}_{t+1} - \hat{y}_t\| \leq \frac{\mu\epsilon^2}{24L_0^2\sigma_{g,1}} =: \vartheta$.*

**Remark:** Lemma 4.7 shows that after performing averaging operations over the sequence $\{y_t\}_{t=1}^T$, the averaged sequence enjoys stronger guarantees. First, each averaged iterate $\hat{y}_t$ is still $O(\epsilon)$-close to the optimal lower-level variable $y_t^*$; Second, two consecutive averaged iterates (i.e., $\hat{y}_t$ and $\hat{y}_{t+1}$) is $O(\epsilon^2)$-close to each other. The stronger guarantees are crucial to control the hypergradient estimation error as described in Lemma 4.8.

**Lemma 4.8.** *Under event $\mathcal{E}_{\text{init}} \cap \mathcal{E}_y^1$ (Option I) or $\mathcal{E}_{\text{init}} \cap \mathcal{E}_y^2$ (Option II), define $\epsilon_t = m_t - \mathbb{E}_t[\bar{\nabla}f(x_t, \hat{y}_t; \bar{\xi}_t)]$, then we have the following averaged cumulative error bound:*

$$\frac{1}{T}\sum_{t=0}^{T-1}\mathbb{E}\|\epsilon_t\| \leq \frac{\bar{\sigma}}{T(1-\beta)} + \sqrt{1-\beta}\bar{\sigma} + \frac{\bar{L}_0}{\sqrt{1-\beta}}\sqrt{\frac{2(\eta^2 + \vartheta^2)}{S}} + \bar{L}_1\sqrt{\frac{2(\eta^2 + \vartheta^2)}{S(1-\beta)}}\frac{1}{T}\sum_{t=0}^{T-1}\mathbb{E}\|\nabla\Phi(x_t)\|,$$

*where $S$ denotes the batch size, and $\bar{\sigma}, \bar{L}_0, \bar{L}_1$ are defined in Lemmas B.4 and B.6.*

**Remark:** Lemma 4.8 characterizes the upper-level hypergradient estimation error under the good event that the lower-level error can be controlled. One can choose hyperparameters appropriately such that the cumulative error (i.e., LHS) grows only sublinearly in terms of $T$, which is important for establishing the fast convergence of our algorithm.

## 5 Experiments

**Deep AUC Maximization with Recurrent Neural Networks**. AUC (Area Under the ROC Curve) [36] is a critical metric in evaluating the performance of binary classification models. It measures the ability of the model to distinguish between positive and negative classes, and

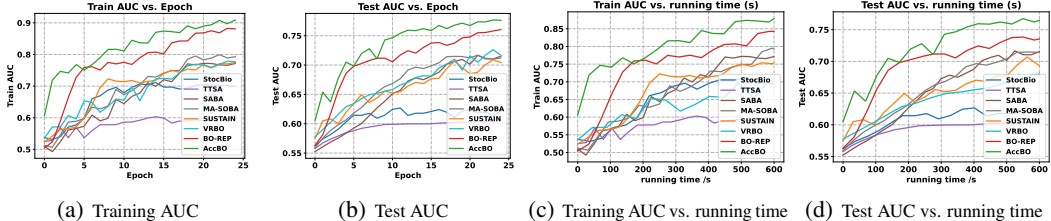

| (a) Training AUC | (b) Test AUC | (c) Training AUC vs. running time | (d) Test AUC vs. running time |

Figure 1: Results of bilevel optimization on deep AUC maximization. Figure (a), (b) are the results over epochs, and Figure (c), (d) are the results over running time.

it is defined as the probability that the prediction score of a positive example is higher than that is a negative example [35]. Deep AUC maximization [53, 72] can be formulated as a min-max optimization problem [53]: $\min_{\boldsymbol{w}\in\mathbb{R}^d,(a,b)\in\mathbb{R}^2} \max_{\alpha\in\mathbb{R}} f(\boldsymbol{w},a,b,\alpha) := \mathbb{E}_{\boldsymbol{z}}[F(\boldsymbol{w},a,b,\alpha;\boldsymbol{z})]$, where $F(\boldsymbol{w},a,b,\alpha;\boldsymbol{z}) = (1-r)(h(\boldsymbol{w};\boldsymbol{x})-a)^2\mathbb{I}_{[c=1]} + r(h(\boldsymbol{w};\boldsymbol{x})-b)^2\mathbb{I}_{[c=-1]} + 2(1+\alpha)(rh(\boldsymbol{w};\boldsymbol{x})\mathbb{I}_{[c=-1]}-(1-r)h(\boldsymbol{w};\boldsymbol{x})\mathbb{I}_{[c=1]})-r(1-r)\alpha^2$, $\boldsymbol{w}$ denotes the model parameter, $\boldsymbol{z}=(\boldsymbol{x},c)$ is the random data sample ($\boldsymbol{x}$ denote the feature vector and $c \in \{+1,-1\}$ denotes the label), $h(\boldsymbol{w},\boldsymbol{x})$ is the score function defined by a neural network, and $r = \Pr(c=1)$ denotes the ratio of positive samples in the population. This min-max formulation is an special case of the bilevel problem with $g = -f$ in (1), which can be reformulated as the following:

$$\min_{\boldsymbol{w}\in\mathbb{R}^d,(a,b)\in\mathbb{R}^2} \mathbb{E}_{\boldsymbol{z}}[F(\boldsymbol{w},a,b,\alpha^*(\boldsymbol{w},a,b);\boldsymbol{z})] \quad \text{s.t.,} \quad \alpha^*(\boldsymbol{w},a,b) \in \arg\min_{\alpha\in\mathbb{R}} -\mathbb{E}_{\boldsymbol{z}}[F(\boldsymbol{w},a,b,\alpha;\boldsymbol{z})]$$

(4)

where $(\boldsymbol{w},a,b)$ denotes the upper-level variable, and $\alpha$ denotes the lower-level variable. In this case, the lower-level is a quadratic function in terms of $\alpha$ and is strongly convex, and the upper-level function is non-convex function with potential unbounded smoothness when using a recurrent neural network as the predictive model.

We aim to perform imbalanced text classification task and maximize the AUC metric. The Deep AUC maximization experiment is performed on imbalanced Sentiment140 [31] dataset (under the license of CC BY 4.0), which is a binary text classification task. Specifically, we follow [73] to make training set imbalanced with a pre-defined imbalanced ratio ($r$), and leave the test set unchanged. Given $r$, we randomly discard the positive samples (with label 1) in original training set until the portion of positive samples equals to $r$. The imbalance ratio $r$ is set to 0.2 in our experiment, which means only 20% data is positive in the training set. We use a two-layer recurrent neural network with input dimension=300, hidden dimension=4096, and output dimension=2 for the model prediction.

We compare with some bilevel optimization baselines, including StocBio [44], TTSA [41], SABA [21], MA-SOBA [12], SUSTAIN [46], VRBO [71] and BO-REP [38]. We show the training and test AUC result with 25 epochs in (a) (b) of Figure 1 and running time in (c), (d) of Figure 1. Our algorithm AccBO achieves highest AUC score among all the baselines over epochs and running time. The running time figure shows AccBO converges to a good result faster than other baselines. The detailed parameter tuning and selection are included in Appendix G.

**Data Hypercleaning.** The Data hypercleaning task tries to learn a set of weights $\boldsymbol{\lambda}$ for the corrupted training data $\mathcal{D}_{tr}$, such that the model trained on the weighted corrupted training set can achieve good performance on the clean validation set $\mathcal{D}_{val}$, where the corrupted training set $\mathcal{D}_{tr} := \{\boldsymbol{x}_i, \bar{y}_i\}$ and the label $\bar{y}_i$ is randomly flipped to one of other labels with probability $0 < p < 1$. The data hyper-cleaning can be formulated as a bilevel optimization problem,

$$\min_{\boldsymbol{\lambda}} \frac{1}{|\mathcal{D}_{\text{val}}|} \sum_{\xi\in\mathcal{D}_{\text{val}}} \mathcal{L}(\boldsymbol{w}^*(\boldsymbol{\lambda});\xi), \text{ s.t. } \boldsymbol{w}^*(\boldsymbol{\lambda}) \in \arg\min_{\boldsymbol{w}} \frac{1}{|\mathcal{D}_{\text{tr}}|} \sum_{\zeta_i\in\mathcal{D}_{\text{tr}}} \sigma(\lambda_i)\mathcal{L}(\boldsymbol{w};\zeta_i) + c\|\boldsymbol{w}\|^2, \quad (5)$$

where $\boldsymbol{w}$ is the model parameter of a neural network, and $\sigma(x) = \frac{1}{1+e^{-x}}$ is the sigmoid function. We perform bilevel optimization algorithms on the noisy text classification dataset Stanford Natural Language Inference (SNLI) [8] (under the license of CC BY 4.0) with a three-layer recurrent neural network with input dimension=300, hidden dimension=4096, and output dimension=3 for the label prediction. Each of sentence-pairs manually labeled as entailment, contradiction, and neutral.

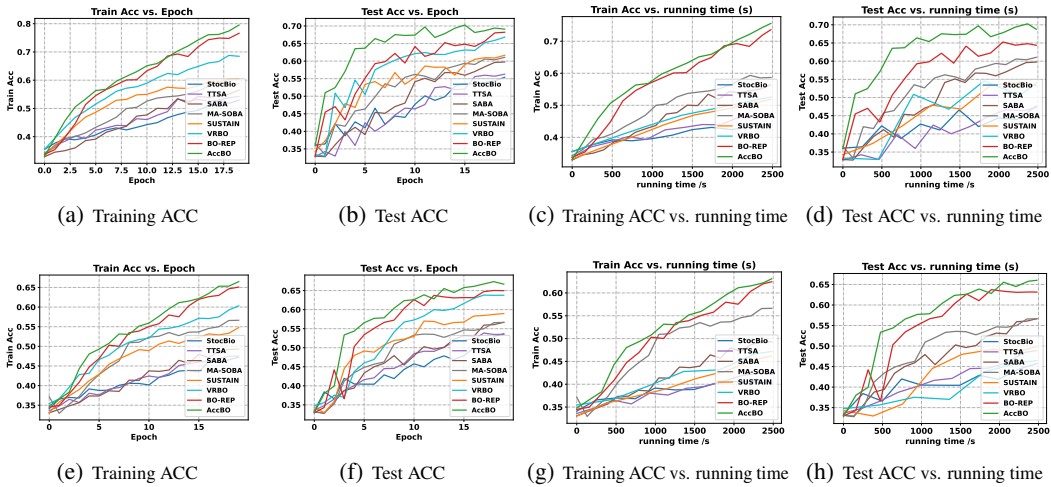

Figure 2: Results of bilevel optimization on data hyper-cleaning with $p = 0.1$. Figure (a), (b), (c), (d) are the results with noise rate $p = 0.1$ where (a), (b) are the results over epochs, and Figure (c), (d) are the results over running time. Figure (e), (f), (g), (h) are the results with noise rate $p = 0.2$.

Specifically, the label of each training data is randomly flipped to one of the other two labels with probability $p$. We set $p = 0.1$ and $p = 0.2$ in the experiments, respectively. We compare all the baselines used in the deep AUC maximization experiment. Different from the formulation (4) for the deep AUC maximization, the lower-level function in (5) is not quadratic function of the lower-level variable. Therefore we choose Option II in Algorithm 2, i.e., periodic updates for the lower-level variable. The results are presented in Figure 2 ($p = 0.1$ and $p = 0.2$). Our algorithm AccBO exhibits the highest classification accuracy on training and test set among all the bilevel baselines, and also shows a high runtime efficiency. More detailed parameter tuning and selection can be found in Appendix G. All the experiments are run on the device of NVIDIA A6000 (48GB memory) GPU and AMD EPYC 7513 32-Core CPU.

## 6  Conclusion

In this paper, we propose a new algorithm named AccBO for solving bilevel optimization problems where the upper-level is nonconvex and unbounded smooth and the lower-level problem is strongly convex. The algorithm achieved $\widetilde{O}(\epsilon^{-3})$ oracle complexity for finding an $\epsilon$-stationary point when the lower-level stochastic gradient variance is $O(\epsilon)$, which matches the rate of the state-of-the-art single-level relaxed smooth optimization [56] and is nearly optimal in terms of dependency on $\epsilon$ [2].

**Limitations.** There are two limitations of our work. One limitation of our work is that the convergence analysis for the Option I of our algorithm relies on the lower-level problem being a quadratic function: only under this case the algorithm becomes a single-loop procedure. Another limitation is that we require the lower-level stochastic gradient has variance $O(\epsilon)$. It remains unclear how to design single-loop algorithms for more general lower-level strongly convex functions and get rid of the small stochastic gradient variance assumption for the lower-level variable.

## Acknowledgments and Disclosure of Funding

We would like to thank the anonymous reviewers for their helpful comments. We would like to thank Tianbao Yang and Qihang Lin for helpful discussions for the quadratic function with distributional drift in the earlier version of our paper. This work has been supported by the Presidential Scholarship, the ORIEI seed funding, and the IDIA P3 fellowship from George Mason University, the Cisco Faculty Research Award, and NSF award #2436217, #2425687. The Computations were run on ARGO, a research computing cluster provided by the Office of Research Computing at George Mason University (URL: https://orc.gmu.edu).

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

# A Technical Lemmas

In this section, we will introduce a few useful lemmas. The following technical lemma on recursive control is crucial for providing high probability guarantee of the lower-level variables $y_t$ and $\hat{y}_t$ in Algorithm 2 at *anytime*. We follow a similar argument as in [20, Proposition 29] with a slight generalization.

**Lemma A.1** (Recursive control on MGF). *Consider scalar stochastic processes* $(V_t)$, $(V'_{t,1})$, $(V'_{t,2})$, $(D_{t,1})$, $(D_{t,2})$ *and* $(X_t)$ *on a probability space with filtration* $(\mathcal{H}_t)$, *which are linked by the inequality*

$$V_{t+1} \le \alpha_t V_t + D_{t,1}\sqrt{V'_{t,1}} + D_{t,2}\sqrt{V'_{t,2}} + X_t + \kappa_t$$

*for some deterministic constants* $\alpha_t \in (-\infty, 1]$ *and* $\kappa_t \in \mathbb{R}$. *Suppose the following properties hold.*

- $V_t, V'_{t,1}$ *and* $V'_{t,2}$ *are non-negative and* $\mathcal{H}_t$-*measurable.*

- $D_{t,i}$ *is mean-zero sub-Gaussian conditioned on* $\mathcal{H}_t$ *with deterministic parameter* $\sigma_i$, *and* $V'_{t,i} \le V_t$ *for* $i = 1, 2$:

$$\mathbb{E}[\exp(\lambda D_{t,i}) \mid \mathcal{H}_t] \le \exp(\lambda^2 \sigma_i^2/2) \quad \text{for all} \quad \lambda \in \mathbb{R}.$$

- $X_t$ *is non-negative and sub-exponential conditioned on* $\mathcal{H}_t$ *with deterministic parameter* $\nu_t$:

$$\mathbb{E}[\exp(\lambda X_t) \mid \mathcal{H}_t] \le \exp(\lambda \nu_t) \quad \text{for all} \quad 0 \le \lambda \le 1/\nu_t.$$

*Then the estimate*

$$\mathbb{E}[\exp(\lambda V_{t+1})] \le \exp(\lambda(\nu_t + \kappa_t))\mathbb{E}[\exp(\lambda(1 + \alpha_t)V_t/2)]$$

*holds for any* $\lambda$ *satisfying* $0 \le \lambda \le \min\left\{\frac{1-\alpha_t}{2(\sigma_1^2+\sigma_2^2)}, \frac{1}{2\nu_t}\right\}$.

*Proof of Lemma A.1.* For any index $t \ge 0$ and any scalar $\lambda \ge 0$, the law of total expectation implies

$$\mathbb{E}[\exp(\lambda V_{t+1})] \le \mathbb{E}\left[\exp(\lambda(\alpha_t V_t + D_{t,1}\sqrt{V'_{t,1}} + D_{t,2}\sqrt{V'_{t,2}} + X_t + \kappa_t))\right]$$

$$= \exp(\lambda \kappa_t)\mathbb{E}\left[\exp(\lambda \alpha_t V_t)\mathbb{E}\left[\exp\left(\lambda\left(D_{t,1}\sqrt{V'_{t,1}} + D_{t,2}\sqrt{V'_{t,2}}\right)\right)\exp(\lambda X_t) \mid \mathcal{H}_t\right]\right].$$

Hölder's inequality in turn yields

$$\mathbb{E}\left[\exp(\lambda \alpha_t V_t)\mathbb{E}\left[\exp\left(\lambda\left(D_{t,1}\sqrt{V'_{t,1}} + D_{t,2}\sqrt{V'_{t,2}}\right)\right)\exp(\lambda X_t) \mid \mathcal{H}_t\right]\right]$$

$$\le \sqrt{\mathbb{E}\left[\exp\left(2\lambda\left(D_{t,1}\sqrt{V'_{t,1}} + D_{t,2}\sqrt{V'_{t,2}}\right)\right) \mid \mathcal{H}_t\right] \cdot \mathbb{E}\left[\exp(2\lambda X_t) \mid \mathcal{H}_t\right]}$$

$$\le \sqrt{\exp\left(2\lambda^2(\sigma_i^2 V'_{t,1} + \sigma_i^2 V'_{t,2})\right)\exp(2\lambda \nu_t)}$$

$$\le \exp\left(\lambda^2(\sigma_1^2 + \sigma_2^2)V_t\right)\exp(\lambda \nu_t)$$

provided $0 \le \lambda \le 1/2\nu_t$, where we use $V'_{t,i} \le V_t$ for $i = 1, 2$ in the last inequality. Therefore, under the condition that

$$0 \le \lambda \le \min\left\{\frac{1 - \alpha_t}{2(\sigma_1^2 + \sigma_2^2)}, \frac{1}{2\nu_t}\right\},$$

the following estimate hold for all $t \ge 0$:

$$\mathbb{E}[\exp(\lambda V_{t+1})] \le \exp(\lambda \kappa_t)\mathbb{E}\left[\exp(\lambda \alpha_t V_t)\exp\left(\lambda^2(\sigma_1^2 + \sigma_2^2)V_t\right)\exp(\lambda \nu_t)\right]$$

$$= \exp(\lambda(\nu_t + \kappa_t))\mathbb{E}\left[\exp\left(\lambda(\alpha_t + \lambda(\sigma_1^2 + \sigma_2^2))V_t\right)\right]$$

$$\le \exp(\lambda(\nu_t + \kappa_t))\mathbb{E}[\exp(\lambda(1 + \alpha_t)V_t/2)],$$

where the last inequality follows by the given range of $\lambda$. Thus the proof is completed. $\qquad\square$

Next, we introduce the following Young's inequality beyond Euclidean norm cases. This lemma serves as an important role when dealing with distributional drift for high probability SNAG analysis.

**Lemma A.2** (Young's inequality). *For any vectors $v_1, v_2 \in \mathbb{R}^d$, positive semidefinite (PSD) matrix $\mathbf{Q} \in \mathbb{R}^{d \times d}$, and scalar $c > 0$, it holds that* [4]

$$\|v_1 + v_2\|_{\mathbf{Q}}^2 \leq (1 + c)\|v_1\|_{\mathbf{Q}}^2 + \left(1 + \frac{1}{c}\right)\|v_2\|_{\mathbf{Q}}^2.$$

*Proof of Lemma A.2.* By definition of $\|\cdot\|_{\mathbf{Q}}$, we have

$$
\begin{aligned}
\|v_1 + v_2\|_{\mathbf{Q}}^2 &= (v_1 + v_2)^\top \mathbf{Q}(v_1 + v_2) \\
&= \|v_1\|_{\mathbf{Q}}^2 + \|v_2\|_{\mathbf{Q}}^2 + 2v_1^\top \mathbf{Q} v_2.
\end{aligned}
\tag{6}
$$

Since $\mathbf{Q} \in \mathbb{R}^{d \times d}$ is PSD, let $\mathbf{Q} = \mathbf{U}\mathbf{U}^\top$ be the Cholesky decomposition, then

$$
\begin{aligned}
2v_1^\top \mathbf{Q} v_2 = 2v_1^\top \mathbf{U}\mathbf{U}^\top v_2 &= 2(\mathbf{U}^\top v_1)^\top (\mathbf{U}^\top v_2) \\
&\leq c\|\mathbf{U}^\top v_1\|^2 + \frac{1}{c}\|\mathbf{U}^\top v_2\|^2 \\
&= c\|v_1\|_{\mathbf{Q}}^2 + \frac{1}{c}\|v_2\|_{\mathbf{Q}}^2,
\end{aligned}
\tag{7}
$$

where we use Young's inequality and definition of $\|\cdot\|_{\mathbf{Q}}$ for the second and third lines, respectively. Combing (6) and (7) gives the result as claimed. □

## B  Auxiliary Lemmas for Bilevel Optimization

In this section, we provide important properties of the objective function $\Phi$ in bilevel optimization problems, as well as characterizations (such as variance and bias) for stochastic hypergradient estimator $\bar{\nabla} f(x, y; \bar{\xi})$ based on Neumann series. For readers' convenience, we only list the results here and defer the detailed proofs to Appendix F.

**Lemma B.1** (Lipschitz property, [38, Lemma 8]). *Under Assumptions 3.1 and 3.2, $y^*(x)$ is $(l_{g,1}/\mu)$-Lipschitz continuous.*

**Lemma B.2** ($(L_0, L_1)$-smoothness, [38, Lemma 9]). *Under Assumptions 3.1 and 3.2, for any $x, x'$ we have*

$$\|\nabla\Phi(x) - \nabla\Phi(x')\| \leq (L_0 + L_1\|\nabla\Phi(x')\|)\|x - x'\| \quad \text{if} \quad \|x - x'\| \leq \frac{1}{\sqrt{2(1 + l_{g,1}^2/\mu^2)(L_{x,1}^2 + L_{y,1}^2)}},$$

*where $(L_0, L_1)$-smoothness constant $L_0$ and $L_1$ are defined as*

$$L_0 = \sqrt{1 + \frac{l_{g,1}^2}{\mu^2}}\left(L_{x,0} + L_{x,1}\frac{l_{g,1}l_{f,0}}{\mu} + \frac{l_{g,1}}{\mu}(L_{y,0} + L_{y,1}l_{f,0}) + l_{f,0}\frac{l_{g,1}l_{g,2} + l_{g,2}\mu}{\mu^2}\right) \quad \text{and} \quad L_1 = \sqrt{1 + \frac{l_{g,1}^2}{\mu^2}}L_{x,1}.$$

**Lemma B.3** (Descent inequality, [38, Lemma 10]). *Suppose Assumptions 3.1 and 3.2 and 3.2 hold. Then for any $x, x'$ we have*

$$\Phi(x) \leq \Phi(x') + \langle\nabla\Phi(x'), x - x'\rangle + \frac{L_0 + L_1\|\nabla\Phi(x')\|}{2}\|x - x'\|^2 \quad \text{if} \quad \|x - x'\| \leq \frac{1}{\sqrt{2(1 + l_{g,1}^2/\mu^2)(L_{x,1}^2 + L_{y,1}^2)}}.$$

**Lemma B.4** ([46, Lemma B.1]). *Under Assumptions 3.1 to 3.4, the bias of the stochastic hypergradient estimate of the upper-level objective satisfies*

$$\|\bar{\nabla} f(x, y) - \mathbb{E}_{\bar{\xi}}[\bar{\nabla} f(x, y; \bar{\xi})]\| \leq \frac{l_{g,1}l_{f,0}}{\mu}\left(1 - \frac{\mu}{l_{g,1}}\right)^Q,$$

*where $Q$ is the number of samples chosen to approximate the Hessian inverse. Moreover, we have*

$$\mathbb{E}_{\bar{\xi}}[\|\bar{\nabla} f(x, y; \bar{\xi}) - \mathbb{E}_{\bar{\xi}}[\bar{\nabla} f(x, y; \bar{\xi})]\|^2] \leq \sigma_{f,1}^2 + \frac{3}{\mu^2}\left[(\sigma_{f,1}^2 + l_{f,0}^2)(\sigma_{g,2}^2 + 2l_{g,1}^2) + \sigma_{f,1}^2 l_{g,1}^2\right] := \bar{\sigma}^2.$$

---

[4] Here we define $\|v\|_{\mathbf{Q}} := \sqrt{v^\top \mathbf{Q} v}$ for any vector $v \in \mathbb{R}^d$ and PSD matrix $\mathbf{Q} \in \mathbb{R}^{d \times d}$.

**Lemma B.5.** *Under Assumptions 3.1 to 3.4, we have*

$$\|\bar{\nabla} f(x,y) - \nabla\Phi(x)\| \leq (\bar{L} + L_{x,1}\|\nabla\Phi(x)\|)\|y - y^*(x)\|,$$

*where constant $\bar{L}$ is defined as*

$$\bar{L} := L_{x,0} + L_{x,1}\frac{l_{g,1}l_{f,0}}{\mu} + \frac{l_{g,1}}{\mu}(L_{y,0} + L_{y,1}l_{f,0}) + l_{f,0}\frac{\mu l_{g,2} + l_{g,1}l_{g,2}}{\mu^2} \leq L_0.$$

**Lemma B.6.** *Under Assumptions 3.1 to 3.4, we have*

*(i) For any fixed $y \in \mathbb{R}^{d_y}$ and any $x, x' \in \mathbb{R}^{d_x}$,*

$$\mathbb{E}_{\bar{\xi}}\|\bar{\nabla} f(x,y;\bar{\xi}) - \bar{\nabla} f(x',y;\bar{\xi})\|^2 \leq (\bar{L}_0^2 + \bar{L}_1^2\|\nabla\Phi(x)\|^2)\|x - x'\|^2.$$

*(ii) For any fixed $x \in \mathbb{R}^{d_x}$ and any $y, y' \in \mathbb{R}^{d_y}$,*

$$\mathbb{E}_{\bar{\xi}}\|\bar{\nabla} f(x,y;\bar{\xi}) - \bar{\nabla} f(x,y';\bar{\xi})\|^2 \leq (\bar{L}_0^2 + \bar{L}_1^2\|\nabla\Phi(x)\|^2)\|y - y'\|^2.$$

*In the above expressions, we define $\bar{L}_0$ and $\bar{L}_1$ as*

$$\bar{L}_0 = \left\{ 4\left(L_{x,0} + L_{x,1}\left(\frac{l_{g,1}l_{f,0}}{\mu} + \left(L_{x,0} + \frac{L_{x,1}l_{g,1}l_{f,0}}{\mu}\right)\|y - y^*(x)\|\right)\right)^2 \right.$$

$$\left. + \frac{6Q}{2\mu l_{g,1} - \mu^2}\left(l_{g,1}^2(L_{y,0} + L_{y,1}l_{f,0})^2 + l_{f,0}^2 l_{g,2}^2 + \frac{l_{f,0}^2 l_{g,1}^2 l_{g,2}^2 Q^2}{(l_{g,1} - \mu)^2}\right) \right\}^{1/2},$$

$$\bar{L}_1 = 2L_{x,1}(1 + L_{x,1}\|y - y^*(x)\|).$$

Note that in Lemma B.6, constant $\bar{L}_0$ depends on the value of $\|y - y^*(x)\|$. When we consider this term in Algorithm 2, it turns into $\|y_t - y_t^*\|$ or $\|\hat{y}_t - y_t^*\|$, which are both as small as $O(\epsilon)$ (and thus bounded) with high probability by Lemmas 4.5 to 4.7. In other words, we can treat this term as another constant for our algorithm and analysis.

## C    Proofs of Results in Section 4.3.1

For convenience, we will restate a few concepts included in Section 4.3.1 here. We consider the sequences of stochastic optimization problems

$$\min_{w \in \mathbb{R}^d} \phi_t(w) \tag{8}$$

indexed by time $t \in \mathbb{N}$. We denote the minimizer and the minimal value of $\phi_t$ as $w_t^*$ and $\phi_t^*$, and we define the *minimizer drift* at time $t$ to be $\Delta_t := \|w_t^* - w_{t+1}^*\|$. With a slight abuse of notation, we consider the SNAG algorithm applied to the sequence $\{\phi_t\}_{t=1}^T$, where $T$ is the total number of iterations:

$$\begin{aligned} z_t &= w_t + \gamma(w_t - w_{t-1}) \\ w_{t+1} &= w_t + \gamma(w_t - w_{t-1}) - \alpha g_t, \end{aligned} \tag{9}$$

where $g_t = \nabla\phi_t(z_t;\xi_t)$ is the stochastic gradient evaluated at $z_t$ with random sample $\xi_t$. Define $\varepsilon_t = g_t - \nabla\phi_t(z_t)$ as the stochastic gradient noise at $t$-th iteration. Define $\mathcal{H}_t = \sigma(\xi_1, \ldots, \xi_{t-1})$ as the filtration, which is the $\sigma$-algebra generated by all random variables until $t$-th iteration. We will make the following standard assumption, as illustrated below [5].

**Assumption C.1.** *The sequences of time-varying functions satisfy that, each function $\phi_t : \mathbb{R}^d \to \mathbb{R}$ is $\mu$-strongly convex and $L$-smooth for some constants $\mu, L > 0$.*

**Assumption C.2** (Sub-Gaussian drift and noise)**.** *There exists constants $\Delta, \sigma > 0$ such that the following holds for all $t \geq 0$:*

*(i) (**Drift**) The drift $\Delta_t^2$ is sub-exponential conditioned on $\mathcal{H}_t$ with parameter $\Delta^2$:*

$$\mathbb{E}\left[\exp(\lambda\Delta_t^2) \mid \mathcal{H}_t\right] \leq \exp(\lambda\Delta^2) \quad \text{for all} \quad 0 \leq \lambda \leq \Delta^{-2}.$$

---

[5]Note that Assumptions C.1 and C.2 are more concrete than that in Section 4.3.1.

*(ii) (**Noise**) The noise $\varepsilon_t$ is norm sub-Gaussian conditioned on $\mathcal{H}_t$ with parameter $\sigma/2$:*

$$\Pr\{\|\varepsilon_t\| \geq \varrho \mid \mathcal{H}_t\} \leq 2\exp(-2\varrho^2/\sigma^2) \quad \textit{for all} \quad \varrho > 0.$$

The following lemma characterize the one-step improvement for stochastic Nesterov accelerated gradient method. Although part of our analysis is similar to [13, 5], our final goal is quite different: we aim to derive a careful formulation (see (13)) such that we can apply Lemma A.1 to recursively control the moment generating function of $V_t$ with distributional drift, thus leading to a high probability bound for $V_t$ at *anytime* (see Lemma C.5), while [13, 5] only show the convergence in expectation without distributional drift.

**Lemma C.3** (Distance recursion, with drift). *Suppose that Assumptions C.1 and C.2 hold. Let $\{w_t\}$ be the iterates produced by update rule (9) with constant learning rate $\alpha \leq 1/2L$ and set constants $\gamma, \rho > 0$, and matrix $\mathbf{P} \in \mathbb{R}^{2d \times 2d}$ as*

$$\gamma = \frac{1 - \sqrt{\mu\alpha}}{1 + \sqrt{\mu\alpha}}, \quad \rho^2 = 1 - \sqrt{\mu\alpha}, \quad \mathbf{P} = \frac{1}{2\alpha}\begin{bmatrix} 1 & \sqrt{\mu\alpha} - 1 \\ \sqrt{\mu\alpha} - 1 & (1 - \sqrt{\mu\alpha})^2 \end{bmatrix} \otimes \mathbf{I}_d. \tag{10}$$

*Define $\theta_t = [(w_t - w_t^*)^\top, (w_{t-1} - w_t^*)^\top]^\top \in \mathbb{R}^{2d}$, also define the potential function and $u_{t,1}, u_{t,2}$ as*

$$V_t = \theta_t^\top \mathbf{P} \theta_t + \phi_t(w_t) - \phi_t(w_t^*), \qquad u_{t,1} = \frac{w_t - w_t^*}{\|w_t - w_t^*\|}, \qquad u_{t,2} = \frac{z_t - w_t}{\|z_t - w_t\|}. \tag{11}$$

*Then for all $t \geq 0$, it holds that*

$$\begin{bmatrix} w_{t+1} - w_t^* \\ w_t - w_t^* \end{bmatrix}^\top \mathbf{P} \begin{bmatrix} w_{t+1} - w_t^* \\ w_t - w_t^* \end{bmatrix} + \phi_t(w_{t+1}) - \phi_t(w_t^*)$$
$$\leq \rho^2 V_t - \alpha(1 - L\alpha)\langle \nabla\phi_t(z_t), \varepsilon_t \rangle + \frac{L\alpha^2}{2}\|\varepsilon_t\|^2. \tag{12}$$

*Specifically, if $\phi_t(w) := \frac{\mu}{2}\|w - w_t^*\|^2$, then we have*

$$V_{t+1} \leq \left(1 - \frac{\sqrt{\mu\alpha}}{2}\right)V_t + \left(1 + \frac{\sqrt{\mu\alpha}}{4}\right)\left[-\sqrt{2\mu}\alpha(1 - L\alpha)\langle u_{t,1}, \varepsilon_t\rangle\sqrt{\frac{\mu}{2}}\|w_t - w_t^*\|\right.$$
$$\left. - \frac{2\mu\sqrt{2\alpha}}{1 + \sqrt{\mu\alpha}}\alpha(1 - L\alpha)\langle u_{t,2}, \varepsilon_t\rangle\frac{1 + \sqrt{\mu\alpha}}{2\sqrt{2\alpha}}\|z_t - w_t\| + \frac{\alpha(1 + L\alpha)}{2}\|\varepsilon_t\|^2\right] + \frac{20\mu\Delta_t^2}{\sqrt{\mu\alpha}}. \tag{13}$$

*Proof of Lemma C.3.* We first apply Lemma A.2 with

$$v_1 + v_2 = \theta_{t+1} = \begin{bmatrix} w_{t+1} - w_{t+1}^* \\ w_t - w_{t+1}^* \end{bmatrix}, \quad v_1 = \begin{bmatrix} w_{t+1} - w_t^* \\ w_t - w_t^* \end{bmatrix}, \quad v_2 = \begin{bmatrix} w_t^* - w_{t+1}^* \\ w_t^* - w_{t+1}^* \end{bmatrix}, \quad \mathbf{Q} = \mathbf{P}$$

to obtain

$$V_{t+1} = \theta_{t+1}^\top \mathbf{P} \theta_{t+1} + \phi_{t+1}(w_{t+1}) - \phi_{t+1}(w_{t+1}^*)$$
$$\leq \left(1 + \frac{\sqrt{\mu\alpha}}{4}\right)\begin{bmatrix} w_{t+1} - w_t^* \\ w_t - w_t^* \end{bmatrix}^\top \mathbf{P} \begin{bmatrix} w_{t+1} - w_t^* \\ w_t - w_t^* \end{bmatrix} + \left(1 + \frac{4}{\sqrt{\mu\alpha}}\right)\begin{bmatrix} w_t^* - w_{t+1}^* \\ w_t^* - w_{t+1}^* \end{bmatrix}^\top \mathbf{P} \begin{bmatrix} w_t^* - w_{t+1}^* \\ w_t^* - w_{t+1}^* \end{bmatrix} + \phi_{t+1}(w_{t+1}) - \phi_{t+1}(w_{t+1}^*)$$
$$= \underbrace{\left(1 + \frac{\sqrt{\mu\alpha}}{4}\right)\left\{\begin{bmatrix} w_{t+1} - w_t^* \\ w_t - w_t^* \end{bmatrix}^\top \mathbf{P} \begin{bmatrix} w_{t+1} - w_t^* \\ w_t - w_t^* \end{bmatrix} + \phi_t(w_{t+1}) - \phi_t(w_t^*)\right\}}_{(A)}$$
$$+ \underbrace{\phi_{t+1}(w_{t+1}) - \phi_{t+1}(w_{t+1}^*) - \left(1 + \frac{\sqrt{\mu\alpha}}{4}\right)(\phi_t(w_{t+1}) - \phi_t(w_t^*))}_{(B)} + \underbrace{\left(1 + \frac{4}{\sqrt{\mu\alpha}}\right)\begin{bmatrix} w_t^* - w_{t+1}^* \\ w_t^* - w_{t+1}^* \end{bmatrix}^\top \mathbf{P} \begin{bmatrix} w_t^* - w_{t+1}^* \\ w_t^* - w_{t+1}^* \end{bmatrix}}_{(C)}.$$

Now we bound terms $(A)$, $(B)$ and $(C)$ individually.

**Bounding** $(A)$.    Let us define vector $\omega_t \in \mathbb{R}^d$ and matrices $\mathbf{A} \in \mathbb{R}^{2d \times 2d}, \mathbf{B} \in \mathbb{R}^{2d \times d}$ as

$$\omega_t = \nabla \phi_t(z_t), \quad \mathbf{A} = \begin{bmatrix} 1 + \gamma & -\gamma \\ 1 & 0 \end{bmatrix} \otimes \mathbf{I}_d, \quad \mathbf{B} = \begin{bmatrix} -\alpha \\ 0 \end{bmatrix} \otimes \mathbf{I}_d.$$

By (9) we have

$$[(w_{t+1} - w_t^*)^\top, (w_t - w_t^*)^\top]^\top = \mathbf{A}\theta_t + \mathbf{B}(\omega_t + \varepsilon_t) \tag{14}$$

Since $\phi_t$ is $\mu$-strongly convex, then

$$\phi_t(w_t) - \phi_t(z_t) \geq \langle \nabla \phi_t(z_t), w_t - z_t \rangle + \frac{\mu}{2} \|w_t - z_t\|^2. \tag{15}$$

By $L$-smoothness of $\phi_t$ and the fact that $w_{t+1} = z_t - \alpha g_t$, we have

$$
\begin{aligned}
\phi_t(z_t) - \phi_t(w_{t+1}) &\geq \langle \nabla \phi_t(z_t), z_t - w_{t+1} \rangle - \frac{L}{2} \|w_{t+1} - z_t\|^2 \\
&= \alpha \langle \nabla \phi_t(z_t), g_t \rangle - \frac{L\alpha^2}{2} \|g_t\|^2 \\
&= \alpha \|\nabla \phi_t(z_t)\|^2 + \alpha \langle \nabla \phi_t(z_t), \varepsilon_t \rangle - \frac{L\alpha^2}{2} (\|\nabla \phi_t(z_t)\|^2 + 2\langle \nabla \phi_t(z_t), \varepsilon_t \rangle + \|\varepsilon_t\|^2) \\
&= \frac{\alpha}{2}(2 - L\alpha)\|\nabla \phi_t(z_t)\|^2 - \frac{L\alpha^2}{2} \|\varepsilon_t\|^2 + \alpha(1 - L\alpha)\langle \nabla \phi_t(z_t), \varepsilon_t \rangle,
\end{aligned}
\tag{16}
$$

where we use $g_t = \nabla \phi_t(z_t) + \varepsilon_t$ in the second equality. Noting that by (9) we have

$$w_t - z_t = -\gamma(w_t - w_t^*) + \gamma(w_{t-1} - w_t^*),$$

and combining (15) and (16) we obtain

$$
\begin{aligned}
\phi_t(w_t) - \phi_t(w_{t+1}) &\geq \langle \nabla \phi_t(z_t), w_t - z_t \rangle + \frac{\mu}{2} \|w_t - z_t\|^2 \\
&\quad + \frac{\alpha}{2}(2 - L\alpha)\|\nabla \phi_t(z_t)\|^2 - \frac{L\alpha^2}{2} \|\varepsilon_t\|^2 + \alpha(1 - L\alpha)\langle \nabla \phi_t(z_t), \varepsilon_t \rangle \\
&= \begin{bmatrix} \theta_t \\ \omega_t \end{bmatrix}^\top \mathbf{X}_1 \begin{bmatrix} \theta_t \\ \omega_t \end{bmatrix} - \frac{L\alpha^2}{2} \|\varepsilon_t\|^2 + \alpha(1 - L\alpha)\langle \nabla \phi_t(z_t), \varepsilon_t \rangle,
\end{aligned}
\tag{17}
$$

where matrix $\mathbf{X}_1 \in \mathbb{R}^{3d \times 3d}$ is defined as

$$\mathbf{X}_1 = \frac{1}{2} \begin{bmatrix} \mu\gamma^2 & -\mu\gamma^2 & -\gamma \\ -\mu\gamma^2 & \mu\gamma^2 & \gamma \\ -\gamma & \gamma & \alpha(2 - L\alpha) \end{bmatrix} \otimes \mathbf{I}_d.$$

Then applying the strong convexity of $\phi_t$ again gives

$$\phi_t(w_t^*) - \phi_t(z_t) \geq \langle \nabla \phi_t(z_t), w_t^* - z_t \rangle + \frac{\mu}{2} \|w_t^* - z_t\|^2. \tag{18}$$

Noting that

$$
\begin{aligned}
w_t^* - z_t &= (w_t^* - w_t) - \gamma(w_t - w_t^*) + \gamma(w_{t-1} - w_t^*) \\
&= -(1 + \gamma)(w_t - w_t^*) + \gamma(w_{t-1} - w_t^*),
\end{aligned}
$$

and combining (16) and (18) we obtain

$$
\begin{aligned}
\phi_t(w_t^*) - \phi_t(w_{t+1}) &\geq \langle \nabla \phi_t(z_t), w_t^* - z_t \rangle + \frac{\mu}{2} \|w_t^* - z_t\|^2 \\
&\quad + \frac{\alpha}{2}(2 - L\alpha)\|\nabla \phi_t(z_t)\|^2 - \frac{L\alpha^2}{2} \|\varepsilon_t\|^2 + \alpha(1 - L\alpha)\langle \nabla \phi_t(z_t), \varepsilon_t \rangle \\
&= \begin{bmatrix} \theta_t \\ \omega_t \end{bmatrix}^\top \mathbf{X}_2 \begin{bmatrix} \theta_t \\ \omega_t \end{bmatrix} - \frac{L\alpha^2}{2} \|\varepsilon_t\|^2 + \alpha(1 - L\alpha)\langle \nabla \phi_t(z_t), \varepsilon_t \rangle,
\end{aligned}
\tag{19}
$$

where matrix $\mathbf{X}_2 \in \mathbb{R}^{3d \times 3d}$ is defined as

$$\mathbf{X}_2 = \frac{1}{2} \begin{bmatrix} \mu(1 + \gamma)^2 & -\mu\gamma(1 + \gamma) & -(1 + \gamma) \\ -\mu\gamma(1 + \gamma) & \mu\gamma^2 & \gamma \\ -(1 + \gamma) & \gamma & \alpha(2 - L\alpha) \end{bmatrix} \otimes \mathbf{I}_d.$$

Next, we multiply (17) by $\rho^2$ and (19) by $1 - \rho^2$, then sum them up to get

$$\rho^2(\phi_t(w_t) - \phi_t(w_t^*)) - (\phi_t(w_{t+1}) - \phi_t(w_t^*))$$
$$\geq \begin{bmatrix} \theta_t \\ \omega_t \end{bmatrix}^\top (\rho^2 \mathbf{X}_1 + (1 - \rho^2)\mathbf{X}_2) \begin{bmatrix} \theta_t \\ \omega_t \end{bmatrix} - \frac{L\alpha^2}{2}\|\varepsilon_t\|^2 + \alpha(1 - L\alpha)\langle \nabla\phi_t(z_t), \varepsilon_t\rangle. \tag{20}$$

By [42, Section 3.1] (see also [5, Corollary 4.9], [4, Theorem 2.3]), we have the following fact:

$$\begin{bmatrix} \mathbf{A}^\top \mathbf{P}\mathbf{A} - \rho^2\mathbf{P} & \mathbf{A}^\top\mathbf{PB} \\ \mathbf{B}^\top\mathbf{PA} & \mathbf{B}^\top\mathbf{PB} \end{bmatrix} - (\rho^2\mathbf{X}_1 + (1-\rho^2)\mathbf{X}_2) \preceq 0, \tag{21}$$

which, combined with (20) yields

$$\begin{bmatrix} w_{t+1} - w_t^* \\ w_t - w_t^* \end{bmatrix}^\top \mathbf{P} \begin{bmatrix} w_{t+1} - w_t^* \\ w_t - w_t^* \end{bmatrix} - \rho^2\theta_t^\top \mathbf{P}\theta_t$$
$$= \begin{bmatrix} \theta_t \\ \omega_t \end{bmatrix}^\top \begin{bmatrix} \mathbf{A}^\top\mathbf{PA} - \rho^2\mathbf{P} & \mathbf{A}^\top\mathbf{PB} \\ \mathbf{B}^\top\mathbf{PA} & \mathbf{B}^\top\mathbf{PB} \end{bmatrix} \begin{bmatrix} \theta_t \\ \omega_t \end{bmatrix} + \varepsilon_t^\top \mathbf{B}^\top\mathbf{PB}\varepsilon_t$$
$$\leq \begin{bmatrix} \theta_t \\ \omega_t \end{bmatrix}^\top (\rho^2\mathbf{X}_1 + (1-\rho^2)\mathbf{X}_2) \begin{bmatrix} \theta_t \\ \omega_t \end{bmatrix} + \frac{\alpha}{2}\|\varepsilon_t\|^2$$
$$\leq -(\phi_t(w_{t+1}) - \phi_t(w_t^*)) + \rho^2(\phi_t(w_t) - \phi_t(w_t^*)) - \alpha(1 - L\alpha)\langle\nabla\phi_t(z_t), \varepsilon_t\rangle + \frac{\alpha + L\alpha^2}{2}\|\varepsilon_t\|^2,$$

where the first inequality uses (14) and (21), along with the fact that $\mathbf{B}^\top\mathbf{PB} = (\alpha/2)\otimes\mathbf{I}_d$ and hence $\lambda_{\max}(\mathbf{B}^\top\mathbf{PB}) = \alpha/2$; and the last inequality follows by (20). Rearrange the above inequality and by definition of the potential function $V_t$, we obtain

$$\begin{bmatrix} w_{t+1} - w_t^* \\ w_t - w_t^* \end{bmatrix}^\top \mathbf{P} \begin{bmatrix} w_{t+1} - w_t^* \\ w_t - w_t^* \end{bmatrix} + \phi_t(w_{t+1}) - \phi_t(w_t^*)$$
$$\leq \rho^2 V_t - \alpha(1 - L\alpha)\langle\nabla\phi_t(z_t), \varepsilon_t\rangle + \frac{\alpha(1 + L\alpha)}{2}\|\varepsilon_t\|^2. \tag{22}$$

Now recall that in (8) our objective function has the form of $\phi_t(w) = \frac{\mu}{2}\|w - w_t^*\|^2$, hence $\nabla\phi_t(z_t) = \mu(z_t - w_t^*)$. Plugging this into the above inequality gives

$$\begin{bmatrix} w_{t+1} - w_t^* \\ w_t - w_t^* \end{bmatrix}^\top \mathbf{P} \begin{bmatrix} w_{t+1} - w_t^* \\ w_t - w_t^* \end{bmatrix} + \phi_t(w_{t+1}) - \phi_t(w_t^*)$$
$$\leq \rho^2 V_t - \mu\alpha(1 - L\alpha)\langle z_t - w_t^*, \varepsilon_t\rangle + \frac{\alpha + L\alpha^2}{2}\|\varepsilon_t\|^2$$
$$= \rho^2 V_t - \mu\alpha(1 - L\alpha)\langle w_t - w_t^*, \varepsilon_t\rangle - \mu\alpha(1 - L\alpha)\langle z_t - w_t, \varepsilon_t\rangle + \frac{\alpha(1 + L\alpha)}{2}\|\varepsilon_t\|^2 \tag{23}$$
$$= \rho^2 V_t - \sqrt{2\mu}\alpha(1 - L\alpha)\langle u_{t,1}, \varepsilon_t\rangle\sqrt{\frac{\mu}{2}}\|w_t - w_t^*\|$$
$$\quad - \frac{2\mu\sqrt{2\alpha}}{1 + \sqrt{\mu\alpha}}\alpha(1 - L\alpha)\langle u_{t,2}, \varepsilon_t\rangle\frac{1 + \sqrt{\mu\alpha}}{2\sqrt{2\alpha}}\|z_t - w_t\| + \frac{\alpha(1 + L\alpha)}{2}\|\varepsilon_t\|^2,$$

where $u_{t,1}$ and $u_{t,2}$ are defined as

$$u_{t,1} = \frac{w_t - w_t^*}{\|w_t - w_t^*\|}, \qquad u_{t,2} = \frac{z_t - w_t}{\|z_t - w_t\|}.$$

Therefore, we conclude that

$$(A) \leq \left(1 + \frac{\sqrt{\mu\alpha}}{4}\right)\left[\rho^2 V_t - \sqrt{2\mu}\alpha(1 - L\alpha)\langle u_{t,1}, \varepsilon_t\rangle\sqrt{\frac{\mu}{2}}\|w_t - w_t^*\|\right.$$
$$\left. - \frac{2\mu\sqrt{2\alpha}}{1 + \sqrt{\mu\alpha}}\alpha(1 - L\alpha)\langle u_{t,2}, \varepsilon_t\rangle\frac{1 + \sqrt{\mu\alpha}}{2\sqrt{2\alpha}}\|z_t - w_t\| + \frac{\alpha(1 + L\alpha)}{2}\|\varepsilon_t\|^2\right]$$
$$\leq \left(1 - \frac{3\sqrt{\mu\alpha}}{4}\right)V_t + \left[-\sqrt{2\mu}\alpha(1 - L\alpha)\langle u_{t,1}, \varepsilon_t\rangle\sqrt{\frac{\mu}{2}}\|w_t - w_t^*\|\right. \tag{24}$$
$$\left. - \frac{2\mu\sqrt{2\alpha}}{1 + \sqrt{\mu\alpha}}\alpha(1 - L\alpha)\langle u_{t,2}, \varepsilon_t\rangle\frac{1 + \sqrt{\mu\alpha}}{2\sqrt{2\alpha}}\|z_t - w_t\| + \frac{\alpha(1 + L\alpha)}{2}\|\varepsilon_t\|^2\right],$$

where the last inequality follows from the definition of $\rho$ and simple calculation

$$\left(1 + \frac{\sqrt{\mu\alpha}}{4}\right)\rho^2 = \left(1 + \frac{\sqrt{\mu\alpha}}{4}\right)(1 - \sqrt{\mu\alpha}) \le 1 - \frac{3\sqrt{\mu\alpha}}{4}.$$

**Bounding** $(B)$. Recall that under distributional drift, the objective function in (8) has the form of $\phi_t(w) = \frac{\mu}{2}\|w - w_t^*\|^2$, then we have

$$\begin{aligned}
(B) &= \phi_{t+1}(w_{t+1}) - \phi_{t+1}(w_{t+1}^*) - \left(1 + \frac{\sqrt{\mu\alpha}}{4}\right)(\phi_t(w_{t+1}) - \phi_t(w_t^*)) \\
&\le \left(1 + \frac{\sqrt{\mu\alpha}}{4}\right)(\phi_{t+1}(w_{t+1}) - \phi_{t+1}(w_{t+1}^*) - \phi_t(w_{t+1}) + \phi_t(w_t^*)) \\
&= \frac{\mu}{2}\left(1 + \frac{\sqrt{\mu\alpha}}{4}\right)(\|w_{t+1} - w_{t+1}^*\|^2 - \|w_{t+1} - w_t^*\|^2) \\
&\le \frac{\mu}{2}\left(1 + \frac{\sqrt{\mu\alpha}}{4}\right)\|w_t^* - w_{t+1}^*\|\|w_{t+1} - w_t^* + w_{t+1} - w_{t+1}^*\| \\
&\le \frac{\mu}{2}\left(1 + \frac{\sqrt{\mu\alpha}}{4}\right)\Delta_t(2\|w_{t+1} - w_{t+1}^*\| + \|w_{t+1}^* - w_t^*\|)
\end{aligned}$$

Since $\phi_{t+1}$ is $\mu$-strongly convex and matrix $\mathbf{P}$ is PSD, then

$$V_{t+1} = \theta_{t+1}^\top \mathbf{P}\theta_{t+1} + \phi_{t+1}(w_{t+1}) - \phi_t(w_{t+1}^*) \ge \phi_{t+1}(w_{t+1}) - \phi_t(w_{t+1}^*)$$

$$\ge \frac{\mu}{2}\|w_{t+1} - w_{t+1}^*\|^2 \qquad \Longrightarrow \qquad \|w_{t+1} - w_{t+1}^*\| \le \sqrt{\frac{2}{\mu}}\sqrt{V_{t+1}}.$$

Plugging the above fact back into the upper bound for $(B)$ gives

$$\begin{aligned}
(B) &\le \frac{\mu}{2}\left(1 + \frac{\sqrt{\mu\alpha}}{4}\right)\Delta_t\left(2\sqrt{\frac{2}{\mu}}\sqrt{V_{t+1}} + \Delta_t\right) \\
&= \sqrt{2\mu}\Delta_t\left(1 + \frac{\sqrt{\mu\alpha}}{4}\right)\sqrt{V_{t+1}} + \frac{\mu}{2}\left(1 + \frac{\sqrt{\mu\alpha}}{4}\right)\Delta_t^2.
\end{aligned} \tag{25}$$

**Bounding** $(C)$. For this part, we handle the distributional drift. By definition of $\mathbf{P}$ in (10), we have

$$\begin{aligned}
(C) &= \left(1 + \frac{4}{\sqrt{\mu\alpha}}\right)\begin{bmatrix} w_t^* - w_{t+1}^* \\ w_t^* - w_{t+1}^* \end{bmatrix}^\top \frac{1}{2\alpha}\begin{bmatrix} \mathbf{I}_d & (\sqrt{\mu\alpha} - 1)\mathbf{I}_d \\ (\sqrt{\mu\alpha} - 1)\mathbf{I}_d & (1 - \sqrt{\mu\alpha})^2\mathbf{I}_d \end{bmatrix}\begin{bmatrix} w_t^* - w_{t+1}^* \\ w_t^* - w_{t+1}^* \end{bmatrix} \\
&= \frac{\mu}{2}\left(1 + \frac{4}{\sqrt{\mu\alpha}}\right)\Delta_t^2,
\end{aligned} \tag{26}$$

where in the last equality we use the basic algebra of block matrix multiplication and the definition of $\Delta_t = \|w_t^* - w_{t+1}^*\|$.

**Final Bound for** $V_{t+1}$. Now we are ready to derive the upper bound for $V_{t+1}$. Combining (24), (25) and (26) together yields

$$\begin{aligned}
V_{t+1} &\le (A) + (B) + (C) \\
&\le \left(1 - \frac{3\sqrt{\mu\alpha}}{4}\right)V_t + \left[-\sqrt{2\mu}\alpha(1 - L\alpha)\langle u_{t,1}, \varepsilon_t\rangle\sqrt{\frac{\mu}{2}}\|w_t - w_t^*\|\right. \\
&\quad \left. -\frac{2\mu\sqrt{2\alpha}}{1 + \sqrt{\mu\alpha}}\alpha(1 - L\alpha)\langle u_{t,2}, \varepsilon_t\rangle\frac{1 + \sqrt{\mu\alpha}}{2\sqrt{2\alpha}}\|z_t - w_t\| + \frac{\alpha(1 + L\alpha)}{2}\|\varepsilon_t\|^2\right] \\
&\quad + \sqrt{2\mu}\Delta_t\left(1 + \frac{\sqrt{\mu\alpha}}{4}\right)\sqrt{V_{t+1}} + \mu\left(1 + \frac{\sqrt{\mu\alpha}}{8} + \frac{2}{\sqrt{\mu\alpha}}\right)\Delta_t^2.
\end{aligned} \tag{27}$$

For simplicity, we define $D$ as the following

$$\begin{aligned}
D &= \left(1 - \frac{3\sqrt{\mu\alpha}}{4}\right)V_t + \left[-\sqrt{2\mu}\alpha(1 - L\alpha)\langle u_{t,1}, \varepsilon_t\rangle\sqrt{\frac{\mu}{2}}\|w_t - w_t^*\|\right. \\
&\quad \left. -\frac{2\mu\sqrt{2\alpha}}{1 + \sqrt{\mu\alpha}}\alpha(1 - L\alpha)\langle u_{t,2}, \varepsilon_t\rangle\frac{1 + \sqrt{\mu\alpha}}{2\sqrt{2\alpha}}\|z_t - w_t\| + \frac{\alpha(1 + L\alpha)}{2}\|\varepsilon_t\|^2\right] + \mu\left(1 + \frac{\sqrt{\mu\alpha}}{8} + \frac{2}{\sqrt{\mu\alpha}}\right)\Delta_t^2.
\end{aligned}$$

Hence (27) turns into

$$V_{t+1} - \sqrt{2\mu}\Delta_t\left(1 + \frac{\sqrt{\mu\alpha}}{4}\right)\sqrt{V_{t+1}} - D \le 0.$$

Solving the above inequality we get

$$\sqrt{V_{t+1}} \le \frac{1}{2}\left[\sqrt{2\mu}\Delta_t\left(1 + \frac{\sqrt{\mu\alpha}}{4}\right) + \sqrt{\left(\sqrt{2\mu}\Delta_t\left(1 + \frac{\sqrt{\mu\alpha}}{4}\right)\right)^2 + 4D}\right]$$

$$\le \sqrt{2\mu}\Delta_t\left(1 + \frac{\sqrt{\mu\alpha}}{4}\right) + \sqrt{D}$$

Then an application of Young's inequality reveals

$$V_{t+1} \le \left(1 + \frac{\sqrt{\mu\alpha}}{4}\right)D + \left(1 + \frac{4}{\sqrt{\mu\alpha}}\right)\left(\sqrt{2\mu}\Delta_t\left(1 + \frac{\sqrt{\mu\alpha}}{4}\right)\right)^2$$

$$= \left(1 + \frac{\sqrt{\mu\alpha}}{4}\right)\left\{\left(1 - \frac{3\sqrt{\mu\alpha}}{4}\right)V_t + \left[-\sqrt{2\mu}\alpha(1 - L\alpha)\langle u_{t,1}, \varepsilon_t\rangle\sqrt{\frac{\mu}{2}}\|w_t - w_t^*\|\right.\right.$$

$$\left.\left. - \frac{2\mu\sqrt{2\alpha}}{1 + \sqrt{\mu\alpha}}\alpha(1 - L\alpha)\langle u_{t,2}, \varepsilon_t\rangle\frac{1 + \sqrt{\mu\alpha}}{2\sqrt{2\alpha}}\|z_t - w_t\| + \frac{\alpha(1 + L\alpha)}{2}\|\varepsilon_t\|^2\right] + \mu\left(1 + \frac{\sqrt{\mu\alpha}}{8} + \frac{2}{\sqrt{\mu\alpha}}\right)\Delta_t^2\right\}$$

$$+ \left(1 + \frac{4}{\sqrt{\mu\alpha}}\right)2\mu\Delta_t^2\left(1 + \frac{\sqrt{\mu\alpha}}{4}\right)^2$$

$$\le \left(1 - \frac{\sqrt{\mu\alpha}}{2}\right)V_t + \left(1 + \frac{\sqrt{\mu\alpha}}{4}\right)\left[-\sqrt{2\mu}\alpha(1 - L\alpha)\langle u_{t,1}, \varepsilon_t\rangle\sqrt{\frac{\mu}{2}}\|w_t - w_t^*\|\right.$$

$$\left. - \frac{2\mu\sqrt{2\alpha}}{1 + \sqrt{\mu\alpha}}\alpha(1 - L\alpha)\langle u_{t,2}, \varepsilon_t\rangle\frac{1 + \sqrt{\mu\alpha}}{2\sqrt{2\alpha}}\|z_t - w_t\| + \frac{\alpha(1 + L\alpha)}{2}\|\varepsilon_t\|^2\right]$$

$$+ \mu\left(1 + \frac{\sqrt{\mu\alpha}}{4}\right)\left(1 + \frac{\sqrt{\mu\alpha}}{8} + \frac{2}{\sqrt{\mu\alpha}}\right)\Delta_t^2 + \left(1 + \frac{4}{\sqrt{\mu\alpha}}\right)2\mu\Delta_t^2\left(1 + \frac{\sqrt{\mu\alpha}}{4}\right)^2,$$

where we plug in the definition of $D$ for the first equality. Since the learning rate $\alpha \le 1/L$ and thus $\mu\alpha \le 1$, then we have

$$\left(1 + \frac{\sqrt{\mu\alpha}}{4}\right)\left(1 + \frac{\sqrt{\mu\alpha}}{8} + \frac{2}{\sqrt{\mu\alpha}}\right) \le \frac{125}{32\sqrt{\mu\alpha}}, \qquad 2\left(1 + \frac{4}{\sqrt{\mu\alpha}}\right)\left(1 + \frac{\sqrt{\mu\alpha}}{4}\right)^2 \le \frac{125}{8\sqrt{\mu\alpha}}.$$

Therefore, we finally conclude that

$$V_{t+1} \le \left(1 - \frac{\sqrt{\mu\alpha}}{2}\right)V_t + \left(1 + \frac{\sqrt{\mu\alpha}}{4}\right)\left[-\sqrt{2\mu}\alpha(1 - L\alpha)\langle u_{t,1}, \varepsilon_t\rangle\sqrt{\frac{\mu}{2}}\|w_t - w_t^*\|\right.$$

$$\left. - \frac{2\mu\sqrt{2\alpha}}{1 + \sqrt{\mu\alpha}}\alpha(1 - L\alpha)\langle u_{t,2}, \varepsilon_t\rangle\frac{1 + \sqrt{\mu\alpha}}{2\sqrt{2\alpha}}\|z_t - w_t\| + \frac{\alpha(1 + L\alpha)}{2}\|\varepsilon_t\|^2\right] + \frac{20\mu\Delta_t^2}{\sqrt{\mu\alpha}},$$

which is as claimed in (13). $\qquad\square$

When there is no drift, the following lemma holds for any general strongly convex functions $\phi$ in $\mathbb{R}^d$.

**Lemma C.4** (Distance recursion, without drift). *Under the same settings as in Lemma C.3 with $\phi_t(w) \equiv \phi(w)$, and $w_t^* \equiv w^*$, where $\phi(w)$ can be any general strongly functions in $\mathbb{R}^d$. We redefine $u_{t,1}, u_{t,2}$ as*

$$u_{t,1} = \frac{\nabla\phi(w_t) - \nabla\phi(w^*)}{\|\nabla\phi(w_t) - \nabla\phi(w^*)\|}, \qquad u_{t,2} = \frac{\nabla\phi(z_t) - \nabla\phi(w_t)}{\|\nabla\phi(z_t) - \nabla\phi(w_t)\|}. \tag{28}$$

*Then for all $t \ge 0$, it holds that*

$$V_{t+1} \le (1 - \sqrt{\mu\alpha})V_t - \sqrt{\frac{2}{\mu}}L\alpha(1 - L\alpha)\langle u_{t,1}, \varepsilon_t\rangle\frac{1}{L}\sqrt{\frac{\mu}{2}}\|\nabla\phi_t(w_t) - \nabla\phi_t(w^*)\|$$

$$- \frac{2L\sqrt{2\alpha}}{1 + \sqrt{\mu\alpha}}\alpha(1 - L\alpha)\langle u_{t,2}, \varepsilon_t\rangle\frac{1 + \sqrt{\mu\alpha}}{2L\sqrt{2\alpha}}\|\nabla\phi_t(z_t) - \nabla\phi_t(w_t)\| + \frac{\alpha(1 + L\alpha)}{2}\|\varepsilon_t\|^2. \tag{29}$$

*Proof of Lemma C.4.* By (12) in Lemma C.3 with $\phi_t(w) \equiv \phi(w)$ and $w_t^* \equiv w^*$, we have

$$V_{t+1} \le \rho^2 V_t - \alpha(1 - L\alpha)\langle \nabla\phi_t(z_t), \varepsilon_t \rangle + \frac{L\alpha^2}{2}\|\varepsilon_t\|^2$$

$$= \rho^2 V_t - \alpha(1 - L\alpha)\langle \nabla\phi_t(w_t) - \nabla\phi_t(w^*), \varepsilon_t \rangle - \alpha(1 - L\alpha)\langle \nabla\phi_t(z_t) - \nabla\phi_t(w_t), \varepsilon_t \rangle + \frac{\alpha(1 + L\alpha)}{2}\|\varepsilon_t\|^2$$

$$= (1 - \sqrt{\mu\alpha})V_t - \sqrt{\frac{2}{\mu}}L\alpha(1 - L\alpha)\langle u_{t,1}, \varepsilon_t \rangle \frac{1}{L}\sqrt{\frac{\mu}{2}}\|\nabla\phi_t(w_t) - \nabla\phi_t(w^*)\|$$

$$- \frac{2L\sqrt{2\alpha}}{1 + \sqrt{\mu\alpha}}\alpha(1 - L\alpha)\langle u_{t,2}, \varepsilon_t \rangle \frac{1 + \sqrt{\mu\alpha}}{2L\sqrt{2\alpha}}\|\nabla\phi_t(z_t) - \nabla\phi_t(w_t)\| + \frac{\alpha(1 + L\alpha)}{2}\|\varepsilon_t\|^2.$$

Hence the proof is completed. $\qquad\square$

The following result shows the first part of Lemma 4.3. To the best of our knowledge, this is the first high probability guarantee with improved rate for SNAG under distributional drift.

**Lemma C.5** (High-probability distance tracking, with drift). *Under the same setting as in Lemma C.3 with $\alpha \le 1/25L$, for any given $\delta \in (0, 1)$ and all $t \in [T]$, the following holds with probability at least $1 - \delta$ over the randomness in $\mathcal{H}_t$:*

$$V_t \le \left(1 - \frac{\sqrt{\mu\alpha}}{4}\right)^t V_0 + \left(\frac{5\sqrt{\alpha}\sigma^2}{\sqrt{\mu}} + \frac{80\Delta^2}{\alpha}\right)\ln\frac{eT}{\delta}. \tag{30}$$

*Proof of Lemma C.5.* We will invoke Lemma A.1 to show the results. To apply Lemma A.1, we first need to show the following two facts:

$$\text{Fact (I)}: \quad \sqrt{\frac{\mu}{2}}\|w_t - w_t^*\| \le \sqrt{V_t} \qquad \& \qquad \text{Fact (II)}: \quad \frac{1 + \sqrt{\mu\alpha}}{2\sqrt{2\alpha}}\|z_t - w_t\| \le \sqrt{V_t}. \tag{31}$$

**Fact (I) verification.** Since $\phi_t$ is $\mu$-strongly convex and matrix $\mathbf{P}$ is PSD, then

$$V_t = \theta_t^\top \mathbf{P}\theta_t + \phi_t(w_t) - \phi_t(w_t^*) \ge \phi_t(w_t) - \phi_t(w_t^*)$$

$$\ge \frac{\mu}{2}\|w_t - w_t^*\|^2 \qquad \Longrightarrow \qquad \sqrt{\frac{\mu}{2}}\|w_t - w_t^*\| \le \sqrt{V_t}.$$

**Fact (II) verification.** By definition of matrix $\mathbf{P}$ and simple calculation we have

$$\sqrt{V_t} \ge \sqrt{\theta_t^\top \mathbf{P}\theta_t} = \sqrt{\frac{1}{2\alpha}}\|(w_t - w_t^*) + (\sqrt{\mu\alpha} - 1)(w_{t-1} - w_t^*)\|$$

$$= \sqrt{\frac{1}{2\alpha}}\|(1 - \sqrt{\mu\alpha})(w_t - w_{t-1}) + \sqrt{\mu\alpha}(w_t - w_t^*)\|$$

$$\ge \sqrt{\frac{1}{2\alpha}}(1 - \sqrt{\mu\alpha})\|w_t - w_{t-1}\| - \sqrt{\frac{\mu}{2}}\|w_t - w_t^*\|$$

$$\ge \sqrt{\frac{1}{2\alpha}}(1 - \sqrt{\mu\alpha})\|w_t - w_{t-1}\| - \sqrt{V_t}.$$

Rearrange the above inequality, and recall the update rule of stochastic Nesterov accelerated gradient method, we have

$$\|z_t - w_t\| = \gamma\|w_t - w_{t-1}\| \le \frac{2\gamma\sqrt{2\alpha}}{1 - \sqrt{\mu\alpha}}\sqrt{V_t} = \frac{2\sqrt{2\alpha}}{1 + \sqrt{\mu\alpha}}\sqrt{V_t},$$

where for the last equality we use the definition of $\gamma$ as in (10). Rearrange it gives (31).

By Lemma C.3 and the choice of $\alpha \le 1/L$, we have

$$V_{t+1} \le \left(1 - \frac{\sqrt{\mu\alpha}}{2}\right)V_t + \left(1 + \frac{\sqrt{\mu\alpha}}{4}\right)\left[\sqrt{2\mu}\alpha(1 - L\alpha)\langle u_{t,1}, -\varepsilon_t \rangle\sqrt{\frac{\mu}{2}}\|w_t - w_t^*\|\right.$$

$$\left. + \frac{2\mu\sqrt{2\alpha}}{1 + \sqrt{\mu\alpha}}\alpha(1 - L\alpha)\langle u_{t,2}, -\varepsilon_t \rangle\frac{1 + \sqrt{\mu\alpha}}{2\sqrt{2\alpha}}\|z_t - w_t\| + \alpha\|\varepsilon_t\|^2\right] + \frac{20\mu\Delta_t^2}{\sqrt{\mu\alpha}}. \tag{32}$$

Note that under Assumption C.2, there exists an absolute constant $c \geq 1$ such that for all $t \geq 0$, $\|\varepsilon_t\|^2$ is sub-exponential conditioned on $\mathcal{H}_t$ with parameter $c\sigma^2$, and $\varepsilon_t$ is mean-zero sub-Gaussian conditioned on $\mathcal{H}_t$ with parameter $c\sigma$ [20, Theorem 30]. For convenience we simply let $c = 1$ here. Thus $\langle u_{t,1}, -\varepsilon_t \rangle$ is mean-zero sub-Gaussian conditioned on $\mathcal{H}_t$ with parameter $\sigma$, and $\Delta_t^2$ is sub-exponential conditioned on $\mathcal{H}_t$ with parameter $\Delta^2$ by assumption. Hence, in light of (32), we apply Lemma A.1 with

$$\mathcal{H}_t = \mathcal{H}_t, \quad V_t = V_t, \quad V_{t,1}' = \frac{\mu}{2}\|w_t - w_t^*\|^2, \quad V_{t,2}' = \frac{(1 + \sqrt{\mu\alpha})^2}{8\alpha}\|z_t - w_t\|^2,$$

$$D_{t,1} = \left(1 + \frac{\sqrt{\mu\alpha}}{4}\right)\sqrt{2\mu}\alpha(1 - L\alpha)\langle u_{t,1}, -\varepsilon_t \rangle, \quad D_{t,2} = \left(1 + \frac{\sqrt{\mu\alpha}}{4}\right)\frac{2\mu\sqrt{2\alpha}}{1 + \sqrt{\mu\alpha}}\alpha(1 - L\alpha)\langle u_{t,2}, -\varepsilon_t \rangle,$$

$$X_t = \left(1 + \frac{\sqrt{\mu\alpha}}{4}\right)\alpha\|\varepsilon_t\|^2 + \frac{20\mu\Delta_t^2}{\sqrt{\mu\alpha}}, \quad \alpha_t = 1 - \frac{\sqrt{\mu\alpha}}{2}, \quad \kappa_t = 0,$$

$$\sigma_1 = \left(1 + \frac{\sqrt{\mu\alpha}}{4}\right)\sqrt{2\mu}\alpha(1 - L\alpha)\sigma, \quad \sigma_2 = \left(1 + \frac{\sqrt{\mu\alpha}}{4}\right)\frac{2\mu\sqrt{2\alpha}}{1 + \sqrt{\mu\alpha}}\alpha(1 - L\alpha)\sigma,$$

$$\nu_t = \left(1 + \frac{\sqrt{\mu\alpha}}{4}\right)\alpha\sigma^2 + \frac{20\mu\Delta^2}{\sqrt{\mu\alpha}},$$

yielding the following recursion

$$\mathbb{E}[\exp(\lambda V_{t+1})] \leq \exp\left(\lambda\left[\left(1 + \frac{\sqrt{\mu\alpha}}{4}\right)\alpha\sigma^2 + \frac{20\mu\Delta^2}{\sqrt{\mu\alpha}}\right]\right)\mathbb{E}\left[\exp\left(\lambda\left(1 - \frac{\sqrt{\mu\alpha}}{4}\right)V_t\right)\right]$$

$$(33)$$

for all $\lambda$ satisfying

$$0 \leq \lambda \leq \min\left\{\frac{2}{125\alpha\sqrt{\mu\alpha}\sigma^2}, \frac{1}{5\alpha\sigma^2/2 + 40\mu\Delta^2/\sqrt{\mu\alpha}}\right\}.$$

We then apply (33) recursively to deduce

$$\mathbb{E}[\exp(\lambda V_{t+1})] \leq \exp\left[\lambda\left(1 - \frac{\sqrt{\mu\alpha}}{4}\right)^t V_0 + \lambda\left(\left(1 + \frac{\sqrt{\mu\alpha}}{4}\right)\alpha\sigma^2 + \frac{20\mu\Delta^2}{\sqrt{\mu\alpha}}\right)\sum_{i=0}^{t-1}\left(1 - \frac{\sqrt{\mu\alpha}}{4}\right)^i\right]$$

$$\leq \exp\left\{\lambda\left[\left(1 - \frac{\sqrt{\mu\alpha}}{4}\right)^t V_0 + 4\left(1 + \frac{\sqrt{\mu\alpha}}{4}\right)\frac{\alpha}{\sqrt{\mu\alpha}}\sigma^2 + \frac{80\Delta^2}{\alpha}\right]\right\}$$

for all $\lambda$ satisfying

$$0 \leq \lambda \leq \min\left\{\frac{2}{125\alpha\sqrt{\mu\alpha}\sigma^2}, \frac{1}{5\alpha\sigma^2/2 + 40\mu\Delta^2/\sqrt{\mu\alpha}}\right\}.$$

Moreover, setting

$$\nu := \frac{5\sqrt{\alpha}\sigma^2}{\sqrt{\mu}} + \frac{80\Delta^2}{\alpha}$$

and taking into account $\alpha \leq 1/25L$, then we have

$$4\left(1 + \frac{\sqrt{\mu\alpha}}{4}\right)\frac{\alpha}{\sqrt{\mu\alpha}}\sigma^2 + \frac{80\Delta^2}{\alpha} \leq \nu$$

and

$$\frac{1}{\nu} = \frac{1}{5\sqrt{\alpha}\sigma^2/\sqrt{\mu} + 80\Delta^2/\alpha} \leq \min\left\{\frac{2}{125\alpha\sqrt{\mu\alpha}\sigma^2}, \frac{1}{5\alpha\sigma^2/2 + 40\mu\Delta^2/\sqrt{\mu\alpha}}\right\}.$$

Thus we obtain

$$\mathbb{E}\left[\exp\left(\lambda\left(V_t - \left(1 - \frac{\sqrt{\mu\alpha}}{4}\right)^t V_0\right)\right)\right] \leq \exp(\lambda\nu) \quad \text{for all} \quad 0 \leq \lambda \leq 1/\nu.$$

Taking $\lambda = 1/\nu$ and applying Markov's inequality and union bound completes the proof. $\quad\square$

The following result shows the second part of Lemma 4.3.

**Lemma C.6** (High-probability distance tracking, without drift). *Under the same setting as in Lemma C.4 with $\alpha \le 1/25L$, for any given $\delta \in (0,1)$ and all $t \in [T]$, the following holds with probability at least $1 - \delta$ over the randomness in $\mathcal{H}_t$:*

$$V_t \le \left(1 - \frac{\sqrt{\mu\alpha}}{4}\right)^t V_0 + \frac{5\sqrt{\alpha}\sigma^2}{\sqrt{\mu}} \ln \frac{eT}{\delta}. \tag{34}$$

*Proof of Lemma C.6.* First it is easy to verify that

$$\frac{1}{L}\sqrt{\frac{\mu}{2}} \|\nabla\phi_t(w_t) - \nabla\phi_t(w^*)\| \le \sqrt{V_t} \qquad \text{and} \qquad \frac{1 + \sqrt{\mu\alpha}}{2L\sqrt{2\alpha}} \|\nabla\phi_t(z_t) - \nabla\phi_t(w_t)\| \le \sqrt{V_t}.$$

Then we apply Lemma A.1 to obtain the final result. We omit the detailed proof here since it follows the same procedure as in proof of Lemma C.5. □

# D   Proofs of Results in Section 4.3.2

We first present the following algebraic fact under suitable choice of parameters.

**Lemma D.1** (Parameter choice, informal). *For any given $\delta \in (0,1)$ and any small $\epsilon$ satisfying*

$$\epsilon \le \left(\frac{170 \cdot 32 e d_0 L_0^2 \tilde{\sigma}_{g,1}^2}{\delta\mu^2} \max\left\{\frac{l_{g,1}}{\tilde{\sigma}_{g,1}}, \frac{\bar{\sigma}}{d_0}\right\}\right)^{1/3}, \tag{35}$$

*if we set parameters $\alpha, \beta, \eta, T$ as*

$$1-\beta = \frac{\mu^2\epsilon^2}{170 \cdot 64 L_0^2 \tilde{\sigma}_{g,1}^2 \ln(P)}, \quad \eta = \min\left\{\frac{\tilde{\sigma}_{g,1}}{l_{g,1}}, \frac{d_0}{\bar{\sigma}}\right\}(1-\beta), \quad \alpha = \frac{1}{\mu}(1-\beta), \quad \sigma_{g,1} = \sqrt{\mu\alpha}\tilde{\sigma}_{g,1}, \quad T = \frac{4d_0}{\eta\epsilon}, \tag{36}$$

*where $\bar{\sigma}$ is defined in Lemma B.4, and $d_0$ and $P$ are defined as*

$$d_0 = \Phi(x_0) - \inf_{x \in \mathbb{R}^{d_x}} \Phi(x), \quad P = \left(\frac{170 \cdot 64 e d_0 L_0^2 \tilde{\sigma}_{g,1}^2}{\delta\mu^2\epsilon^3} \max\left\{\frac{l_{g,1}}{\tilde{\sigma}_{g,1}}, \frac{\bar{\sigma}}{d_0}\right\}\right)^2. \tag{37}$$

*Then the following holds for all $t \in [T]$:*

$$\left(\frac{4\alpha\tilde{\sigma}_{g,1}^2}{\mu} + \frac{160\eta^2 l_{g,1}^2}{\mu^3\alpha}\right) \ln \frac{eT}{\delta} \le \frac{\epsilon^2}{64 L_0^2}.$$

*Proof of Lemma D.1.* By Lemma B.1, we have $\Delta_t = \|y_t^* - y_{t+1}^*\| \le \frac{l_{g,1}}{\mu}\|x_t - x_{t+1}\| = \eta l_{g,1}/\mu$. Thus in our bilevel setting, we choose $\Delta = \eta l_{g,1}/\mu$, where $\Delta$ is defined in Section 4.3.1. By choice of $\alpha, \eta, T$ as in (36), we have

$$\left(\frac{10\alpha\tilde{\sigma}_{g,1}^2}{\mu} + \frac{160\eta^2 l_{g,1}^2}{\mu^3\alpha}\right) \ln \frac{eT}{\delta} = \left(\frac{10(1-\beta)\tilde{\sigma}_{g,1}^2}{\mu^2} + \frac{160\eta^2 l_{g,1}^2}{\mu^2(1-\beta)}\right) \ln\left(\frac{4ed_0}{\delta\eta\epsilon}\right)$$

$$\le \frac{170(1-\beta)\tilde{\sigma}_{g,1}^2}{\mu^2} \ln\left(\frac{4ed_0}{\delta\epsilon(1-\beta)} \max\left\{\frac{l_{g,1}}{\tilde{\sigma}_{g,1}}, \frac{\bar{\sigma}}{4d_0}\right\}\right).$$

Now we choose $\beta$ to be

$$1-\beta = \frac{\mu^2\epsilon^2}{170 \cdot 64 L_0^2 \tilde{\sigma}_{g,1}^2 \ln(P)}, \quad \text{where} \quad P = \left(\frac{170 \cdot 64 e d_0 L_0^2 \tilde{\sigma}_{g,1}^2}{\delta\mu^2\epsilon^3} \max\left\{\frac{l_{g,1}}{\tilde{\sigma}_{g,1}}, \frac{\bar{\sigma}}{d_0}\right\}\right)^2.$$

Then we have

$$\frac{170(1-\beta)\tilde{\sigma}_{g,1}^2}{\mu^2} \ln\left(\frac{4ed_0 l_{g,1}}{\delta\epsilon\tilde{\sigma}_{g,1}(1-\beta)}\right) = \frac{\epsilon^2}{64 L_0^2 \ln(P)} \ln\left(\sqrt{P}\ln(P)\right) \le \frac{\epsilon^2}{64 L_0^2},$$

where we use the fact that $\ln(\sqrt{P}\ln(P)) \le \ln(P) \le \ln^2(P)$ for any $P \ge 4$ by choice of $\epsilon$ as in (35). □

In the rest of this section, we assume Assumptions 3.1 to 3.4 hold. In addition, the failure probability $\delta \in (0,1)$ and $\epsilon > 0$ are chosen in the same way as in Theorem 4.1.

## D.1 Proof of Lemma 4.4

**Lemma D.2** (Warm-start, Restatement of Lemma 4.4). *Let $\{y_t^{\text{init}}\}$ be the iterates produced by line 2 of Algorithm 2. Set $\alpha^{\text{init}} = \mu\alpha^2 = \widetilde{\Theta}(\epsilon^4)$ with $\alpha$ defined in (36), $\sigma_{g,1} = (\mu\alpha)^{1/4}\tilde{\sigma}_{g,1}$, and $\phi_t(y) \equiv g(x_0, y)$. Then $\|y_{T_0}^{\text{init}} - y_0^*\| \leq \sqrt{\frac{\mu\alpha}{32}} \frac{\epsilon}{L_0}$ holds with probability at least $1 - \delta$ over the randomness in $\widetilde{\mathcal{F}}^{\text{init}}$ (we denote this event as $\mathcal{E}_{\text{init}}$) in $T_0 = \widetilde{O}(\epsilon^{-2})$ iterations, where*

$$T_0 = \ln\left(\frac{\mu^3\alpha^3\epsilon^2}{256L_0^2\|y_0^{\text{init}} - y_0^*\|^2}\right) \Big/ \ln\left(1 - \frac{\mu\alpha}{4}\right) = \widetilde{O}(\epsilon^{-2}). \tag{38}$$

*Proof of Lemma D.2.* By Lemmas C.6 and D.1 and $\mu$-strong convexity of $g$ in $y$, we have with probability at least $1 - \delta$ over the randomness in $\mathcal{F}^{\text{init}}$ that

$$\|y_{T_0}^{\text{init}} - y_0^*\|^2 \leq \frac{2}{\mu}\left(1 - \frac{\sqrt{\mu^2\alpha^2}}{4}\right)^{T_0} U_0^{\text{init}} + \frac{10\mu\alpha^2\tilde{\sigma}_{g,1}^2}{\mu}\ln\frac{eT_0}{\delta}$$

$$\leq \frac{2}{\mu}\left(1 - \frac{\mu\alpha}{4}\right)^{T_0} U_0^{\text{init}} + \frac{\mu\alpha\epsilon^2}{64L_0^2},$$

where the first inequality uses the choice of $\alpha^{\text{init}} = \mu\alpha^2$. By $l_{g,1}$-smoothness of $g$ we have

$$U_0^{\text{init}} \leq \frac{2 - \sqrt{\mu^2\alpha^2} + \mu^2\alpha^2}{2\mu\alpha^2}\|y_0^{\text{init}} - y_0^*\|^2 + g(x_0, y_0^{\text{init}}) - g(x_0, y_0^*)$$

$$\leq \frac{3}{2\mu\alpha^2}\|y_0^{\text{init}} - y_0^*\|^2 + \frac{l_{g,1}}{2}\|y_0^{\text{init}} - y_0^*\|^2 \leq \frac{2}{\mu\alpha^2}\|y_0^{\text{init}} - y_0^*\|^2,$$

where the last inequality uses $\alpha \leq 1/l_{g,1}$. Now we set

$$\frac{2}{\mu}\left(1 - \frac{\mu\alpha}{4}\right)^{T_0}\frac{2}{\mu\alpha^2}\|y_0^{\text{init}} - y_0^*\|^2 + \frac{\mu\alpha\epsilon^2}{64L_0^2} \leq \frac{\mu\alpha\epsilon^2}{32L_0^2},$$

which gives

$$T_0 \geq \ln\left(\frac{\mu^3\alpha^3\epsilon^2}{256L_0^2\|y_0^{\text{init}} - y_0^*\|^2}\right) \Big/ \ln\left(1 - \frac{\mu\alpha}{4}\right).$$

By choice of $\alpha$ as in (36) and simple calculation we obtain $T_0 = \widetilde{O}(\epsilon^{-2})$ when $\epsilon$ is small. $\qquad\square$

## D.2 Proof of Lemma 4.5

**Lemma D.3** (Option I, Restatement of Lemma 4.5). *Under event $\mathcal{E}_{\text{init}}$, let $\{y_t\}$ be the iterates produced by Option I. Set $\alpha = \widetilde{\Theta}(\epsilon^2)$ as in (36), $\sigma_{g,1} = (\mu\alpha)^{1/4}\tilde{\sigma}_{g,1}$, and $\phi_t(y) = g(x_t, y) = \frac{\mu}{2}\|y - y_t^*\|^2$. Then for any $t \in [T]$, Algorithm 2 guarantees with probability at least $1 - \delta$ over the randomness in $\widetilde{\mathcal{F}}_T^1$ (we denote this event as $\mathcal{E}_y^1$) that $\|y_t - y_t^*\| \leq \epsilon/2L_0$.*

*Proof of Lemma D.3.* By Lemmas C.5 and D.2 we have

$$\|y_t - y_t^*\|^2 \leq \frac{2}{\mu}\left(1 - \frac{\sqrt{\mu\alpha}}{4}\right)^t U_0 + \left(\frac{10\alpha\tilde{\sigma}_{g,1}^2}{\mu} + \frac{160\eta^2 l_{g,1}^2}{\mu^3\alpha}\right)\ln\frac{eT}{\delta}$$

$$\leq \frac{2}{\mu}\left(1 - \frac{\sqrt{\mu\alpha}}{4}\right)^t \frac{2}{\alpha}\|y_{T_0}^{\text{init}} - y_0^*\|^2 + \left(\frac{10\alpha\tilde{\sigma}_{g,1}^2}{\mu} + \frac{160\eta^2 l_{g,1}^2}{\mu^3\alpha}\right)\ln\frac{eT}{\delta}$$

$$\leq \frac{4}{\mu\alpha}\|y_{T_0}^{\text{init}} - y_0^*\|^2 + \frac{\epsilon^2}{64L_0^2} \leq \frac{\epsilon^2}{4L_0^2}.$$

Thus we conclude that for all $t \in [T]$, we have $\|y_t - y_t^*\| \leq \epsilon/2L_0$. $\qquad\square$

### D.3  Proof of Lemma 4.6

**Lemma D.4** (Option II, Restatement of Lemma 4.6). *Under event $\mathcal{E}_{\text{init}}$, let $\{y_t\}$ be the iterates produced by Option II. Set $\alpha = \widetilde{\Theta}(\epsilon^2)$ as in (36), $\sigma_{g,1} = (\mu\alpha)^{1/4}\tilde{\sigma}_{g,1}$, and run SNAG in each update round for*

$$N = \ln\left(\frac{\mu\alpha}{128}\right) \Big/ \ln\left(1 - \frac{\sqrt{\mu\alpha}}{4}\right) = \widetilde{O}(\epsilon^{-1})$$

*steps in every $I = \frac{\mu\epsilon}{2(1-\beta)L_0\tilde{\sigma}_{g,1}} = \widetilde{O}(\epsilon^{-1})$ iterations, set $\phi_t(y) = g(x_t, y)$ when $t$ is a multiple of $I$ (i.e., $x_t$ is fixed for each update round of Option II so $g$ can be general functions). Then for any $t \in [T]$, Algorithm 2 guarantees with probability at least $1 - \delta$ over the randomness in $\sigma(\cup_{t \leq T}\widetilde{\mathcal{F}}_t^2)$ (we denote this event as $\mathcal{E}_y^2$) that $\|y_t - y_t^*\| \leq \epsilon/L_0$.*

*Proof of Lemma D.4.* At the beginning of the first round, by Lemmas D.2 and D.3 we have $\|y_0 - y_0^*\| \leq \epsilon/2L_0$, then we do not update the lower-level variable until $t = I$-th iteration, then for $t = I$, we have

$$\|y_I - y_I^*\| = \|y_0 - y_I^*\| \leq \|y_0 - y_0^*\| + \sum_{i=1}^{I} \|y_i^* - y_{i-1}^*\|$$

$$\leq \frac{\epsilon}{2L_0} + \frac{\eta l_{g,1}}{\mu}I = \frac{\epsilon}{L_0},$$

where in the last equality we plug in the definition of $\eta$ and $I$. By $l_{g,1}$-smoothness of $g$ we have

$$U_I \leq \frac{2 - 2\sqrt{\mu\alpha} + \mu\alpha}{2\alpha}\|y_I - y_0^*\|^2 + g(x_I, y_I) - g(x_I, y_I^*)$$

$$\leq \frac{3}{2\alpha}\|y_I - y_I^*\|^2 + \frac{l_{g,1}}{2}\|y_I - y_I^*\|^2 \leq \frac{2\epsilon^2}{\alpha L_0^2}$$

Then for $N$ steps update in the inner loops of $t = I$-th iteration, we set

$$\frac{2}{\mu}\left(1 - \frac{\sqrt{\mu\alpha}}{4}\right)^N \frac{2\epsilon^2}{\alpha L_0^2} + \frac{\epsilon^2}{64L_0^2} \leq \frac{\epsilon^2}{16L_0^2},$$

which gives

$$N \geq \ln\left(\frac{\mu\alpha}{128}\right) \Big/ \ln\left(1 - \frac{\sqrt{\mu\alpha}}{4}\right)$$

By choice of $\alpha$ as in (36) and simple calculation we obtain $N = \widetilde{O}(\epsilon^{-1})$ when $\epsilon$ is small. Now we have

$$\|y_{I+1} - y_I^*\|^2 = \|y_I^N - y_I^*\|^2 \leq \frac{2}{\mu}\left(1 - \frac{\sqrt{\mu\alpha}}{4}\right)^N \frac{2\epsilon^2}{\alpha L_0^2} + \frac{\epsilon^2}{64L_0^2} \leq \frac{\epsilon^2}{16L_0^2},$$

which yields

$$\|y_{I+1} - y_{I+1}^*\| \leq \|y_{I+1} - y_I^*\| + \|y_I^* - y_{I+1}^*\| \leq \frac{\epsilon}{4L_0} + \frac{\eta l_{g,1}}{\mu} \leq \frac{\epsilon}{2L_0},$$

where we choose $1 - \beta$ to be small (see (56) for details) such that $\eta$ is small enough to make above inequality holds. Repeating the same process yields the result. $\square$

### D.4  Proof of Lemma 4.7

**Lemma D.5** (Averaging, Restatement of Lemma 4.7). *Under Assumptions 3.1 to 3.4 and event $\mathcal{E}_{\text{init}} \cap \mathcal{E}_y^1$ (Option I) or $\mathcal{E}_{\text{init}} \cap \mathcal{E}_y^2$ (Option II), we further set $\tau = \sqrt{\mu\alpha}$ in the averaging step (line 21 of Algorithm 2). Then for any $t \geq 0$ we have*

$$\|\hat{y}_t - y_t^*\| \leq \frac{2\epsilon}{L_0} \quad and \quad \|\hat{y}_{t+1} - \hat{y}_t\| \leq \frac{\mu\epsilon^2}{24L_0^2\tilde{\sigma}_{g,1}} =: \vartheta.$$

*Proof of Lemma D.5.* We will first show the following result by induction, i.e., for any $t \geq 0$, the averaged sequence $\{\hat{y}_t\}$ satisfies

$$\|\hat{y}_t - y_t^*\| \leq \frac{(1-\tau)\eta l_{g,1}}{\tau\mu} + \frac{\epsilon}{L_0}. \tag{39}$$

For $t = 0$, by Lemma D.2 we have

$$\|\hat{y}_0 - y_0^*\| = \|y_{T_0}^{\text{init}} - y_0^*\| \leq \sqrt{\frac{\mu\alpha}{32}}\frac{\epsilon}{L_0} = \sqrt{\frac{1-\beta}{32}}\frac{\epsilon}{L_0} \leq \frac{\epsilon}{L_0},$$

thus the base case holds. Now suppose (39) holds for some $t \geq 0$, then for time step $t + 1$ we have

$$\begin{aligned}
\|\hat{y}_{t+1} - y_{t+1}^*\| &= \|(1-\tau)(\hat{y}_t - y_{t+1}^*) + \tau(y_{t+1} - y_{t+1}^*)\| \\
&= \|(1-\tau)(\hat{y}_t - y_t^*) + (1-\tau)(y_t^* - y_{t+1}^*) + \tau(y_{t+1} - y_{t+1}^*)\| \\
&\leq (1-\tau)\|\hat{y}_t - y_t^*\| + (1-\tau)\|y_t^* - y_{t+1}^*\| + \tau\|y_{t+1} - y_{t+1}^*\| \\
&\leq (1-\tau)\left(\frac{(1-\tau)\eta l_{g,1}}{\tau\mu} + \frac{\epsilon}{L_0}\right) + \frac{(1-\tau)\eta l_{g,1}}{\mu} + \frac{\tau\epsilon}{L_0} \\
&\leq \frac{(1-\tau)\eta l_{g,1}}{\tau\mu} + \frac{\epsilon}{L_0},
\end{aligned}$$

where we use induction hypothesis in the second inequality. Therefore, we have that (39) holds for any $t \geq 0$. Also, as a consequence, for any $t \geq 0$ we have

$$\begin{aligned}
\|\hat{y}_{t+1} - \hat{y}_t\| &= \|\tau(y_{t+1} - \hat{y}_t)\| \\
&\leq \tau\|y_{t+1} - y_{t+1}^*\| + \tau\|y_{t+1}^* - y_t^*\| + \tau\|y_t^* - \hat{y}_t\| \\
&\leq \tau\left(\frac{\epsilon}{L_0} + \frac{\eta l_{g,1}}{\mu} + \frac{(1-\tau)\eta l_{g,1}}{\tau\mu} + \frac{\epsilon}{L_0}\right) \\
&= \tau\left(\frac{\eta l_{g,1}}{\tau\mu} + \frac{2\epsilon}{L_0}\right).
\end{aligned}$$

Now we plug in the definition of $\alpha, \beta, \tau, \eta$ as in (36) to obtain

$$\|\hat{y}_t - y_t^*\| \leq \frac{(1-\tau)\eta l_{g,1}}{\tau\mu} + \frac{\epsilon}{L_0} = \frac{\tilde{\sigma}_{g,1}}{\mu}\sqrt{1-\beta} + \frac{\epsilon}{L_0} \leq \frac{2\epsilon}{L_0}$$

and

$$\|\hat{y}_{t+1} - \hat{y}_t\| \leq \tau\left(\frac{\eta l_{g,1}}{\tau\mu} + \frac{2\epsilon}{L_0}\right) \leq \frac{\mu\epsilon^2}{24L_0^2\tilde{\sigma}_{g,1}}.$$

$\square$

## D.5 Proof of Lemma 4.8

**Lemma D.6** (Restatement of Lemma 4.8). *Under Assumptions 3.1 to 3.4 and event $\mathcal{E}_{\text{init}} \cap \mathcal{E}_y^1$ (Option I) or $\mathcal{E}_{\text{init}} \cap \mathcal{E}_y^2$ (Option II), define $\epsilon_t = m_t - \mathbb{E}_t[\bar{\nabla}f(x_t, \hat{y}_t; \bar{\xi}_t)]$, then we have the following averaged cumulative error bound:*

$$\frac{1}{T}\sum_{t=0}^{T-1}\mathbb{E}\|\epsilon_t\| \leq \frac{\bar{\sigma}}{T(1-\beta)} + \sqrt{1-\beta}\bar{\sigma} + \frac{\bar{L}_0}{\sqrt{1-\beta}}\sqrt{\frac{2(\eta^2+\vartheta^2)}{S}} + \bar{L}_1\sqrt{\frac{2(\eta^2+\vartheta^2)}{S(1-\beta)}}\frac{1}{T}\sum_{t=0}^{T-1}\mathbb{E}\|\nabla\Phi(x_t)\|.$$

*Proof of Lemma D.6.* Define $\epsilon_t = m_t - \mathbb{E}_t[\bar{\nabla}f(x_t, \hat{y}_t; \bar{\xi}_t)]$, also define $\tilde{\epsilon}_t$ and $\hat{\epsilon}_t$ as

$$\tilde{\epsilon}_t = \bar{\nabla}f(x_t, \hat{y}_t; \bar{\xi}_t) - \mathbb{E}_t[\bar{\nabla}f(x_t, \hat{y}_t; \bar{\xi}_t)],$$
$$\hat{\epsilon}_t = \bar{\nabla}f(x_t, \hat{y}_t; \bar{\xi}_t) - \bar{\nabla}f(x_{t-1}, \hat{y}_{t-1}; \bar{\xi}_t) - \mathbb{E}_t[\bar{\nabla}f(x_t, \hat{y}_t; \bar{\xi}_t)] + \mathbb{E}_t[\bar{\nabla}f(x_{t-1}, \hat{y}_{t-1}; \bar{\xi}_t)].$$

By definition of $\epsilon_t$, $\tilde{\epsilon}_t$ and $\hat{\epsilon}_t$, we have the following recursion for any $t \geq 0$:

$$\epsilon_{t+1} = \beta\epsilon_t + (1-\beta)\hat{\epsilon}_{t+1} + \beta\tilde{\epsilon}_{t+1}. \tag{40}$$

Then we apply (40) recursively to obtain

$$\epsilon_t = \beta^t \epsilon_0 + \beta \sum_{i=1}^{t} \beta^{t-i} \tilde{\epsilon}_i + (1-\beta) \sum_{i=1}^{t} \beta^{t-i} \hat{\epsilon}_i,$$

which by triangle inequality and total expectation gives

$$\mathbb{E}\|\epsilon_t\| = \beta^t \underbrace{\mathbb{E}\|\epsilon_0\|}_{Err_1} + (1-\beta)\,\mathbb{E}\underbrace{\left\|\sum_{i=1}^{t} \beta^{t-i}\hat{\epsilon}_i\right\|}_{Err_2} + \beta\,\mathbb{E}\underbrace{\left\|\sum_{i=1}^{t} \beta^{t-i}\tilde{\epsilon}_i\right\|}_{Err_3}. \tag{41}$$

**Bounding $Err_1$.** By definition of $\epsilon_0$ and Lemma B.4, along with Jensen's inequality, we have

$$\mathbb{E}\|\epsilon_0\| \leq \sqrt{\mathbb{E}\|\epsilon_0\|^2} \leq \bar{\sigma}. \tag{42}$$

**Bounding $Err_2$.** We apply Lemma B.4 and follow the similar procedure as in [38, Lemma D.9] to obtain

$$\mathbb{E}\left\|\sum_{i=1}^{t} \beta^{t-i}\tilde{\epsilon}_i\right\| \leq \sqrt{\mathbb{E}\left\|\sum_{i=1}^{t} \beta^{t-i}\tilde{\epsilon}_i\right\|^2} \leq \sqrt{\sum_{i=1}^{t} \beta^{2(t-i)}\mathbb{E}\|\tilde{\epsilon}_i\|^2} \leq \frac{\bar{\sigma}}{\sqrt{1-\beta}}, \tag{43}$$

where we use Jensen's inequality for the first step.

**Bounding $Err_3$.** We will first use induction to show that for $0 \leq i \leq t+1$, the following inequality holds:

$$\mathbb{E}\left[\left\|\sum_{j=1}^{t} \beta^{t-j}\hat{\epsilon}_j\right\|\right] \leq \sqrt{\frac{2(\eta^2 + \vartheta^2)}{S}}\bar{L}_1 \sum_{j=t+1-i}^{t} \beta^{t-j}\mathbb{E}\|\nabla\Phi(x_j)\| + \mathbb{E}\left[\sqrt{\frac{2(\eta^2 + \vartheta^2)\bar{L}_0^2}{S} \sum_{j=1}^{i} \beta^{2j-2} + \left\|\sum_{j=1}^{t-i} \beta^{t-j}\hat{\epsilon}_j\right\|^2}\right]. \tag{44}$$

Then it's easy to check that by setting $i = t+1$ we can obtain the bound. When $i = 0$, (44) holds obviously since

$$\mathbb{E}\left[\left\|\sum_{j=1}^{t} \beta^{t-j}\hat{\epsilon}_j\right\|\right] \leq \mathbb{E}\left[\sqrt{\left\|\sum_{j=1}^{t} \beta^{t-j}\hat{\epsilon}_j\right\|^2}\right] = \mathbb{E}\left[\left\|\sum_{j=1}^{t} \beta^{t-j}\hat{\epsilon}_j\right\|\right].$$

Hence the base case stands. Now suppose (44) holds for some $i \geq 0$, and we aim to show that (44) holds for $i + 1$. In fact, we have

$$\mathbb{E}\left[\sqrt{\frac{2(\eta^2 + \vartheta^2)\bar{L}_0^2}{S}\sum_{j=1}^{i}\beta^{2j-2} + \left\|\sum_{j=1}^{t-i}\beta^{t-j}\hat{\epsilon}_j\right\|^2}\right] \tag{45}$$

$$= \mathbb{E}_{\mathcal{F}_{t-i-1}}\left[\mathbb{E}_{t-i-1}\left[\sqrt{\frac{2(\eta^2 + \vartheta^2)\bar{L}_0^2}{S}\sum_{j=1}^{i}\beta^{2j-2} + \left\|\sum_{j=1}^{t-i}\beta^{t-j}\hat{\epsilon}_j\right\|^2}\right]\right] \tag{46}$$

$$\leq \mathbb{E}_{\mathcal{F}_{t-i-1}}\left[\mathbb{E}_{t-i-1}\left[\sqrt{\left[\frac{2(\eta^2 + \vartheta^2)\bar{L}_0^2}{S}\sum_{j=1}^{i}\beta^{2j-2} + \left\|\sum_{j=1}^{t-i}\beta^{t-j}\hat{\epsilon}_j\right\|^2\right]}\right]\right] \tag{47}$$

$$= \mathbb{E}_{\mathcal{F}_{t-i-1}}\left[\mathbb{E}_{t-i-1}\left[\sqrt{\left[\frac{2(\eta^2 + \vartheta^2)\bar{L}_0^2}{S}\sum_{j=1}^{i}\beta^{2j-2} + \beta^{2i}\|\hat{\epsilon}_{t-i}\|^2 + \left\|\sum_{j=1}^{t-i-1}\beta^{t-j}\hat{\epsilon}_j\right\|^2\right]}\right]\right] \tag{48}$$

$$\leq \mathbb{E}_{\mathcal{F}_{t-i-1}}\left[\mathbb{E}_{t-i-1}\left[\sqrt{\left[\frac{2(\eta^2 + \vartheta^2)\bar{L}_0^2}{S}\sum_{j=1}^{i}\beta^{2j-2} + \frac{\beta^{2i}}{S}2(\bar{L}_0^2 + \bar{L}_1^2\|\nabla\Phi(x_{t-i})\|^2)(\eta^2 + \vartheta^2) + \left\|\sum_{j=1}^{t-i-1}\beta^{t-j}\hat{\epsilon}_j\right\|^2\right]}\right]\right] \tag{49}$$

$$= \mathbb{E}_{\mathcal{F}_{t-i-1}}\left[\sqrt{\frac{2\beta^{2i}}{S}\bar{L}_1^2(\eta^2 + \vartheta^2)\|\nabla\Phi(x_{t-i})\|^2 + \frac{2(\eta^2 + \vartheta^2)\bar{L}_0^2}{S}\sum_{j=1}^{i+1}\beta^{2j-2} + \left\|\sum_{j=1}^{t-i-1}\beta^{t-j}\hat{\epsilon}_j\right\|^2}\right] \tag{50}$$

$$\leq \mathbb{E}_{\mathcal{F}_{t-i-1}}\left[\sqrt{\frac{2(\eta^2 + \vartheta^2)}{S}}\beta^i\bar{L}_1\|\nabla\Phi(x_{t-i})\| + \sqrt{\frac{2(\eta^2 + \vartheta^2)\bar{L}_0^2}{S}\sum_{j=1}^{i+1}\beta^{2j-2} + \left\|\sum_{j=1}^{t-i-1}\beta^{t-j}\hat{\epsilon}_j\right\|^2}\right], \tag{51}$$

where (46) follows by law of total expectation, (47) follows by Jensen's inequality, (48) uses the fact that $\hat{\epsilon}_j$ for $j < t - i$ are $\mathcal{F}_{t-i-1}$-measurable, and are uncorrelated with $\hat{\epsilon}_{t-i}$; for (49) we use Lemmas B.6 and D.5 to derive

$$\mathbb{E}_{t-i-1}[\|\hat{\epsilon}_{t-i}\|^2] = \mathbb{E}_{t-i-1}\left[\|\bar{\nabla}f(x_{t-i}, \hat{y}_{t-i}; \bar{\xi}_{t-i}) - \bar{\nabla}f(x_{t-i-1}, \hat{y}_{t-i-1}; \bar{\xi}_{t-i})\right.$$
$$\left. -\mathbb{E}_{t-i}[\bar{\nabla}f(x_{t-i}, \hat{y}_{t-i}; \bar{\xi}_{t-i})] + \mathbb{E}_{t-i}[\bar{\nabla}f(x_{t-i-1}, \hat{y}_{t-i-1}; \bar{\xi}_{t-i})]\|^2\right]$$
$$\leq \mathbb{E}_{t-i-1}\left[\|\bar{\nabla}f(x_{t-i}, \hat{y}_{t-i}; \bar{\xi}_{t-i}) - \bar{\nabla}f(x_{t-i-1}, \hat{y}_{t-i-1}; \bar{\xi}_{t-i})\|^2\right]$$
$$\leq 2\mathbb{E}_{t-i-1}\left[\|\bar{\nabla}f(x_{t-i}, \hat{y}_{t-i}; \bar{\xi}_{t-i}) - \bar{\nabla}f(x_{t-i}, \hat{y}_{t-i-1}; \bar{\xi}_{t-i})\|^2\right]$$
$$+ 2\mathbb{E}_{t-i-1}\left[\|\bar{\nabla}f(x_{t-i}, \hat{y}_{t-i-1}; \bar{\xi}_{t-i}) - \bar{\nabla}f(x_{t-i-1}, \hat{y}_{t-i-1}; \bar{\xi}_{t-i})\|^2\right]$$
$$\leq \frac{2}{S}(\bar{L}_0^2 + \bar{L}_1^2\|\nabla\Phi(x_{t-i})\|^2)(\|\hat{y}_{t-i} - \hat{y}_{t-i-1}\|^2 + \|x_{t-i} - x_{t-i-1}\|^2)$$
$$= \frac{2}{S}(\bar{L}_0^2 + \bar{L}_1^2\|\nabla\Phi(x_{t-i})\|^2)(\eta^2 + \vartheta^2).$$

And (50) follows from the fact that $x_{t-i}$ is $\mathcal{F}_{t-i-1}$-measurable, (51) uses $\sqrt{a + b} \leq \sqrt{a} + \sqrt{b}$ for $a, b \geq 0$. Hence the induction proof is completed. We set $i = t + 1$ to obtain

$$\mathbb{E}\left[\left\|\sum_{i=1}^{t}\beta^{t-i}\hat{\epsilon}_i\right\|\right] \leq \sqrt{\frac{2(\eta^2 + \vartheta^2)}{S}}\bar{L}_1\sum_{i=0}^{t}\beta^{t-i}\mathbb{E}\|\nabla\Phi(x_i)\| + \sqrt{\frac{2(\eta^2 + \vartheta^2)\bar{L}_0^2}{S}\sum_{i=0}^{t}\beta^{2i}}$$
$$\leq \sqrt{\frac{2(\eta^2 + \vartheta^2)}{S}}\bar{L}_1\sum_{i=0}^{t}\beta^{t-i}\mathbb{E}\|\nabla\Phi(x_i)\| + \frac{\bar{L}_0}{\sqrt{1-\beta}}\sqrt{\frac{2(\eta^2 + \vartheta^2)}{S}}. \tag{52}$$

**Final Bound.**    Combining (42), (43) and (52) yields

$$\mathbb{E}\|\epsilon_t\| \leq \beta^t\bar{\sigma} + \sqrt{1-\beta}\bar{\sigma} + \frac{\bar{L}_0}{\sqrt{1-\beta}}\sqrt{\frac{2(\eta^2 + \vartheta^2)}{S}} + \sqrt{\frac{2(\eta^2 + \vartheta^2)}{S}}\bar{L}_1\sum_{i=0}^{t}\beta^{t-i}\mathbb{E}\|\nabla\Phi(x_i)\|.$$

Taking summation and dividing $1/T$ on both sides gives the final result

$$\frac{1}{T}\sum_{t=0}^{T-1}\mathbb{E}\|\epsilon_t\| \leq \frac{\bar{\sigma}}{T(1-\beta)} + \sqrt{1-\beta}\bar{\sigma} + \frac{\bar{L}_0}{\sqrt{1-\beta}}\sqrt{\frac{2(\eta^2+\vartheta^2)}{S}} + \bar{L}_1\sqrt{\frac{2(\eta^2+\vartheta^2)}{S(1-\beta)}}\frac{1}{T}\sum_{t=0}^{T-1}\mathbb{E}\|\nabla\Phi(x_t)\|.$$

$\square$

## E  Proof of Theorem 4.1

Before starting the proof of main results, i.e., Theorem 4.1, we first need the following lemma.

**Lemma E.1.** *Suppose that Assumptions 3.1 to 3.4 hold. For any $\eta$ satisfying*

$$\eta \leq \frac{1}{\sqrt{2(1+l_{g,1}^2/\mu^2)(L_{x,1}^2+L_{y,1}^2)}},$$

*it holds that*

$$\left(1 - \frac{1}{2}\eta L_1 - 2L_1\|\hat{y}_t - y_t^*\|\right)\frac{1}{T}\sum_{t=0}^{T-1}\mathbb{E}\|\nabla\Phi(x_t)\|$$

$$\leq \frac{\Phi(x_0)-\Phi(x_T)}{T\eta} + \frac{2}{T}\sum_{t=0}^{T-1}\mathbb{E}\|\epsilon_t\| + \frac{2l_{g,1}l_{f,0}}{\mu}\left(1-\frac{\mu}{l_{g,1}}\right)^Q + \frac{2L_0}{T}\sum_{t=0}^{T-1}\|\hat{y}_t - y_t^*\| + \frac{1}{2}\eta L_0.$$

*Proof of Lemma E.1.* Define $h_t = m_t - \nabla\Phi(x_t)$. Then we apply Lemma B.3 to obtain

$$\begin{aligned}
\Phi(x_{t+1}) &\leq \Phi(x_t) + \langle\nabla\Phi(x_t), x_{t+1}-x_t\rangle + \frac{L_0+L_1\|\nabla\Phi(x_t)\|}{2}\|x_{t+1}-x_t\|^2 \\
&= \Phi(x_t) - \eta\langle\nabla\Phi(x_t), \frac{m_t}{\|m_t\|}\rangle + \frac{1}{2}\eta^2(L_0+L_1\|\nabla\Phi(x_t)\|) \\
&= \Phi(x_t) - \eta\langle m_t - h_t, \frac{m_t}{\|m_t\|}\rangle + \frac{1}{2}\eta^2(L_0+L_1\|\nabla\Phi(x_t)\|) \\
&= \Phi(x_t) - \eta\|m_t\| + \eta\langle h_t, \frac{m_t}{\|m_t\|}\rangle + \frac{1}{2}\eta^2(L_0+L_1\|\nabla\Phi(x_t)\|) \\
&\leq \Phi(x_t) - \eta\|m_t\| + \eta\|h_t\| + \frac{1}{2}\eta^2(L_0+L_1\|\nabla\Phi(x_t)\|) \\
&\leq \Phi(x_t) - \eta\|\nabla\Phi(x_t)\| + 2\eta\|h_t\| + \frac{1}{2}\eta^2(L_0+L_1\|\nabla\Phi(x_t)\|),
\end{aligned} \tag{53}$$

where for the last two lines we use Cauchy-Schwarz inequality and $\|h_t\| = \|\nabla\Phi(x_t)+h_t\| \geq \|\nabla\Phi(x_t)\| - \|h_t\|$. Now expanding $h_t$ by triangle inequality, we have

$$\begin{aligned}
\|h_t\| &= \|m_t - \nabla\Phi(x_t)\| \\
&\leq \|m_t - \mathbb{E}_t[\bar{\nabla}f(x_t,\hat{y}_t;\bar{\xi}_t)]\| + \|\mathbb{E}_t[\bar{\nabla}f(x_t,\hat{y}_t;\bar{\xi}_t)] - \bar{\nabla}f(x_t,\hat{y}_t)\| + \|\bar{\nabla}f(x_t,\hat{y}_t) - \nabla\Phi(x_t)\| \\
&\leq \|\epsilon_t\| + \frac{l_{g,1}l_{f,0}}{\mu}\left(1-\frac{\mu}{l_{g,1}}\right)^Q + (L_0+L_1\|\nabla\Phi(x_t)\|)\|\hat{y}_t - y_t^*\|,
\end{aligned}$$

where we use definition of $\epsilon_t$, Lemmas B.4 and B.5 in the last inequality. Plugging the above inequality back into (53) we obtain

$$\begin{aligned}
\Phi(x_{t+1}) &\leq \Phi(x_t) - \eta\|\nabla\Phi(x_t)\| + 2\eta\|\epsilon_t\| + 2\eta\frac{l_{g,1}l_{f,0}}{\mu}\left(1-\frac{\mu}{l_{g,1}}\right)^Q \\
&\quad + 2\eta(L_0+L_1\|\nabla\Phi(x_t)\|)\|\hat{y}_t - y_t^*\| + \frac{1}{2}\eta^2(L_0+L_1\|\nabla\Phi(x_t)\|) \\
&= \Phi(x_t) - \left(\eta - \frac{1}{2}\eta^2 L_1 - 2\eta L_1\|\hat{y}_t - y_t^*\|\right)\|\nabla\Phi(x_t)\| + 2\eta\|\epsilon_t\| \\
&\quad + 2\eta\frac{l_{g,1}l_{f,0}}{\mu}\left(1-\frac{\mu}{l_{g,1}}\right)^Q + 2\eta L_0\|\hat{y}_t - y_t^*\| + \frac{1}{2}\eta^2 L_0.
\end{aligned}$$

Dividing $1/T\eta$ on both sides, then taking telescope sum and total expectation, and rearranging it finally yields

$$\left(1 - \frac{1}{2}\eta L_1 - 2L_1\|\hat{y}_t - y_t^*\|\right)\frac{1}{T}\sum_{t=0}^{T-1}\mathbb{E}\|\nabla\Phi(x_t)\|$$

$$\leq \frac{\Phi(x_0) - \Phi(x_T)}{T\eta} + \frac{2}{T}\sum_{t=0}^{T-1}\mathbb{E}\|\epsilon_t\| + \frac{2l_{g,1}l_{f,0}}{\mu}\left(1 - \frac{\mu}{l_{g,1}}\right)^Q + \frac{2L_0}{T}\sum_{t=0}^{T-1}\|\hat{y}_t - y_t^*\| + \frac{1}{2}\eta L_0.$$

$\square$

**Theorem E.2** (Restatement of Theorem 4.1). *Suppose Assumptions 3.1 to 3.4 hold. Let $\{x_t\}$ be the iterates produced by Algorithm 2. For any given $\delta \in (0,1)$ and any small $\epsilon > 0$ satisfying*

$$\epsilon \leq \min\left\{\frac{L_0}{32L_1}, \frac{l_{g,1}L_0}{\mu\bar{L}_1}, \frac{L_0}{8\bar{L}_1}, \frac{L_0l_{g,1}\tilde{\sigma}_{g,1}}{\mu^2}, \frac{L_0}{\mu}\sqrt{\frac{l_{g,1}\tilde{\sigma}_{g,1}}{L_1}}, \left(\frac{164\cdot 32ed_0L_0^2\tilde{\sigma}_{g,1}^2}{\delta\mu^2}\max\left\{\frac{l_{g,1}}{\tilde{\sigma}_{g,1}}, \frac{\bar{\sigma}}{d_0}\right\}\right)^{1/3}\right\},$$

$$(54)$$

*if $\sigma_{g,1}$ satisfies*

$$\sigma_{g,1} = \left(\min\left\{\frac{\mu^2\epsilon^2}{164\cdot 16L_0^2\tilde{\sigma}_{g,1}^2\ln(P)}, \frac{l_{g,1}}{4\tilde{\sigma}_{g,1}L_1}, \frac{\epsilon^2}{4\bar{\sigma}^2}\right\}\right)^{1/4}\tilde{\sigma}_{g,1} \tag{55}$$

*with $\tilde{\sigma}_{g,1} = O(1)$, and we set parameters $\alpha, \alpha^{\mathrm{init}}, \beta, \gamma, \eta, \tau, I, N, S, Q, T_0$ as*

$$1 - \beta = \min\left\{\frac{\mu^2\epsilon^2}{164\cdot 16L_0^2\tilde{\sigma}_{g,1}^2\ln(P)}, \frac{l_{g,1}}{4\tilde{\sigma}_{g,1}L_1}, \frac{\epsilon^2}{4\bar{\sigma}^2}\right\}, \quad \eta = \min\left\{\frac{\tilde{\sigma}_{g,1}}{l_{g,1}}, \frac{d_0}{\bar{\sigma}}\right\}(1-\beta), \tag{56}$$

$$\alpha^{\mathrm{init}} = \frac{1-\beta}{\mu + l_{g,1}} \quad \alpha = \frac{1-\beta}{\mu}, \quad \gamma = \frac{1-\sqrt{\mu\alpha}}{1+\sqrt{\mu\alpha}}, \quad \tau = 1 - \sqrt{\mu\alpha}, \tag{57}$$

$$T_0 = \ln\left(\frac{\mu^3\alpha^3\epsilon^2}{256L_0^2\|y_0^{\mathrm{init}} - y_0^*\|^2}\right)\Big/\ln\left(1 - \frac{\mu\alpha}{4}\right), \tag{58}$$

$$I = \frac{\mu\epsilon}{2(1-\beta)L_0\tilde{\sigma}_{g,1}}, \quad N = \ln\left(\frac{\mu\alpha}{128}\right)\Big/\ln\left(1 - \frac{\sqrt{\mu\alpha}}{4}\right), \tag{59}$$

$$S = \max\left\{128\ln(P), \frac{128\bar{L}_0^2}{L_0^2}\ln(P), \frac{\mu^2\bar{L}_0^2}{l_{g,1}^2L_0^2}\right\}, \quad Q = \ln\left(1 - \frac{\mu}{l_{g,1}}\right)\Big/\ln\left(\frac{\mu\epsilon}{l_{g,1}l_{f,0}}\right), \tag{60}$$

*where $d_0$ and $P$ are defined in (37). Then with probability at least $1 - 2\delta$ over the randomness in $\sigma(\mathcal{F}^{\mathrm{init}} \cup \widetilde{\mathcal{F}}_T^1)$ (for Option I) or $\sigma(\mathcal{F}^{\mathrm{init}} \cup (\cup_{t\leq T}\widetilde{\mathcal{F}}_t^2))$ (for Option II), Algorithm 2 guarantees $\frac{1}{T}\sum_{t=1}^T\mathbb{E}\|\nabla\Phi(x_t)\| \leq 20\epsilon$ within $T = \frac{4d_0}{\eta\epsilon} = \widetilde{O}(1/\epsilon^3)$ iterations, where the expectation is taken over the randomness in $\mathcal{F}_T$. For Option I, it requires $T_0 + SQT = \widetilde{O}(1/\epsilon^3)$ oracle calls of stochastic gradient or Hessian/Jacobian vector product. For Option II, it requires $T_0 + \frac{NT}{I} + SQT = \widetilde{O}(1/\epsilon^3)$ oracle calls of stochastic gradient or Hessian/Jacobian vector product.*

*Proof of Theorem E.2.* By Lemmas D.6 and E.1, we have

$$\left(1 - \frac{1}{2}\eta L_1 - 2L_1\|\hat{y}_t - y_t^*\|\right)\frac{1}{T}\sum_{t=0}^{T-1}\mathbb{E}\|\nabla\Phi(x_t)\|$$

$$\leq \frac{\Phi(x_0) - \Phi(x_T)}{T\eta} + \frac{2}{T}\sum_{t=0}^{T-1}\mathbb{E}\|\epsilon_t\| + \frac{2l_{g,1}l_{f,0}}{\mu}\left(1 - \frac{\mu}{l_{g,1}}\right)^Q + \frac{2L_0}{T}\sum_{t=0}^{T-1}\|\hat{y}_t - y_t^*\| + \frac{1}{2}\eta L_0$$

$$\leq \frac{d_0}{T\eta} + \frac{2l_{g,1}l_{f,0}}{\mu}\left(1 - \frac{\mu}{l_{g,1}}\right)^Q + \frac{2L_0}{T}\sum_{t=0}^{T-1}\|\hat{y}_t - y_t^*\| + \frac{1}{2}\eta L_0$$

$$+ \frac{2\bar{\sigma}}{T(1-\beta)} + 2\sqrt{1-\beta}\bar{\sigma} + \frac{2\bar{L}_0}{\sqrt{1-\beta}}\sqrt{\frac{2(\eta^2 + \vartheta^2)}{S}} + 2\bar{L}_1\sqrt{\frac{2(\eta^2 + \vartheta^2)}{S(1-\beta)}}\frac{1}{T}\sum_{t=0}^{T-1}\mathbb{E}\|\nabla\Phi(x_t)\|.$$

Rearranging the above inequality gives

$$\underbrace{\left(1 - \frac{1}{2}\eta L_1 - 2L_1\|\hat{y}_t - y_t^*\| - 2\bar{L}_1\sqrt{\frac{2(\eta^2 + \vartheta^2)}{S(1-\beta)}}\right)\frac{1}{T}\sum_{t=0}^{T-1}\mathbb{E}\|\nabla\Phi(x_t)\|}_{\text{(LHS)}}$$

$$\leq \underbrace{\frac{d_0}{T\eta} + \frac{2l_{g,1}l_{f,0}}{\mu}\left(1 - \frac{\mu}{l_{g,1}}\right)^Q + \frac{2L_0}{T}\sum_{t=0}^{T-1}\|\hat{y}_t - y_t^*\| + \frac{1}{2}\eta L_0 + \frac{2\bar{\sigma}}{T(1-\beta)} + 2\sqrt{1-\beta}\bar{\sigma} + \frac{2\bar{L}_0}{\sqrt{1-\beta}}\sqrt{\frac{2(\eta^2 + \vartheta^2)}{S}}}_{\text{(RHS)}}.$$

**Bounding (LHS).** By Lemma D.5, we have

$$\text{(LHS)} \geq 1 - \frac{\tilde{\sigma}_{g,1}L_1}{2l_{g,1}}(1 - \beta) - 2L_1\frac{2\epsilon}{L_0} - 2\bar{L}_1\sqrt{\frac{2(\eta^2 + \vartheta^2)}{S(1-\beta)}} \tag{61}$$

$$\geq 1 - \frac{1}{8} - \frac{1}{8} - \frac{1}{4} = \frac{1}{2}$$

**Bounding (RHS).** By choice of parameters, we have

$$\text{(RHS)} \leq \frac{1}{4}\epsilon + 2\epsilon + 4\epsilon + \epsilon + \frac{1}{2}\epsilon + \epsilon + \epsilon \leq 10\epsilon. \tag{62}$$

Combining (61) and (62) finally yields

$$\frac{1}{T}\sum_{t=0}^{T-1}\mathbb{E}\|\nabla\Phi(x_t)\| \leq 20\epsilon.$$

$\square$

# F   Omitted Proofs in Appendix B

## F.1   Proof of Lemma B.5

**Lemma F.1** (Restatement of Lemma B.5). *Under Assumptions 3.1 to 3.4, we have*

$$\|\bar{\nabla}f(x,y) - \nabla\Phi(x)\| \leq (\bar{L} + L_{x,1}\|\nabla\Phi(x)\|)\|y - y^*(x)\|,$$

*where constant $\bar{L}$ is defined as*

$$\bar{L} := L_{x,0} + L_{x,1}\frac{l_{g,1}l_{f,0}}{\mu} + \frac{l_{g,1}}{\mu}(L_{y,0} + L_{y,1}l_{f,0}) + l_{f,0}\frac{\mu l_{g,2} + l_{g,1}l_{g,2}}{\mu^2} \leq L_0.$$

*Proof of Lemma B.5.* Recall that the exact expressions of $\bar{\nabla}f(x,y)$ and $\nabla\Phi(x)$ are

$$\bar{\nabla}f(x,y) = \nabla_x f(x,y) - \nabla_{xy}^2 g(x,y)[\nabla_{yy}^2 g(x,y)]^{-1}\nabla_y f(x,y)$$

and

$$\nabla\Phi(x) = \nabla_x f(x,y^*(x)) - \nabla_{xy}^2 g(x,y^*(x))[\nabla_{yy}^2 g(x,y^*(x))]^{-1}\nabla_y f(x,y^*(x)).$$

Then by Assumption 3.2 we have

$$\|\bar{\nabla}f(x,y) - \nabla\Phi(x)\| \leq \|\nabla_x f(x,y) - \nabla_x f(x,y^*(x))\|$$
$$+ \|\nabla_{xy}^2 g(x,y)[\nabla_{yy}^2 g(x,y)]^{-1}\nabla_y f(x,y) - \nabla_{xy}^2 g(x,y^*(x))[\nabla_{yy}^2 g(x,y^*(x))]^{-1}\nabla_y f(x,y^*(x))\|$$
$$\leq (L_{x,0} + L_{x,1}\|\nabla_x f(x,y^*(x))\|)\|y - y^*(x)\|$$
$$+ \|\nabla_{xy}^2 g(x,y)[\nabla_{yy}^2 g(x,y)]^{-1}\nabla_y f(x,y) - \nabla_{xy}^2 g(x,y^*(x))[\nabla_{yy}^2 g(x,y)]^{-1}\nabla_y f(x,y)\|$$
$$+ \|\nabla_{xy}^2 g(x,y^*(x))[\nabla_{yy}^2 g(x,y)]^{-1}\nabla_y f(x,y) - \nabla_{xy}^2 g(x,y^*(x))[\nabla_{yy}^2 g(x,y^*(x))]^{-1}\nabla_y f(x,y)\|$$
$$+ \|\nabla_{xy}^2 g(x,y^*(x))[\nabla_{yy}^2 g(x,y^*(x))]^{-1}\nabla_y f(x,y) - \nabla_{xy}^2 g(x,y^*(x))[\nabla_{yy}^2 g(x,y^*(x))]^{-1}\nabla_y f(x,y^*(x))\|$$

$$\leq \left( L_{x,0} + L_{x,1} \left( \frac{l_{g,1} l_{f,0}}{\mu} + \|\nabla \Phi(x)\| \right) \right) \|y - y^*(x)\|$$

$$+ \frac{l_{f,0}}{\mu} l_{g,2} \|y - y^*(x)\| + \frac{l_{f,0} l_{g,1}}{\mu^2} l_{g,2} \|y - y^*(x)\| + \frac{l_{g,1}}{\mu} (L_{y,0} + L_{y,1} \|\nabla_y f(x, y^*(x))\|) \|y - y^*(x)\|$$

$$= \left( L_{x,0} + L_{x,1} \frac{l_{g,1} l_{f,0}}{\mu} + \frac{l_{g,1}}{\mu} (L_{y,0} + L_{y,1} l_{f,0}) + l_{f,0} \frac{\mu l_{g,2} + l_{g,1} l_{g,2}}{\mu^2} + L_{x,1} \|\nabla \Phi(x)\| \right) \|y - y^*(x)\|.$$

By definition of $\bar{L}$ we conclude the proof. $\qquad\square$

## F.2 Proof of Lemma B.6

**Lemma F.2** (Restatement of Lemma B.6). *Under Assumptions 3.1 to 3.4, we have*

(i) *For any fixed $y \in \mathbb{R}^{d_y}$ and any $x_1, x_2 \in \mathbb{R}^{d_x}$,*

$$\mathbb{E}_{\bar{\xi}} \|\bar{\nabla} f(x_1, y; \bar{\xi}) - \bar{\nabla} f(x_2, y; \bar{\xi})\|^2 \leq (\bar{L}_0^2 + \bar{L}_1^2 \|\nabla \Phi(x_1)\|^2) \|x_1 - x_2\|^2.$$

(ii) *For any fixed $x \in \mathbb{R}^{d_x}$ and any $y_1, y_2 \in \mathbb{R}^{d_y}$,*

$$\mathbb{E}_{\bar{\xi}} \|\bar{\nabla} f(x, y_1; \bar{\xi}) - \bar{\nabla} f(x, y_2; \bar{\xi})\|^2 \leq (\bar{L}_0^2 + \bar{L}_1^2 \|\nabla \Phi(x_1)\|^2) \|x_1 - x_2\|^2.$$

*In the above expressions, we define $\bar{L}_0$ and $\bar{L}_1$ as*

$$\bar{L}_0 = \left\{ 4 \left( L_{x,0} + L_{x,1} \left( \frac{l_{g,1} l_{f,0}}{\mu} + \left( L_{x,0} + \frac{L_{x,1} l_{g,1} l_{f,0}}{\mu} \right) \|y_1 - y_1^*\| \right) \right)^2 \right.$$

$$\left. + \frac{6Q}{2\mu l_{g,1} - \mu^2} \left( l_{g,1}^2 (L_{y,0} + L_{y,1} l_{f,0})^2 + l_{f,0}^2 l_{g,2}^2 + \frac{l_{f,0}^2 l_{g,1}^2 l_{g,2}^2 K^2}{(l_{g,1} - \mu)^2} \right) \right\}^{1/2},$$

$$\bar{L}_1 = 2L_{x,1}(1 + L_{x,1} \|y_1 - y_1^*\|).$$

*Proof of Lemma B.6.* We show statement $(i)$ of the lemma, and $(ii)$ follows by similar arguments. For any fixed $y \in \mathbb{R}^{d_y}$ and any $x_1, x_2 \in \mathbb{R}^{d_x}$, by definition of $\bar{\nabla} f(x, y; \bar{\xi})$ we have

$$\|\bar{\nabla} f(x_1, y; \bar{\xi}) - \bar{\nabla} f(x_2, y; \bar{\xi})\|^2$$

$$\leq 2\|\nabla_x F(x_1, y; \xi) - \nabla_x F(x_2, y; \xi)\|^2 + 2 \left\| \nabla_{xy}^2 G(x_1, y; \zeta^{(0)}) \left[ \frac{Q}{l_{g,1}} \prod_{i=1}^{\mathsf{q}} \left( I - \frac{1}{l_{g,1}} \nabla_{yy}^2 G(x_1, y; \zeta^{(i)}) \right) \right] \nabla_y F(x_1, y; \xi) \right.$$

$$\left. - \nabla_{xy}^2 G(x_2, y; \zeta^{(0)}) \left[ \frac{Q}{l_{g,1}} \prod_{i=1}^{\mathsf{q}} \left( I - \frac{1}{l_{g,1}} \nabla_{yy}^2 G(x_2, y; \zeta^{(i)}) \right) \right] \nabla_y F(x_2, y; \xi) \right\|^2$$

$$\leq 2(L_{x,0} + L_{x,1} \|\nabla_x f(x_1, y)\|)^2 \|x_1 - x_2\|^2 + 2 \left\| \nabla_{xy}^2 G(x_1, y; \zeta^{(0)}) \left[ \frac{Q}{l_{g,1}} \prod_{i=1}^{\mathsf{q}} \left( I - \frac{1}{l_{g,1}} \nabla_{yy}^2 G(x_1, y; \zeta^{(i)}) \right) \right] \nabla_y F(x_1, y; \xi) \right.$$

$$\left. - \nabla_{xy}^2 G(x_2, y; \zeta^{(0)}) \left[ \frac{Q}{l_{g,1}} \prod_{i=1}^{\mathsf{q}} \left( I - \frac{1}{l_{g,1}} \nabla_{yy}^2 G(x_2, y; \zeta^{(i)}) \right) \right] \nabla_y F(x_2, y; \xi) \right\|^2.$$

For the second term above, we have

$$\left\| \nabla_{xy}^2 G(x_1, y; \zeta^{(0)}) \left[ \frac{Q}{l_{g,1}} \prod_{i=1}^{\mathsf{q}} \left( I - \frac{1}{l_{g,1}} \nabla_{yy}^2 G(x_1, y; \zeta^{(i)}) \right) \right] \nabla_y F(x_1, y; \xi) \right.$$

$$\left. - \nabla_{xy}^2 G(x_2, y; \zeta^{(0)}) \left[ \frac{Q}{l_{g,1}} \prod_{i=1}^{\mathsf{q}} \left( I - \frac{1}{l_{g,1}} \nabla_{yy}^2 G(x_2, y; \zeta^{(i)}) \right) \right] \nabla_y F(x_2, y; \xi) \right\|^2$$

$$\leq 3l_{g,1}^2 \frac{Q^2}{l_{g,1}^2} \left( 1 - \frac{\mu}{l_{g,1}} \right)^{2\mathsf{q}} \|\nabla_y F(x_1, y; \xi) - \nabla_y F(x_2, y; \xi)\|^2$$

$$+ 3l_{f,0}^2 \frac{Q^2}{l_{g,1}^2} \left( 1 - \frac{\mu}{l_{g,1}} \right)^{2\mathsf{q}} \|\nabla_{xy}^2 G(x_1, y; \zeta^{(0)}) - \nabla_{xy}^2 G(x_2, y; \zeta^{(0)})\|^2$$

$$+ 3l_{g,1}^2 l_{f,0}^2 \left\| \frac{Q}{l_{g,1}} \prod_{i=1}^{\mathsf{q}} \left( I - \frac{1}{l_{g,1}} \nabla_{yy}^2 G(x_1, y; \zeta^{(i)}) \right) - \frac{Q}{l_{g,1}} \prod_{i=1}^{\mathsf{q}} \left( I - \frac{1}{l_{g,1}} \nabla_{yy}^2 G(x_2, y; \zeta^{(i)}) \right) \right\|^2$$

$$\leq 3Q^2 \left( 1 - \frac{\mu}{l_{g,1}} \right)^{2\mathsf{q}} (L_{y,0} + L_{y,1} \| \nabla_y f(x_1, y) \|)^2 \| x_1 - x_2 \|^2 + 3 \frac{l_{f,0}^2 Q^2}{l_{g,1}^2} \left( 1 - \frac{\mu}{l_{g,1}} \right)^{2\mathsf{q}} l_{g,2}^2 \| x_1 - x_2 \|^2$$

$$+ 3l_{f,0}^2 Q^2 \left\| \prod_{i=1}^{\mathsf{q}} \left( I - \frac{1}{l_{g,1}} \nabla_{yy}^2 G(x_1, y; \zeta^{(i)}) \right) - \prod_{i=1}^{\mathsf{q}} \left( I - \frac{1}{l_{g,1}} \nabla_{yy}^2 G(x_2, y; \zeta^{(i)}) \right) \right\|^2.$$

Then we take expectation with respect to $\mathsf{q}$ and obtain

$$\mathbb{E}_{\mathsf{q}} \left\| \nabla_{xy}^2 G(x_1, y; \zeta^{(0)}) \left[ \frac{Q}{l_{g,1}} \prod_{i=1}^{\mathsf{q}} \left( I - \frac{1}{l_{g,1}} \nabla_{yy}^2 G(x_1, y; \zeta^{(i)}) \right) \right] \nabla_y F(x_1, y; \xi) \right.$$

$$\left. - \nabla_{xy}^2 G(x_2, y; \zeta^{(0)}) \left[ \frac{Q}{l_{g,1}} \prod_{i=1}^{\mathsf{q}} \left( I - \frac{1}{l_{g,1}} \nabla_{yy}^2 G(x_2, y; \zeta^{(i)}) \right) \right] \nabla_y F(x_2, y; \xi) \right\|^2$$

$$\leq \left( 3Q^2 (L_{y,0} + L_{y,1} \| \nabla_y f(x_1, y) \|)^2 \| x_1 - x_2 \|^2 + 3 \frac{l_{f,0}^2 Q^2}{l_{g,1}^2} l_{g,2}^2 \| x_1 - x_2 \|^2 \right) \mathbb{E}_{\mathsf{q}} \left[ \left( 1 - \frac{\mu}{l_{g,1}} \right)^{2\mathsf{q}} \right]$$

$$+ 3l_{f,0}^2 Q^2 \mathbb{E}_{\mathsf{q}} \left\| \prod_{i=1}^{\mathsf{q}} \left( I - \frac{1}{l_{g,1}} \nabla_{yy}^2 G(x_1, y; \zeta^{(i)}) \right) - \prod_{i=1}^{\mathsf{q}} \left( I - \frac{1}{l_{g,1}} \nabla_{yy}^2 G(x_2, y; \zeta^{(i)}) \right) \right\|^2$$

$$\leq \left( 3Q^2 (L_{y,0} + L_{y,1} \| \nabla_y f(x_1, y) \|)^2 \| x_1 - x_2 \|^2 + 3 \frac{l_{f,0}^2 Q^2}{l_{g,1}^2} l_{g,2}^2 \| x_1 - x_2 \|^2 \right) \cdot \frac{l_{g,1}^2}{Q(2\mu l_{g,1} - \mu^2)}$$

$$+ 3l_{f,0}^2 Q^2 \cdot \frac{l_{g,1}^2 l_{g,2}^2 Q}{(l_{g,1} - \mu)^2 (2\mu l_{g,1} - \mu^2)} \| x_1 - x_2 \|^2$$

$$\leq \frac{3Q}{2\mu l_{g,1} - \mu^2} \left( l_{g,1}^2 (L_{y,0} + L_{y,1} l_{f,0})^2 \| x_1 - x_2 \|^2 + l_{f,0}^2 l_{g,2}^2 \| x_1 - x_2 \|^2 \right) + \frac{3l_{f,0}^2 l_{g,1}^2 l_{g,2}^2 Q^3}{(l_{g,1} - \mu)^2 (2\mu l_{g,1} - \mu^2)} \| x_1 - x_2 \|^2$$

$$= \frac{3Q}{2\mu l_{g,1} - \mu^2} \left( l_{g,1}^2 (L_{y,0} + L_{y,1} l_{f,0})^2 + l_{f,0}^2 l_{g,2}^2 + \frac{l_{f,0}^2 l_{g,1}^2 l_{g,2}^2 Q^2}{(l_{g,1} - \mu)^2} \right) \| x_1 - x_2 \|^2$$

Finally, taking expectation on both sides yields

$$\mathbb{E}_{\bar{\xi}} \| \bar{\nabla} f(x_1, y; \bar{\xi}) - \bar{\nabla} f(x_2, y; \bar{\xi}) \|^2 \leq 2(L_{x,0} + L_{x,1} \| \nabla_x f(x_1, y) \|)^2 \| x_1 - x_2 \|^2$$

$$+ \frac{6Q}{2\mu l_{g,1} - \mu^2} \left( l_{g,1}^2 (L_{y,0} + L_{y,1} l_{f,0})^2 + l_{f,0}^2 l_{g,2}^2 + \frac{l_{f,0}^2 l_{g,1}^2 l_{g,2}^2 Q^2}{(l_{g,1} - \mu)^2} \right) \| x_1 - x_2 \|^2.$$

Since for any $y \in \mathbb{R}^{d_y}$, we have

$$\| \nabla_x f(x_1, y) - \nabla_x f(x_1, y_1^*) \| \leq (L_{x,0} + L_{x,1} \| \nabla_x f(x_1, y_1^*) \|) \| y_1 - y_1^* \|$$

$$\leq \left( L_{x,0} + L_{x,1} \left( \frac{l_{g,1} l_{f,0}}{\mu} + \| \nabla \Phi(x_1) \| \right) \right) \| y_1 - y_1^* \|$$

$$= \left( L_{x,0} + \frac{L_{x,1} l_{g,1} l_{f,0}}{\mu} + L_{x,1} \| \nabla \Phi(x_1) \| \right) \| y_1 - y_1^* \|,$$

which yields

$$\| \nabla_x f(x_1, y) \| \leq \| \nabla_x f(x_1, y_1^*) \| + \left( L_{x,0} + \frac{L_{x,1} l_{g,1} l_{f,0}}{\mu} + L_{x,1} \| \nabla \Phi(x_1) \| \right) \| y_1 - y_1^* \|$$

$$\leq \frac{l_{g,1} l_{f,0}}{\mu} + \| \nabla \Phi(x_1) \| + \left( L_{x,0} + \frac{L_{x,1} l_{g,1} l_{f,0}}{\mu} + L_{x,1} \| \nabla \Phi(x_1) \| \right) \| y_1 - y_1^* \|$$

$$= \left( \frac{l_{g,1} l_{f,0}}{\mu} + \left( L_{x,0} + \frac{L_{x,1} l_{g,1} l_{f,0}}{\mu} \right) \| y_1 - y_1^* \| \right) + (1 + L_{x,1} \| y_1 - y_1^* \|) \| \nabla \Phi(x_1) \|.$$

Therefore, we conclude that

$$
\mathbb{E}_{\bar{\xi}} \| \bar{\nabla} f(x_1, y; \bar{\xi}) - \bar{\nabla} f(x_2, y; \bar{\xi}) \|^2 \leq 2(L_{x,0} + L_{x,1} \| \nabla_x f(x_1, y) \|)^2 \| x_1 - x_2 \|^2
$$

$$
+ \frac{6Q}{2\mu l_{g,1} - \mu^2} \left( l_{g,1}^2 (L_{y,0} + L_{y,1} l_{f,0})^2 + l_{f,0}^2 l_{g,2}^2 + \frac{l_{f,0}^2 l_{g,1}^2 l_{g,2}^2 Q^2}{(l_{g,1} - \mu)^2} \right) \| x_1 - x_2 \|^2
$$

$$
\leq 2 \left( L_{x,0} + L_{x,1} \left( \frac{l_{g,1} l_{f,0}}{\mu} + \left( L_{x,0} + \frac{L_{x,1} l_{g,1} l_{f,0}}{\mu} \right) \| y_1 - y_1^* \| \right) + L_{x,1} (1 + L_{x,1} \| y_1 - y_1^* \|) \| \nabla \Phi(x_1) \| \right)^2 \| x_1 - x_2 \|^2
$$

$$
+ \frac{6Q}{2\mu l_{g,1} - \mu^2} \left( l_{g,1}^2 (L_{y,0} + L_{y,1} l_{f,0})^2 + l_{f,0}^2 l_{g,2}^2 + \frac{l_{f,0}^2 l_{g,1}^2 l_{g,2}^2 Q^2}{(l_{g,1} - \mu)^2} \right) \| x_1 - x_2 \|^2
$$

$$
\leq 4 \left( L_{x,0} + L_{x,1} \left( \frac{l_{g,1} l_{f,0}}{\mu} + \left( L_{x,0} + \frac{L_{x,1} l_{g,1} l_{f,0}}{\mu} \right) \| y_1 - y_1^* \| \right) \right)^2 \| x_1 - x_2 \|^2
$$

$$
+ 4 L_{x,1}^2 (1 + L_{x,1} \| y_1 - y_1^* \|)^2 \| \nabla \Phi(x_1) \|^2 \| x_1 - x_2 \|^2
$$

$$
+ \frac{6Q}{2\mu l_{g,1} - \mu^2} \left( l_{g,1}^2 (L_{y,0} + L_{y,1} l_{f,0})^2 + l_{f,0}^2 l_{g,2}^2 + \frac{l_{f,0}^2 l_{g,1}^2 l_{g,2}^2 Q^2}{(l_{g,1} - \mu)^2} \right) \| x_1 - x_2 \|^2
$$

$$
= (\bar{L}_0^2 + \bar{L}_1^2 \| \nabla \Phi(x_1) \|^2) \| x_1 - x_2 \|^2,
$$

where we use the definition of $\bar{L}_0$ and $\bar{L}_1$ in the last equality. $\qquad\square$

## G   Additional Experimental Details

**Hyerparameter setting.**   We tune the best hyperparameters for each algorithm, including upper-/lower-level step size, the number of inner loops, momentum parameters, etc. The upper-level learning rate $\eta_{up}$ and lower-level learing rate $\eta_{low}$ are tuned in the range of $[0.001, 0.1]$ for all the baselines on experiments of AUC maximization and data hyper-cleaning, the best $(\eta_{up}, \eta_{low})$ on **AUC maximization** are summarized as follows: StocBio: $(0.01, 0.001)$, TTSA: $(0.005, 0.01)$, SABA: $(0.01, 0.005)$, MA-SOBA: $(0.01, 0.005)$, SUSTAIN: $(0.03, 0.01)$, VRBO: $(0.05, 0.01)$, BO-REP: $(0.001, 0.001)$, AccBO: $(0.005, 0.005)$. The best learning rate on the experiment of **data hyper-cleaning** are summarized as follows: Stocbio: $(0.01, 0.002)$, TTSA: $(0.001, 0.01)$, SABA: $(0.05, 0.02)$, MA-SOBA: $(0.01, 0.01)$, SUSTAIN: $(0.05, 0.05)$, VRBO: $(0.1, 0.05)$, BO-REP: $(0.02, 0.01)$, AccBO: $(0.1, 0.1)$. Note that SUSTAIN decays its upper-/lower-level step size with epoch ($t$) by $\eta_{up} = \eta_{up}/(t+2)^{1/3}, \eta_{low} = \eta_{up}/(t+2)^{1/3}$, while other algorithms use a constant learning rate. The number for neumann series estimation in StocBiO and VRBO is fixed to 3, while it is uniformly sampled from $\{1, 2, 3\}$ in TTSA, SUSTAIN, and AccBO. In AUC maximization, AccBO uses Option I (Option II in data hyper-cleaning) to update the lower-level variable, and sets the Nestrov momentum parameter $\gamma = 0.5$, the averaging parameter $\tau = 0.5$ ($\gamma = 0.1$ and $\tau = 0.5$ in data hyper-cleaning). In AUC maximization, the batch size is set to be 32 for all algorithms except VRBO, which uses larger batch size of 64 (tuned in the range of $\{32, 64, 128, 256, 512\}$) at the checkpoint step and 32 otherwise. In data hyper-cleaning, the batch size is set to be 128 for all algorithms except VRBO, which uses larger batch size of 256 (tuned in the range of $\{63, 128, 256, 512, 1024\}$) at the checkpoint step and 128 otherwise. AccBO uses Option II in data hyper-cleaning, and the periodical update for low-level variable sets the iterations $N = 3$ and update interval $I = 2$. Other hyperparameters setting keep the same in AUC maximization and data hyper-cleaning: The momentum parameter $\beta$ is fixed to 0.9 in AccBO, MA-SOBA, BO-REP. The warm start steps for lower-level variable in AccBO is set to 3. The number of inner loops for StocBio is set to 3. BO-REP uses the periodical update for low-level variable, and set the iterations $N = 3$ and the update interval $I = 2$.

