# OpenReview forum: "An Accelerated Algorithm for Stochastic Bilevel Optimization under Unbounded Smoothness"
_NeurIPS.cc/2024/Conference — NeurIPS 2024 poster_

### Official Review · Reviewer_2WtS · 2024-06-28

**Soundness:** 3
**Presentation:** 3
**Contribution:** 2
**Rating:** 6
**Confidence:** 3

**Summary:**

This work develops the first algorithm that achieves the near-optimal oracle complexity $\widetilde{O}(\epsilon^{-3})$ (the number of evaluations to gradient or Hessian/Jacobian vector product) to achieve an $\epsilon$-stationary point of a stochastic bilevel optimization problem with unbounded smooth nonconvex upper-level function and strongly-convex lower-level function. The algorithm design is novel which updates the upper-level variable by normalized stochastic gradient descent with recursive momentum and updates the lower-level variable by the stochastic Nesterov accelerated gradient (SNAG) method.

**Strengths:**

This work studies a specific stochastic bilevel optimization problem with significant applications in sequential data learning. The upper-level function has unbounded smoothness which is more general than the widely used Lipscthiz smoothness assumption. The oracle complexity is improved on this specific bilevel optimization problem compared with existing works. The algorithm design not only combines but also adjusts STORM variance reduction and Nesterov acceleration. The 2 real-world experiments look convincing. The presentation is clean and clear.

**Weaknesses:**

The major weakness is Assumption 4.2 as mentioned in the question (1) below. Some points could be clarified as mentioned in other questions below.

**Questions:**

(1) **My major concern** is that Assumption 4.2 is made on algorithm-generated variables not on problem setting, which cannot be verified. Is it possible to bound $\Delta_t$ in terms of $\|x_{t+1}-x_t\|$ to get rid of Assumption 4.2? If not, at least Assumption 4.2 should also be mentioned in the main result (Theorem 4.1). **I would like to raise my rating if all the assumptions become problem-related.**

(2) In line 48 "each realization of the stochastic oracle calls is Lipschitz with respect to its argument", do you mean $F(x,y,\xi)$ and $G(x,y,\xi)$ are Lipschitz continuous functions of $(x,y)$ for each $\xi$? You might express in formula to make it more clear.

(3) You could also cite [1] as a paper on relaxed smoothness:
[1] Convex and Non-convex Optimization Under Generalized Smoothness

(4) In line 127, do you mean "... is a good approximation of $\nabla \Phi(x)$ **if $y\approx y^*(x)$**"?

(5) Is there an intuition why line 21 (averaging of y) is needed in Algorithm 2? Is it possible to implement $N=\mathcal{O}(1)$ steps of SNAG such that $\mathbb{E}\|y_{t+1}-y^*(x_t)\|\le \frac{1}{2}\mathbb{E}\|y_t-y^*(x_t)\|$ and remove averaging of y for every outer iteration t, and obtain convergence result purely in expectation instead of high probability? This yields the same oracle complexity per round since $I,N=\widetilde{\mathcal{O}}(\epsilon^{-1})$ are of the same order, and may get rid of Assumption 4.2 by bounding $\Delta_t$ in terms of $\|x_{t+1}-x_t\|$.

(6) You said your upper-level algorithm design is inspired from [17,49]. However, in line 23 (STORM variance reduction) of Algorithm 2, your second and third additive terms use the same samples $\overline{\xi}_t$ while [49] uses different samples. The algorithm design differs from [17] more. Such difference still exists even if Algorithm 2 is reduced to single-level. Why do you use such different design?

(7) Also in line 23, does $\overline{\nabla}f(x_t,\hat{y}_t;\overline{\xi}_t)$ denote  the average of the expression right below Eq. (2) over samples $s=1,\ldots,S$? You could define more clearly.

(8) At the beginning of Section 5 experiments, $ph(w; x)I_{[y=-1]}$ appears in your AIC maximization. Do you mean $rh(w; x)I_{[c=-1]}$? If not, what do $p$ and $y$ mean?

**Limitations:**

The conclusion section mentions the limitation that the convergence analysis for the Option I of Algorithm 2 relies on the lower-level problem being a one-dimensional quadratic function. Actually I think this is fine since Option II works for more general strongly convex lower-level function. The major limitation from my perspective in mentioned in my above question (1) about the undesired Assumption 4.2.
The authors said in the checklist that they do not see any negative societal impacts in their theoretical work, which I agree.

---

> ### Author Rebuttal · Authors · 2024-08-06
>
> **Thank you for taking the time to review our paper. We have addressed each of your concerns below.**
>
> **A1.** Assumption 4.2 used in Section 4.3.1 is **indeed problem-related and can be verified** when applied to the bilevel optimization setting. Note that the main goal of Section 4.3.1 is to provide a **general framework** for proving the convergence of SNAG under distributional drift. This framework can be leveraged as a tool to control the lower-level error in bilevel optimization and derive Lemma 4.7. In other words, we can view the lower-level problem of bilevel optimization as a **special case** within the more general framework provided in Section 4.3.1. We have mentioned the connection between Section 4.3.1 and Section 4.3.2 at lines 232-239 and lines 270-273.
>
> Now we verify that Assumption 4.2 holds in the bilevel optimization setting. We can regard the change of the upper-level variable $x_t$ at each iteration as the distributional drift for the lower-level problem. Thus, using the notation in Section 4.3.1, we can set the objective sequences as $\phi_t(y) = g(x_t,y)$, the minimizers at time $t$ and $t+1$ as $w_t^* = y^*(x_t)$ and $w_{t+1}^* = y^*(x_{t+1})$, the minimizer drift at time $t$ as $\Delta_t = \\|y^*(x_t) - y^*(x_{t+1})\\|$, and the stochastic gradient as $g_t = \nabla_y G(x_t,z_t;\pi_t)$. Moreover,
> - By Assumption 3.2, $g(x_t,y)$ is $\mu$-strongly convex and $l_{g,1}$-smooth in $y$.
> - By Assumption 3.3, the noise $\nabla_y G(x_t,y_t;\pi_t) - \nabla_y g(x_t,y_t)$ is norm sub-Gaussian conditioned on $\widetilde{\mathcal{F}}\_t\^1$ with parameter $\sigma_{g,1}/2$.
> - By Lemma B.1 at Appendix B and line 24 of Algorithm 2 (i.e., update for $x_t$), $y^*(x)$ is $\frac{l_{g,1}}{\mu}$-Lipschitz continuous, and $x_{t+1} = x_t - \eta\frac{m_t}{\\|m_t\\|}$. Then for all $t$, we have $\\|y^*(x_{t+1}) - y^*(x_t)\\| \leq \frac{l_{g,1}}{\mu}\\|x_{t+1}-x_t\\| = \frac{\eta l_{g,1}}{\mu}$. This implies that the minimizer drift $\Delta_t = \\|y^*(x_{t+1}) - y^*(x_t)\\|$ is almost surely bounded by $\frac{\eta l_{g,1}}{\mu}$. Hence, $\Delta_t^2 = \\|y^*(x_{t+1}) - y^*(x_t)\\|^2$ is also sub-exponential conditioned on $\widetilde{\mathcal{F}}\_t\^1$ with parameter $\Delta^2 = (\eta l_{g,1}/\mu)^2$.
>
> Thus the lower-level problem of the bilevel setting satisfies Assumption 4.2 with $\mu = \mu$, $L = l_{g,1}$, $\mathcal{H}\_t = \widetilde{\mathcal{F}}\_t\^1$, $\Delta = \eta l_{g,1}/\mu$ and $\sigma = \sigma_{g,1}/2$. Then we can apply the general lemma (i.e., Lemma 4.3) provided in Section 4.3.1 to establish the lower-level error control as stated in Section 4.3.2 (i.e., Lemma 4.4-4.7). We will make it more clear in the revised version of this paper.
>
> **A2.** Yes, your understanding is correct. It is formally stated in Assumption 3.4 that $F(x,y;\xi)$ and $G(x,y;\zeta)$ are Lipschitz continuous functions of $(x,y)$ for each $\xi$ and $\zeta$. We will mention it at line 48 in the revised version.
>
> **A3.** Thank you for your suggestion. We will mention [(Li et al., 2023)](https://arxiv.org/pdf/2306.01264) in the related work section.
>
> **A4.** Yes, we will fix it in the revised version.
>
> **A5.** Thank you for your insightful question. First, the main reason for the averaging step is that we need to ensure the condition $\\|\hat{y}\_{t+1} - \hat{y}\_t\\| \leq \vartheta = O(\epsilon^2)$ for all $t$. We achieve this in Lemma 4.7 by choosing a small $\tau$ to make $\\{\hat{y}\_t\\}$ change slowly. This condition is crucial for controlling the average hypergradient estimation error in Lemma 4.8, which depends on $\sqrt{1-\beta} + \sqrt{\frac{\eta^2+\vartheta^2}{1-\beta}}$. Thus we need at least $1-\beta = O(\epsilon^2)$, $\eta=O(\epsilon^2)$, and $\vartheta = O(\epsilon^2)$ to guarantee $\frac{1}{T}\sum_{t=0}^{T-1}\mathbb{E}\\|\epsilon_t\\| \leq O(\epsilon)$.
>
> Second, we indeed need high probability analysis for the lower-level variable. When the upper-level problem is unbounded smooth, the hypergradient bias depends on both the lower-level approximation error and the hypergradient itself, $\\|\hat{y}\_t-y_t^*\\|\\|\nabla\Phi(x_t)\\|$, see lines 803-804 of Lemma E.1 for details. These two elements are statistically dependent, necessitating high probability analysis for the lower-level problem. Moreover, Appendix D.3 shows that when $\\|y_t-y_t^*\\| = O(\epsilon)$, we need $N=\widetilde{O}(1/\epsilon)$ steps to guarantee $\\|y_{t+1} - y_t^*\\| \leq \frac{1}{2}\\|y_t-y_t^*\\|$ with high probability (lines 749 and 755). Please refer to the proof of Lemma 4.6 (lines 748-757) in Appendix D.3 for details.
>
> Third, as mentioned in **A1**, we can show that our bilevel problem setting satisfies Assumption 4.2. This allows us to apply Lemma 4.3 in Section 4.3.1 to control the lower-level approximation error with high probability.
>
> **A6.** Please note that in Corollary 3.2 and 3.4 of [(Liu et al., 2023)](https://arxiv.org/pdf/2302.06032), they set $k=K$ in Algorithm 1. This means their second and third additive terms use the same samples. Our algorithm differs from that in [(Cutkosky and Orabona, 2019)](https://arxiv.org/pdf/1905.10018) mainly in the update for $x_t$, where we use normalized SGD and they use SGD (both with recursive momentum). This difference is due to our assumption that the upper-level problem is relaxed smooth, necessitating the use of normalization to reduce the effect of noise. In contrast, [Cutkosky and Orabona (2019)](https://arxiv.org/pdf/1905.10018) assumes the single-level objective is $L$-smooth, thus SGD (with recursive momentum) is sufficient for their algorithm design.
>
> **A7.** Thank you for catching this issue. In the revised version, we will use $\widetilde{\nabla}f(x_t,\hat{y}_t;\bar{\xi}_t)$ to denote the average of the stochastic hypergradient estimators more clearly.
>
> **A8.** Thank you for noticing this, and we apologize for the typo. It should be $rh(w;x)\mathbb{I}\_{[c=-1]}$ instead of $ph(w;x)\mathbb{I}\_{[y=-1]}$. We will fix this typo in the revised version.

---

> > ### Comment · Reviewer_2WtS · 2024-08-07
> > **Reviewer 2WtS agrees with the authors and will keep rating 5.**
> >
> > Reviewer 2WtS agrees with the authors and will keep rating 5.

---

> ### Author Response · Authors · 2024-08-07
> **Thank you for your feedback!**
>
> Dear Reviewer 2WtS,
>
> We are glad that our responses addressed your major concerns. Thank you for reviewing our paper. You mentioned earlier that "I would like to raise my rating if all the assumptions become problem-related." We have now verified that Assumption 4.2 holds in the bilevel optimization setting (under Assumptions 3.1-3.4) and it is indeed problem-related. Could you please consider raising your score?
>
> Please let us know if you have any further questions or concerns.
>
> Best regards,
>
> Authors

---

> > ### Comment · Reviewer_2WtS · 2024-08-07
> > **Just raised rating**
> >
> > Dear Authors,
> >
> > Reviewer 2WtS just raised rating from 5 to 6, and soundness from 2 to 3.
> > Best regards,
> >
> > Reviewer 2WtS

---

> > > ### Author Response · Authors · 2024-08-07
> > > **Thank you!**
> > >
> > > Dear Reviewer 2WtS,
> > >
> > > Thank you for raising the rating.
> > >
> > > Best regards,
> > >
> > > Authors

---

### Official Review · Reviewer_o6Pr · 2024-07-11

**Soundness:** 3
**Presentation:** 3
**Contribution:** 3
**Rating:** 6
**Confidence:** 3

**Summary:**

The authors consider bilevel optimization problems under the unbounded smoothness assumption of the outer function. They propose AccBO, an AID-based method that uses Stochastic Nesterov Accelerated Gradient to approximate the lower-level solution and Neumann approximations to estimate the inverse Hessian-Vector product involved in the hypergradient expression. Then, a normalized STORM-like update is performed for the outer variable. They show a $\tilde{\mathcal{O}}(\epsilon^{-3})$ complexity of their algorithm. Finally, numerical experiments are provided on the deep AUC maximization and data hypercleaning problems.

**Strengths:**

- **S1**: The paper is well-written
- **S2**: The algorithms proposed by the authors achieve SOTA complexity for stochastic bilevel optimization under unbounded smoothness assumption.

**Weaknesses:**

### Major

- **W1**: Some errors in the code may invalidate the experimental results. See the comment **C2** in the Code section for details.

- **W2**: AccBO comes with many hyperparameters, which can be a concern for practical applications.

- **W3**: l.219: "is nearly optimal in terms of the dependency of $\epsilon$". The authors should be careful when talking about optimality of an algorithm. In particular, the algorithm AccBO does not belong to the algorithm class considered in [1] which does not take into account the biasedness of the hypergradient estimate due to the approximation of the inverse Hessian-vector product and the solution of the inner optimization problem. To my knowledge, the only works considering lower bounds for bilevel problems are [2, 5].

- **W4**: The works [2, 3, 4] also use Nesterov acceleration for the inner optimization problem, but in the deterministic setting. They should be mentioned in the paper.

### Code
Even though it is a theoretical paper, I have several comments regarding the experiments code:

- **C1**: In the checklist, the authors indicate "The code and data are attached as a supplement with instructions for reproducibility." However, the code documentation is insufficient to run the experiments. The data are not present in the folders and there is no instruction indicating where to download them. As a consequence, when I launch the commmand `python main.py --methods accbo ` as indicated in the `README.md`, I get an error.

- **C2**: It seems that the implementation of the different methods do not correspond to the one described in their respective original paper, which might invalidate the empirical results:
    - For `TTSA`, `SUSTAIN` and `AccBO`, the estimation of the inverse Hessian-vector product is computed by the formula
    $$
        v = \eta\sum_{q=-1}^Q\prod_{j=Q-q}^Q(I-\eta \nabla^2_{yy}G(x,y;\zeta^{(j)}))\nabla_y F(x,y;\xi)
    $$
    while in the original paper (and even the present paper for AccBO), the authors use the formula
    $$
        v = \eta\prod_{j=1}^{q(Q)}(I-\eta \nabla^2_{yy}G(x,y;\zeta^{(j)}))\nabla_y F(x,y;\xi)
    $$
    with $q(Q)\sim\mathcal{U}{\{1,\dots,Q-1\}}$.

    - For `SABA`, the SAGA-like variance reduction is absent in the code while being the core of the algorithm. Moreover, `SABA` uses constant step sizes.

    - For `VRBO`, the oracles in the case `step % self.args.update_interval == 0` should be computed with a larger batch-size (or even with full batches) than in the inner for-loop.

- **C3**: The code of `VRBO` for the datacleaning task is missing. The method to compute the hypergradient in `ma_soba.py` is missing as well for the datacleaning task (but not AUC experiment, thus I assume it is almost the same).

### Minor
- l.41: "transforms" -> "transformers"
- In `README.md` (both): `python main.py --medthods [algorithm] ` -> `python main.py --methods [algorithm] `

I am inclined to raise my score if my concerns, particularly regarding the experiments, are addressed.

**Questions:**

N/A

---

> ### Author Rebuttal · Authors · 2024-08-06
>
> **Thank you for taking the time to review our paper. We have addressed each of your concerns below.**
>
> **W1. Code issue.**
>
> **A1.** Please see **A5** to **A7** for details.
>
> **W2. AccBO comes with many hyperparameters.**
>
> **A2.** Please note that existing literature on bilevel optimization with variance reduction [(Khanduri et al., 2021](https://arxiv.org/pdf/2102.07367); [Yang et al., 2021](https://arxiv.org/pdf/2106.04692); [Guo et al. 2021)](https://arxiv.org/pdf/2105.02266) all involve many hyperparameters. Compared to previous works, AccBO has more hyperparameters since our bilevel problem setting is more challenging: our upper-level problem is unbounded smooth, while others consider $L$-smooth upper-level functions. Hence their algorithm designs cannot be applied to our setting.
>
> To achieve acceleration in our problem setting, we update the lower-level variable $y_t$ using the SNAG method with averaging, resulting in additional hyperparameters $\gamma$ and $\tau$ (line 8 and 21 of Algorithm 2), as well as $I$ and $N$ (only for Option II). In practice, $\gamma$ and $\tau$ need tuning, while $I$ and $N$ are set to default values as in [(Hao et al., 2024)](https://arxiv.org/pdf/2401.09587.pdf).
>
> **W3. Optimality and lower bound issue.**
>
> **A3.** Thank you for your insightful question. First, please note that existing literature on bilevel optimization with variance reduction [(Khanduri et al., 2021](https://arxiv.org/pdf/2102.07367); [Yang et al., 2021)](https://arxiv.org/pdf/2106.04692) state that their $\widetilde{O}(1/\epsilon^3)$ complexity results are near-optimal in terms of the dependency on $\epsilon$; see Section 1.1 of [(Yang et al., 2021)](https://arxiv.org/pdf/2106.04692) and Section 1 of [(Khanduri et al., 2021)](https://arxiv.org/pdf/2102.07367) for details.
>
> Regarding existing works considering lower bounds for bilevel problems, please note that the lower bounds in [(Ji & Liang, 2023)](https://arxiv.org/pdf/2102.03926.pdf) apply only to deterministic and convex/strongly-convex upper-level problems, and the lower bounds in [(Dagréou et al., 2023)](https://arxiv.org/pdf/2302.08766.pdf) apply only to nonconvex smooth finite-sum objectives in the stochastic setting. In contrast, we study the general expectation form in the stochastic setting, where the upper-level problem is nonconvex and unbounded smooth. Therefore, their lower bounds cannot be applied to our problem setting.
>
> Now we will establish a $\Omega(1/\epsilon^3)$ lower bound via a simple reduction to single-level stochastic nonconvex smooth optimization, assuming the stochastic gradient oracle has mean-squared smoothness.
>
> It is shown in [(Arjevani et al., 2023)](https://arxiv.org/pdf/1912.02365.pdf) that with the mean-squared smoothness assumption, $\Omega(1/\epsilon^3)$ complexity is necessary for finding an $\epsilon$-stationary point when using stochastic first-order methods for single-level stochastic nonconvex optimization problems. Note that our problem class is more expressive than the function class considered in [(Arjevani et al., 2023)](https://arxiv.org/pdf/1912.02365.pdf), making our problem harder. This is because standard smoothness is a special case of relaxed smoothness and single-level optimization is a special case of bilevel optimization.
>
> For example, if we consider an easy case where the upper-level function does not depend on the lower-level problem (e.g., $\Phi(x) = f(x)$ such that $\Phi$ is independent of $y^*(x)$) and is mean-squared smooth, then the $\Omega(1/\epsilon^3)$ lower bound in [(Arjevani et al., 2023)](https://arxiv.org/pdf/1912.02365.pdf) can be applied in our setting. Therefore the $\widetilde{O}(1/\epsilon^3)$ complexity achieved in this paper is already optimal up to logarithmic factors in terms of the dependency on $\epsilon$.
>
> **W4. The works [2, 3, 4] should be mentioned in the paper.**
>
> **A4.** Thank you for your suggestion. We will mention [2, 3, 4] in the related work section. However, could you please clarify what [2, 3, 4] refer to? It seems that the references you mentioned are missing in your review.
>
> **C1. Data and instructions are missing.**
>
> **A5.** Thank you for catching this issue. We forgot to include the instructions on how to download the data. We have fixed it (including the data and instructions) in the revised version of the code. Please reach out to AC for the code link.
>
> **C2. Implementation issues.**
>
> **A6.** Please refer to the global rebuttal PDF file for new experimental results and hyperparameter settings.
>
> - **Neumann series.** In practice, we have fixed the implementation of Neumann series using random $\texttt{q}(Q)$ and it does not affect the performance too much. In theory, the expectations of $v_1$ and $v_2$ (defined as the two formulas you wrote) are the same. This means that both $v_1$ and $v_2$ can be applied to our algorithm and analysis, and using either one does not affect the convergence rate of AccBO. We prefer $v_1$ over $v_2$ in practice due to less randomness in $v_1$, leading to slightly better and more stable performance.
> - **SABA.** We have fixed the implementation of SABA and now use a constant step size for SABA. See results in the rebuttal PDF file.
> - **VRBO.** We choose a larger batch size for the outer loop (i.e., at the checkpoint) than for the inner loop. In experiments, we tune the batch size for the outer loop and set it to be twice that of the inner loop. We rerun the two experiments, see results in the rebuttal PDF file.
>
> **C3. The VRBO code and the hypergradient computation method in `ma_soba.py` are missing for the datacleaning task.**
>
> **A7.** The file `ma_soba.py` used in the data-cleaning task is similar to the one used in the AUC task. We have added the implementation of VRBO and the hypergradient computation method in `ma_soba.py` for the data hypercleaning task. **Please reach out to AC for our anonymized code link.**
>
> **Minor Issues**
>
> We will fix the typos in the revised version of the paper and code.

---

> ### Author Response · Authors · 2024-08-09
> **If you haven't received the anonymized code link, please reach out to AC for details.**
>
> Dear Reviewer o6Pr,
>
> Thank you for reviewing our paper. We have carefully addressed your concerns regarding the optimality of our algorithm, the implementation of the inverse Hessian-vector product and SABA, as well as the learning rate and batch size choices for the baselines SABA and VRBO.
>
> We have updated our code and rerun the experiments. The new results, along with the hyperparameter settings, are included in the global rebuttal PDF file. **Please reach out to the AC for the anonymized code link if needed**, as we were notified that "If you were asked by the reviewers to provide code, please send an anonymized link to the AC in a separate comment."
>
> Please let us know if our responses address your concerns accurately. We appreciate your time and efforts and are open to discussing any further questions you may have.
>
> Best regards,
>
> Authors

---

> > ### Comment · Reviewer_o6Pr · 2024-08-10
> >
> > I thank the authors for fixing their code and modified my score accordingly.

---

> > > ### Author Response · Authors · 2024-08-10
> > > **Thank you for your feedback!**
> > >
> > > Dear Reviewer o6Pr,
> > >
> > > We are glad that our responses addressed your concerns. Thank you for reviewing our paper.
> > >
> > > Best,
> > >
> > > Authors

---

### Official Review · Reviewer_W5hu · 2024-07-12

**Soundness:** 3
**Presentation:** 3
**Contribution:** 2
**Rating:** 6
**Confidence:** 3

**Summary:**

This paper investigates a class of stochastic bilevel optimization problems where the upper-level function is nonconvex with potentially unbounded smoothness, and the lower-level problem is strongly convex. To improve the convergence rate, the authors propose a new Accelerated Bilevel Optimization algorithm named AccBO. This algorithm updates the upper-level variable using normalized stochastic gradient descent and the lower-level variable using the stochastic Nesterov accelerated gradient descent algorithm with averaging. The proof shows that AccBO has an oracle complexity of $ \tilde{O}(1/\epsilon^3) $, relying on a novel lemma describing the dynamics of the stochastic Nesterov accelerated gradient descent algorithm under distribution drift. Experimental results demonstrate that AccBO significantly outperforms baseline algorithms in various tasks, achieving the predicted theoretical acceleration.

**Strengths:**

1.The paper introduces AccBO, a novel algorithm that leverages normalized recursive momentum for the upper-level variable and Nesterov momentum for the lower-level variable. This dual approach to acceleration in bilevel optimization under stochastic settings is a significant advancement.

2.AccBO achieves an oracle complexity of  $ \tilde{O}(1/\epsilon^3) $ for finding an $\epsilon$-stationary point. This is a substantial improvement over the previously best-known complexity of        $ \tilde{O}(1/\epsilon^4)  $            for similar problems.The paper provides new proof techniques, particularly in analyzing the dynamics of stochastic Nesterov accelerated gradient descent under distribution drift.

3.The effectiveness of AccBO is empirically verified across various tasks, including deep AUC maximization and data hypercleaning. The results show that AccBO achieves the predicted theoretical acceleration and significantly outperforms baseline algorithms.

4.The paper extends the bilevel optimization framework to handle upper-level functions with unbounded smoothness, which is a realistic scenario in many neural network applications. This makes the proposed algorithm applicable to a broader range of practical problems.

**Weaknesses:**

The convergence analysis for Option I of the algorithm is restricted to the lower-level problem being a one-dimensional quadratic function. This limitation reduces the generality and applicability of the theoretical results for more complex lower-level problems.

**Questions:**

In reference [33], Assumption 2 states $|| \nabla_y f(x, y^*(x)) || \leq M$, while in this paper, Assumption 3.2 states$ || \nabla_y f(x, y) || \leq l_{f,0} $ . Can this assumption be relaxed to a less stringent form?

**Limitations:**

The convergence analysis for Option I of the AccBO algorithm assumes that the lower-level problem is a one-dimensional quadratic function.  This limits the algorithm's applicability to scenarios where the lower-level function deviates significantly from this simple form, especially in higher dimensions or more complex optimization landscapes.

---

> ### Author Rebuttal · Authors · 2024-08-06
>
> **Thank you for taking the time to review our paper. We have addressed each of your concerns below.**
>
> **Q1. The convergence analysis for Option I of the algorithm is restricted to the lower-level problem being a one-dimensional quadratic function. This limitation reduces the generality and applicability of the theoretical results for more complex lower-level problems.**
>
> **A1.** Thank you for your insightful question. Please note that Option II of Algorithm 2 works for the general case of high-dimensional functions and also enjoys an accelerated convergence rate of $\widetilde{O}(1/\epsilon^3)$. Therefore, it can indeed handle complex lower-level problems in high dimensions and is more general than Option I.
>
> One can regard Option I as a way to leverage the nice structure of the lower-level problem, making it easier to implement in practice. When the lower-level function is one-dimensional and quadratic (e.g., deep AUC maximization), we can invoke Option I, making AccBO a single-loop algorithm, which is easier to implement. In the general case, when the lower-level objective is a strongly convex function in high dimensions (e.g., data hypercleaning), we can invoke Option II, resulting in AccBO becoming a double-loop algorithm.
>
> **Q2. In [(Hao et al., 2024)](https://arxiv.org/pdf/2401.09587.pdf), Assumption 2 states** $\\|\nabla_y f(x,y^*(x))\\| \leq M$**, while in this paper, Assumption 3.2 states** $\\|\nabla_y f(x,y)\\| \leq l_{f,0}$**. Can this assumption be relaxed to a less stringent form?**
>
> **A2.** In this paper, we use the Neumann series approach to handle the Hessian inverse and hypergradient approximation. Therefore, we need to upper bound the bias term $\\|\bar{\nabla} f(x_t,\hat{y}\_t) - \nabla\Phi(x_t)\\|$ when proving the convergence of AccBO (see lines 803-804 of Lemma E.1 in Appendix E for details). The assumption $\\|\nabla_y f(x,y)\\| \leq l_{f,0}$ is particularly crucial for deriving Lemma B.5 in Appendix F.1 under our current analysis, which is necessary for proving Lemma E.1.
>
> In contrast, [(Hao et al., 2024)](https://arxiv.org/pdf/2401.09587.pdf) uses the linear system approach to handle the Hessian inverse and hypergradient approximation, so bias terms like $\\|\bar{\nabla} f(x,y) - \nabla\Phi(x)\\|$ do not appear in their analysis. As a result, they only need a slightly relaxed assumption, $\\|\nabla_y f(x,y^*(x))\\| \leq M$, to show the smoothness properties of the bilevel optimization problem.
>
> In fact, many existing works in the bilevel literature that use the Neumann series approach to handle the Hessian inverse require this assumption (i.e., $\\|\nabla_y f(x,y)\\| \leq l_{f,0}$). For example, see Assumption 1 of [(Ghadimi and Wang, 2018)](https://arxiv.org/pdf/1802.02246), Assumption 2 of [(Ji et al., 2021)](https://arxiv.org/pdf/2010.07962), Assumption 1 of [(Hong et al., 2023)](https://arxiv.org/pdf/2007.05170), Assumption 1 of [(Khanduri et al., 2021)](https://arxiv.org/pdf/2102.07367), Assumption 1 of [(Yang et al., 2021)](https://arxiv.org/pdf/2106.04692), and Assumption 1 of [(Chen et al., 2021)](https://proceedings.neurips.cc/paper_files/paper/2021/file/d4dd111a4fd973394238aca5c05bebe3-Paper.pdf).
>
> **Q3. The convergence analysis for Option I of the AccBO algorithm assumes that the lower-level problem is a one-dimensional quadratic function. This limits the algorithm's applicability to scenarios where the lower-level function deviates significantly from this simple form, especially in higher dimensions or more complex optimization landscapes.**
>
> **A3.** We have addressed your concern about this limitation in **A1**. Please see **A1** for details.

---

### Official Review · Reviewer_wLtJ · 2024-07-13

**Soundness:** 2
**Presentation:** 3
**Contribution:** 2
**Rating:** 5
**Confidence:** 3

**Summary:**

This paper considers a class of stochastic bilevel optimization problems where the upper-level function is nonconvex with potentially unbounded smoothness, and the lower-level problem is strongly convex. Their novel algorithms achieve an oracle complexity of $O(1/\epsilon^3)$ to find an $\epsilon$-stationary point.

**Strengths:**

1. The setting is novel: they consider problems with potentially unbounded smoothness.
2. Their complexity strictly improves the state-of-the-art oracle complexity for unbounded smooth nonconvex upper-level problems and strongly convex lower-level problems.

**Weaknesses:**

1. In the algorithm, for the update of the normalized stochastic gradient, it seems that it can be reformulated to tune the learning rate by dividing by the norm of the stochastic gradient. Therefore, the novelty and reasonableness of this step may be unclear.
2. In the experiments, it is common to use the same learning rate for baselines and the proposed algorithms for a fair comparison. However, in the experimental details, the authors provide the best learning rate pairs for different baselines, which seems unusual.

**Questions:**

1. Regarding the main challenge, the authors state why previous algorithms and analyses do not work. However, this explanation is not clear to me. Can you elaborate on how your work addresses the problem of hypergradient bias? Also, how does your potential function argument differ from previous work?

---

> ### Author Rebuttal · Authors · 2024-08-06
>
> **Thank you for taking the time to review our paper. We have addressed each of your concerns below.**
>
> **Q1. In the algorithm, for the update of the normalized stochastic gradient, it seems that it can be reformulated to tune the learning rate by dividing by the norm of the stochastic gradient. Therefore, the novelty and reasonableness of this step may be unclear.**
>
> **A1.** Thank you for your insightful question. We would like to emphasize that we use *normalized stochastic gradient descent with recursive momentum* for updating the upper-level variable $x_t$ to reduce the effects of stochastic gradient noise, as well as the effects of unbounded smoothness and gradient norm. A similar approach of using normalized stochastic gradient descent with standard momentum is outlined in the Section 3.1 of [(Hao et al., 2024)](https://arxiv.org/pdf/2401.09587.pdf).
>
> The novelty and reasonableness of our upper-level update lie in the following two aspects: first, we use recursive momentum update for $m_t$ (line 23 of Algorithm 2) to achieve variance reduction and acceleration, while [Hao et al., (2024)](https://arxiv.org/pdf/2401.09587.pdf) uses moving average estimation for $m_t$; second, we choose a larger learning rate $\eta$ to achieve acceleration. Specifically, for any given small $\epsilon>0$ (i.e., the target gradient norm $\epsilon$), our algorithm's learning rate is $\eta=\widetilde{\Theta}(\epsilon^2)$ , while that of [(Hao et al., 2024)](https://arxiv.org/pdf/2401.09587.pdf) is $\eta=\widetilde{\Theta}(\epsilon^3)$. The large learning rate is crucial for proving acceleration.
>
> **Q2. In the experiments, it is common to use the same learning rate for baselines and the proposed algorithms for a fair comparison. However, in the experimental details, the authors provide the best learning rate pairs for different baselines, which seems unusual.**
>
> **A2.** We would like to clarify that it is indeed a common practice in the bilevel optimization literature to compare different baselines using the best-tuned learning rate pairs. For example, see Appendix A and B of [(Ji et al., 2021)](https://arxiv.org/pdf/2010.07962), Appendix B of [(Yang et al., 2021)](https://arxiv.org/pdf/2106.04692), Section 5 of [(Hong et al., 2023)](https://arxiv.org/pdf/2007.05170), Section 4 of [(Khanduri et al., 2021)](https://arxiv.org/pdf/2102.07367), and Section 4 of [(Chen et al., 2022)](https://arxiv.org/pdf/2102.04671).
>
> **Q3. Regarding the main challenge, the authors state why previous algorithms and analyses do not work. However, this explanation is not clear to me. Can you elaborate on how your work addresses the problem of hypergradient bias? Also, how does your potential function argument differ from previous work?**
>
> **A3.** Thank you for your insightful question, and we apologize for the ambiguity. Most previous works, such as [(Khanduri et al., 2021](https://arxiv.org/pdf/2102.07367); [Yang et al., 2021)](https://arxiv.org/pdf/2106.04692), assume that the upper-level objective function is $L$-smooth. Their algorithms and analyses rely on the $L$-smoothness property to leverage some nice forms of hypergradient bias, which typically depend on the approximation error of the lower-level variable, and either the Neumann series approximation error or the linear system solution approximation error when handling the Hessian inverse. Most importantly, their hypergradient bias does not depend on the (ground-truth) hypergradient $\nabla\Phi(x_t)$ itself.
>
> In contrast, when the upper-level problem is unbounded smooth (i.e., $(L_{x,0}, L_{x,1}, L_{y,0}, L_{y,1})$-smooth as illustrated in Assumption 3.1), the hypergradient bias depends on both the approximation error of the lower-level variable and the hypergradient itself, for example, $\\|\hat{y}_t-y_t^*\\|\\|\nabla\Phi(x_t)\\|$. Please see lines 803-804 of Lemma E.1 in Appendix E for details. These two elements are statistically dependent, necessitating high probability analysis for the lower-level problem, which means the standard expectation analysis used in previous works cannot be applied to our setting. Another recent work, [(Hao et al., 2024)](https://arxiv.org/pdf/2401.09587.pdf), studies the same setting as ours, but their algorithm does not achieve acceleration.
>
> Let us provide more details to explain how we handle the hypergradient bias (lines 803-804 of Lemma E.1). Specifically:
> - In Lemma D.6, we control the first term $\\|\epsilon_t\\| = \\|m_t - \mathbb{E}_t[\bar{\nabla} f(x_t,\hat{y}_t;\bar{\xi}_t)]\\|$ by Jensen's inequality and induction.
> - In Lemma B.4, we control the second term $\\|\mathbb{E}_t[\bar{\nabla} f(x_t,\hat{y}_t;\bar{\xi}_t)] - \bar{\nabla} f(x_t,\hat{y}_t)\\|$ by leveraging the Neumann series approximation result from [(Khanduri et al., 2021)](https://arxiv.org/pdf/2102.07367).
> - In Lemma B.5, we control the third term $\\|\bar{\nabla} f(x_t,\hat{y}_t) - \nabla\Phi(x_t)\\|$ using the triangle inequality and the relaxed smoothness property of the upper-level function.
>
> With a suitable choice of parameters, the hypergradient bias can be small on average.
>
> Regarding the potential function argument, we directly use the objective $\Phi(x)$ itself as the potential function. This approach differs from previous works that incorporate both the function value and the approximation error from the lower-level problem. Our problem setting and analysis do not align with those in previous works, necessitating this different approach.

---

> ### Author Response · Authors · 2024-08-11
> **Looking forward to post-rebuttal feedback!**
>
> Dear Reviewer wLtJ,
>
> Thank you for reviewing our paper. We have carefully addressed your concerns regarding the novelty and reasonableness of the learning rate choice, the fairness of comparisons between our proposed algorithm and other baselines, and how our work tackles the challenge of hypergradient bias.
>
> Please let us know if our responses address your concerns accurately. If so, we kindly ask you to consider raising the rating of our work. We appreciate your time and efforts and are open to discussing any further questions you may have.
>
> Best,
>
> Authors

---

> > ### Comment · Reviewer_wLtJ · 2024-08-11
> >
> > I appreciate the authors' reply. I understand the rationale for a larger learning rate from the convergence rate perspective. However, I still find the gradient normalization on line 24 in Algorithm 2 somewhat problematic in claiming this larger learning rate. From my perspective, when the norm of $m_t$ is larger, it would allow for a higher learning rate. Please correct me if I'm mistaken.

---

> ### Author Response · Authors · 2024-08-11
> **Thanks for your reply!**
>
> Thank you very much for your reply, below we provide further explanations to your additional concerns.
>
> - First, we would like to clarify that the learning rate should be $\eta$ instead of $\frac{\eta}{\\|m_t\\|}$ when considering normalized stochastic gradient descent. For example, see Algorithm 1 of [(Cutkosky and Mehta, 2020)](https://arxiv.org/pdf/2002.03305) (Input: $\dots$, learning rate $\eta$, $\dots$), Algorithm 1 of [(Jin et al., 2021)](https://arxiv.org/pdf/2110.12459) (Input: $\dots$, learning rate $\gamma$, $\dots$), and Algorithms 1 and 2 of [(Liu et al., 2023b)](https://arxiv.org/pdf/2302.06763) (middle of page 6, "the step size $\eta$ is chosen by balancing every other term to get the right convergence rate") for details. We follow the same practice and refer the learning rate of the update $x_{t+1} = x_t - \eta\frac{m_t}{\\|m_t\\|}$ (i.e., line 24 of Algorithm 2) as $\eta$, rather than $\frac{\eta}{\\|m_t\\|}$. Additionally, our choice of learning rate $\eta$ is independent of the magnitude of $\\|m_t\\|$, please see Theorem 4.1 and Eq. (55) in Appendix E on page 31 for more details.
>
> - Second, please note that we study the same problem setting as [(Hao et al., 2024)](https://arxiv.org/pdf/2401.09587.pdf), where the upper-level problem is unbounded smooth (i.e., $(L_{x,0}, L_{x,1}, L_{y,0}, L_{y,1})$-smooth as illustrated in Assumption 3.1). Both our proposed Algorithm 2 (i.e., AccBO) and Algorithm 1 in [(Hao et al., 2024)](https://arxiv.org/pdf/2401.09587.pdf) use normalization and choose a fixed learning rate $\eta$. We mention allowing a larger learning rate because, for any given small $\epsilon>0$ (i.e., the target gradient norm $\epsilon$), AccBO's learning rate is $\eta=\widetilde{\Theta}(\epsilon^2)$, while that of [(Hao et al., 2024)](https://arxiv.org/pdf/2401.09587.pdf) is $\eta=\widetilde{\Theta}(\epsilon^3)$. In other words, we say that we choose a larger learning rate $\eta$ specifically for comparison with [(Hao et al., 2024)](https://arxiv.org/pdf/2401.09587.pdf). The large learning rate is crucial for achieving acceleration: it improves the complexity from $\widetilde{O}(\epsilon^{-4})$ as in [(Hao et al., 2024)](https://arxiv.org/pdf/2401.09587.pdf) to $\widetilde{O}(\epsilon^{-3})$ as in this paper.
>
> Please let us know if you have further questions. Thanks!
>
> Best,
>
> Authors

---

> > ### Comment · Reviewer_wLtJ · 2024-08-11
> >
> > Thank you for your answer. I will raise my score accordingly.
> >
> > Best.

---

> > > ### Author Response · Authors · 2024-08-11
> > > **Thank you!**
> > >
> > > Dear Reviewer wLtJ,
> > >
> > > We are glad that our answer addressed your concerns. Thank you for reviewing our paper.
> > >
> > > Best,
> > >
> > > Authors

---

### Author Rebuttal · Authors · 2024-08-06

**General Response to All Reviewers**

Thank you to all the reviewers for taking the time to review our paper and provide valuable feedback. We have addressed each of your concerns individually, and below is a summary of the key changes made during the rebuttal phase.

1. We would like to clarify that Option I of AccBO is not a drawback of our approach, it is actually a benefit. Option I leverages the simpler structure of the lower-level problem, making it easier to implement. For one-dimensional quadratic functions (e.g., deep AUC maximization), Option I turns AccBO into a single-loop algorithm, simplifying implementation. Option II makes AccBO a double-loop algorithm, which works for the general case of high-dimensional strongly convex functions (e.g., data hypercleaning) and also enjoys an accelerated convergence rate of $\widetilde{O}(1/\epsilon^3)$, the same as Option I. This makes it capable of handling complex high-dimensional lower-level problems and more general than Option I.

2. We have updated the code and experiments, and our original implementation on Neumann series is not wrong. We have added new experiments based on the reviewer's request (e.g., different learning rate schemes and batch sizes), and the results are reported in the global rebuttal PDF file. It shows that our algorithm AccBO still significantly outperforms other methods. If you want to check the updated code, please reach out to AC (we have sent the anonymized code link to the AC).

3. We have fixed the typos and minor issues, and we will update them in the revised version of the paper.

---

### Decision · Program_Chairs · 2024-09-25

**Decision:**

Accept (poster)

**Comment:**

There was some disagreement among the reviewers about this submission. Based on re-reading the whole discussion (taking into account reviewer's levels of expertise) and my own evaluation of the paper, I recommend to accept.